# Nance-Horan Syndrome-like 1 protein negatively regulates Scar/WAVE-Arp2/3 activity and inhibits lamellipodia stability and cell migration

Ah-Lai Law[1,5], Shamsinar Jalal[1,8], Tommy Pallett [1,8], Fuad Mosis[1], Ahmad Guni[1], Simon Brayford[2], Lawrence Yolland[2], Stefania Marcotti [2], James A. Levitt[3], Simon P. Poland[3], Maia Rowe-Sampson[1,2], Anett Jandke [1,6], Robert Köchl [4], Giordano Pula[1,7], Simon M. Ameer-Beg [3], Brian Marc Stramer [2] & Matthias Krause [1✉]

Cell migration is important for development and its aberrant regulation contributes to many diseases. The Scar/WAVE complex is essential for Arp2/3 mediated lamellipodia formation during mesenchymal cell migration and several coinciding signals activate it. However, so far, no direct negative regulators are known. Here we identify Nance-Horan Syndrome-like 1 protein (NHSL1) as a direct binding partner of the Scar/WAVE complex, which co-localise at protruding lamellipodia. This interaction is mediated by the Abi SH3 domain and two binding sites in NHSL1. Furthermore, active Rac binds to NHSL1 at two regions that mediate leading edge targeting of NHSL1. Surprisingly, NHSL1 inhibits cell migration through its interaction with the Scar/WAVE complex. Mechanistically, NHSL1 may reduce cell migration efficiency by impeding Arp2/3 activity, as measured in cells using a Arp2/3 FRET-FLIM biosensor, resulting in reduced F-actin density of lamellipodia, and consequently impairing the stability of lamellipodia protrusions.

[1] Krause Group, Randall Centre for Cell and Molecular Biophysics, King's College London, New Hunt's House, Guy's Campus, London SE1 1UL, UK. [2] Stramer Group, Randall Centre for Cell and Molecular Biophysics, King's College London, New Hunt's House, Guy's Campus, London SE1 1UL, UK. [3] Ameer-Beg Group, Richard Dimbleby Cancer Research Laboratories, Comprehensive Cancer Centre, School of Cancer and Pharmaceutical Sciences, King's College London, New Hunt's House, Guy's Campus, London SE1 1UL, UK. [4] School of Immunology and Microbial Sciences, King's College London, Guy's Campus, London SE1 1UL, UK. [5] Present address: School of Life Sciences, University of Bedfordshire, Luton LU1 3JU, UK. [6] Present address: Immunosurveillance Laboratory, The Francis Crick Institute, 1 Midland Road, London NW1 1AT, UK. [7] Present address: Institute for Clinical Chemistry and Laboratory Medicine, University Medical Center Hamburg (UKE), Martinistrasse 52, O26, 20246 Hamburg, Germany. [8] These authors contributed equally: Shamsinar Jalal, Tommy Pallett. ✉email: Matthias.Krause@kcl.ac.uk

Cell migration is essential for embryonic development, homoeostasis and wound healing, and its deregulation causes developmental defects, impaired wound healing and cancer metastasis[1]. Mesenchymal cell migration depends on the polymerisation of actin filaments, which push the plasma membrane forward to form membrane extensions such as lamellipodia. The establishment of adhesions underneath lamellipodia stabilise protrusions allowing cells to advance[2]. Lamellipodium formation is controlled by Rac, a small GTPase of the Rho family. Active Rac, tyrosine phosphorylation and binding to phosphoinositides activates the Scar/WAVE complex at the very edge of lamellipodia and thereby recruits the Arp2/3 complex to nucleate branched actin networks. The Scar/WAVE complex is a heteropentameric, autoinhibited protein complex, consisting of Scar/WAVE1-3, Nap1, Sra1/Pir121, Abi1-3 and HSPC300[2–4]. We have previously shown that Lamellipodin (Lpd) localises to the very edge of lamellipodia[5] and directly binds active Rac and the Scar/WAVE complex[6]. Rac increases the interaction between Lpd and the Scar/WAVE complex[6]. Lpd functions to promote cell migration via the Scar/WAVE complex[6,7], which is consistent with a positive role for the Scar/WAVE complex in enhancing migration[8–11]. We postulate that the Scar/WAVE complex needs also to be negatively regulated at the leading edge to tightly control lamellipodial protrusion dynamics and cell steering. However, so far, it is not known how the Scar/WAVE complex is directly inhibited at the leading edge.

Here, we identify NHSL1 (Nance–Horan Syndrome-like 1) protein as a negative regulator of cell migration and we find that this is mediated by its interaction with the Scar/WAVE complex. We show that NHSL1 directly binds to the Scar/WAVE complex and co-localises with it at the very edge of protruding lamellipodia. We find that active Rac binds to NHSL1 at two regions of the protein, which mediate leading-edge targeting of NHSL1. The negative regulatory function of NHSL1 in cell migration may be due to its role in lamellipodia since we show that it reduces lamellipodia stability. NHSL1 acts to reduce Arp2/3 activity, which is consistent with our finding that NHSL1 reduces the F-actin content of lamellipodia via its interaction with the Scar/WAVE complex. In NHSL1 CRISPR knockout cells we observe a reduction in total cellular Scar/WAVE2, Arp2/3 complex and F-actin levels indicating the permanent over-activation of Scar/WAVE and Arp2/3 complexes caused by loss of NHSL1 may lead to their degradation. Taken together, our data suggest that NHSL1 negatively regulates the Scar/WAVE complex, and hence reduces Arp2/3 activity, to control lamellipodia stability and consequently cell migration efficiency.

## Results

**NHSL1 localises to the very edge of lamellipodia.** NHSL1 belongs to the poorly investigated Nance–Horan Syndrome protein family along with Nance–Horan Syndrome (NHS) and NHSL2 proteins. Mutations in the NHS gene cause Nance–Horan syndrome, which is characterised by dental abnormalities, developmental delay and congenital cataracts[12–16]. Expression analysis of NHSL1 by northern blot revealed a 7.5 kb ubiquitously expressed isoform and several shorter isoforms with a tissue-specific distribution (Fig. 1a). In contrast, NHS expression was detected in the brain, kidney, lung and thymus[13] suggesting that the NHS protein has evolved a more specialised function. We generated a rabbit polyclonal anti-NHSL1 antiserum (4457) (Fig. 1e) and a monoclonal antibody (clone C286F5E1). Both NHSL1 antibodies detect endogenous NHSL1 as a 260 kDa protein by immunoblotting in the melanoma cell line B16-F1 and the breast epithelial cell line MCF10A (Fig. 1b–d and Supplementary Fig. 1a–c). B16-F1 cells display large lamellipodia when plated on 2D substrates and have been used extensively in other studies to

characterise lamellipodia dynamics and cell migration[17] and are thus used throughout this study.

NHS localises to lamellipodia but the function of NHS and NHS-like proteins in cell migration are not known[14]. We found that NHSL1 also localises to the very edge of protruding lamellipodia and, in addition, to vesicular structures when tagged with EGFP either N- or C-terminally (Fig. 1f, g and Supplementary Movie 1). This localisation is not due to space-filling as B16-F1 cells display very thin lamellipodia and subtraction of the intensity of co-expressed cytoplasmic mScarlet does not remove the leading-edge signal of EGFP-NHSL1 (Supplementary Fig. 1d and Supplementary Movie 2). We previously showed that Lpd also localises at the very edge of lamellipodia[5]. When we co-expressed NHSL1-EGFP with mScarlet-Lpd in B16-F1 cells we found that both co-localise at the edge of lamellipodia (Supplementary Fig. 1e and Supplementary Movie 3). This prompted us to test whether both proteins can form a complex in cells. We tested this by co-expressing EGFP-Lpd with Myc-NHSL1 in HEK cells but could not detect an interaction after pulling down EGFP-Lpd with GFP-trap beads (Supplementary Fig. 1f).

**NHSL1 negatively regulates cell migration.** To explore the role of NHSL1 in cell migration, we created stable NHSL1 CRISPR knockout B16-F1 cell lines by knocking in a stop codon into exon 2 (Supplementary Fig. 2a), which is common to all isoforms. Clonal cell lines were tested by western blot and qPCR analysis, which revealed that NHSL1 expression was greatly reduced in clone CRISPR 2 but less reduced in clone CRISPR 21 (Fig. 2a, b and Supplementary Fig. 3a, e). The partially reduced mRNA and protein levels (Fig. 2a, b and Supplementary Fig. 3a, e) in the CRISPR 21 cell line are consistent with Cas9 induced indels only in some alleles (Supplementary Fig. 2a), whereas the absence of NHSL1 wild-type alleles (Supplementary Fig. 2a, b) and mRNA in the CRISPR 2 cell line (Supplementary Fig. 3e) indicates that the CRISPR 2 line represents a full NHSL1 knockout.

In random cell migration assays on fibronectin, cell speeds significantly increased by 52% for both NHSL1 CRISPR clones compared to wild-type B16-F1 cells (Fig. 2c, d and Supplementary Fig. 3b–d). Traditionally, cell migration persistence is measured as the directionality ratio: the ratio of the straight-line distance and the total trajectory path length of each track. This leads to inaccurate measurements as this ratio is affected by different track lengths as well as cell speeds[18]. Instead, we quantified cell persistence using the Dunn persistence method[6] by calculating the directionality ratio over a short interval of the movie, defined by a time ratio, TR, and then averaging over all these intervals comprising the whole track to obtain the "Mean Track Persistence" (MTP). This analysis revealed that cell migration persistence was also significantly increased by 53 and 51% for the CRISPR 2 and CRISPR 21 cell lines, respectively (Fig. 2c, e). This increase in persistence could also be seen when we calculated the directionality ratio over time (Supplementary Fig. 4a) and the direction autocorrelation, a measure of how the angle of displacement vectors correlate with themselves which is independent of speed[18,19] (Supplementary Fig. 4b). Analysing mean square displacement also indicated that NHSL1 CRISPR 2 and CRISPR 21 cells explore a larger area (Fig. 2f). In agreement, we found that directional migration into a scratch wound was significantly increased in NHSL1 knockdown MCF10A cells (Supplementary Fig. 5a–c and Supplementary Movie 4). Together these results suggest that NHSL1 negatively regulates cell migration speed and persistence.

**NHSL1 is a binding partner of active Rac.** To characterise the region of NHSL1 required for leading-edge recruitment (Fig. 1f, g), four EGFP-tagged fragments covering the entire length of NHSL1

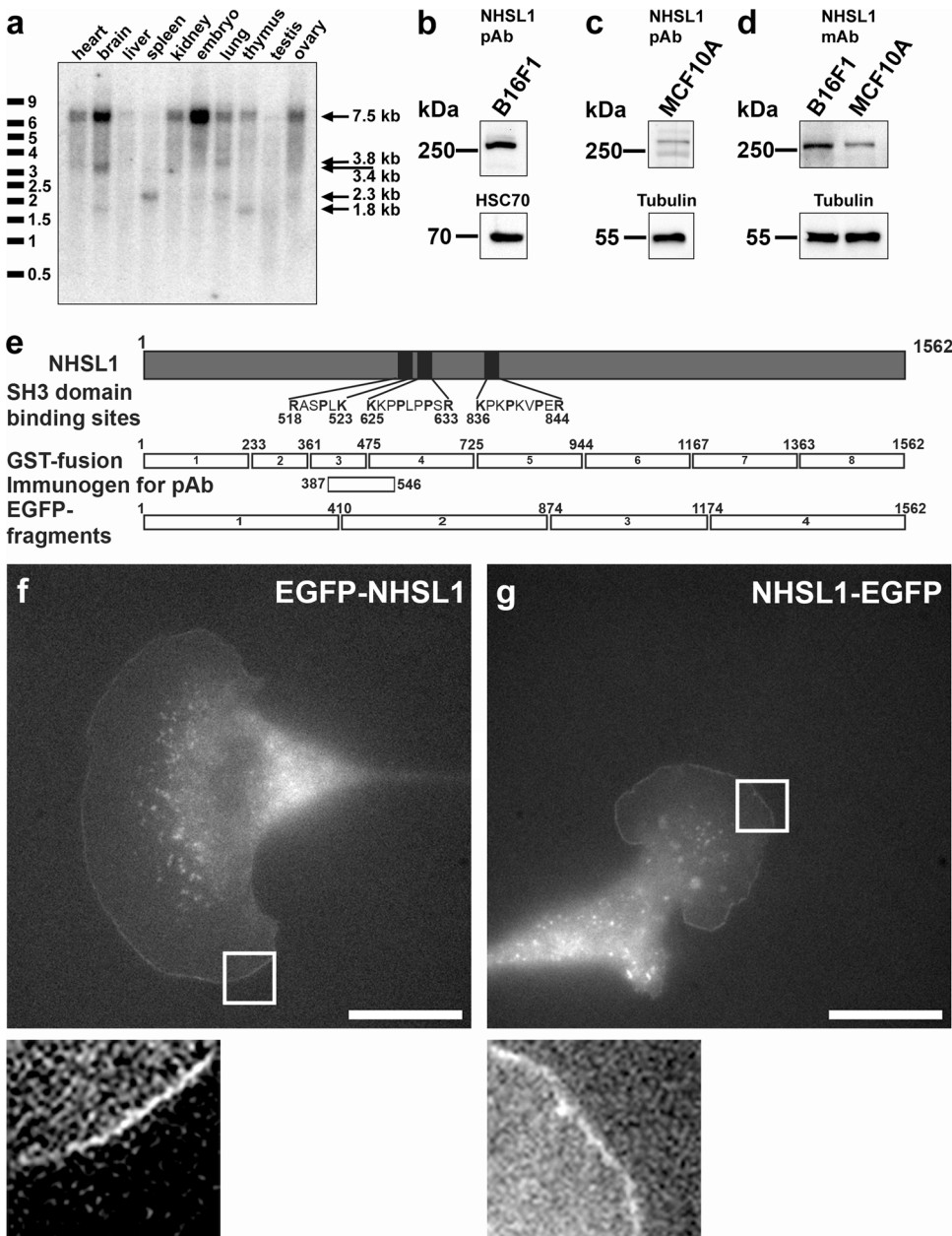

**Fig. 1 NHSL1 localises to the very edge of lamellipodia. a** Northern blot analysis of NHSL1 expression in different murine tissues. Arrows show a large 7.5 kb isoform that is ubiquitously expressed and various shorter isoforms with varying tissue specific distribution. **b–d** Western blots detecting NHSL1 protein in the indicated cell lines using the **b, c** polyclonal (4457) or **d** monoclonal antibody (C286F5E1). For full blots see Supplementary Fig. 1a–c. **e** Scheme of NHSL1 protein (grey rectangle) containing three putative SH3 binding sites (indicated black boxes). GST-fusion proteins 1–8 and EGFP-fusion proteins 1–4 of NHSL1 are indicated (white rectangles). Immunogen for the NHSL1 polyclonal antibody is indicated. **f, g** N-terminal EGFP-tagged NHSL1 (**f**) and C-terminal EGFP-tagged NHSL1 (**g**) were expressed in B16-F1 cells and, after plating on laminin, localisation of both EGFP-tagged NHSL1 constructs were imaged live. Scale bar: 20 μm. Inset represents a magnified view of the white box. Scale bar in the inset: 1.25 μm. Representative blots or images shown from **a** one and **b–d**, **f**, **g** three independent biological experiments. See also related Supplementary Movie 1.

(Fig. 3a) were expressed in B16-F1 cells. Live-cell imaging revealed that only fragments 2 and 3 were localised to lamellipodia (Fig. 3d, e and Supplementary Movie 5). In addition, fragments 1 and 3 were detected at vesicular structures (Fig. 3c, e and Supplementary Movie 5).

Since Rac is an important regulator of lamellipodia formation, we hypothesised that Rac may bind NHSL1 for its recruitment to lamellipodia. We, therefore, tested for interaction between dominant active (DA) Myc-tagged Rac1-Q61L and EGFP-tagged NHSL1 or EGFP only as a control in HEK cells. Myc-tagged DA Rac1 was pulled down with Myc-trap beads and interaction with EGFP-NHSL1 was evaluated by western blot against EGFP. This analysis revealed that NHSL1 is in complex with active Rac (Fig. 3g and Supplementary Fig. 6a). This was verified in a reciprocal experiment in which EGFP-Rac1-Q61L or EGFP was pulled down with GFP-trap beads and the interaction with Myc-NHSL1 was evaluated by western blot against Myc (Supplementary Fig. 6b). Again, we observed an interaction between dominant active Rac and NHSL1 suggesting that NHSL1 is a binding partner of active Rac.

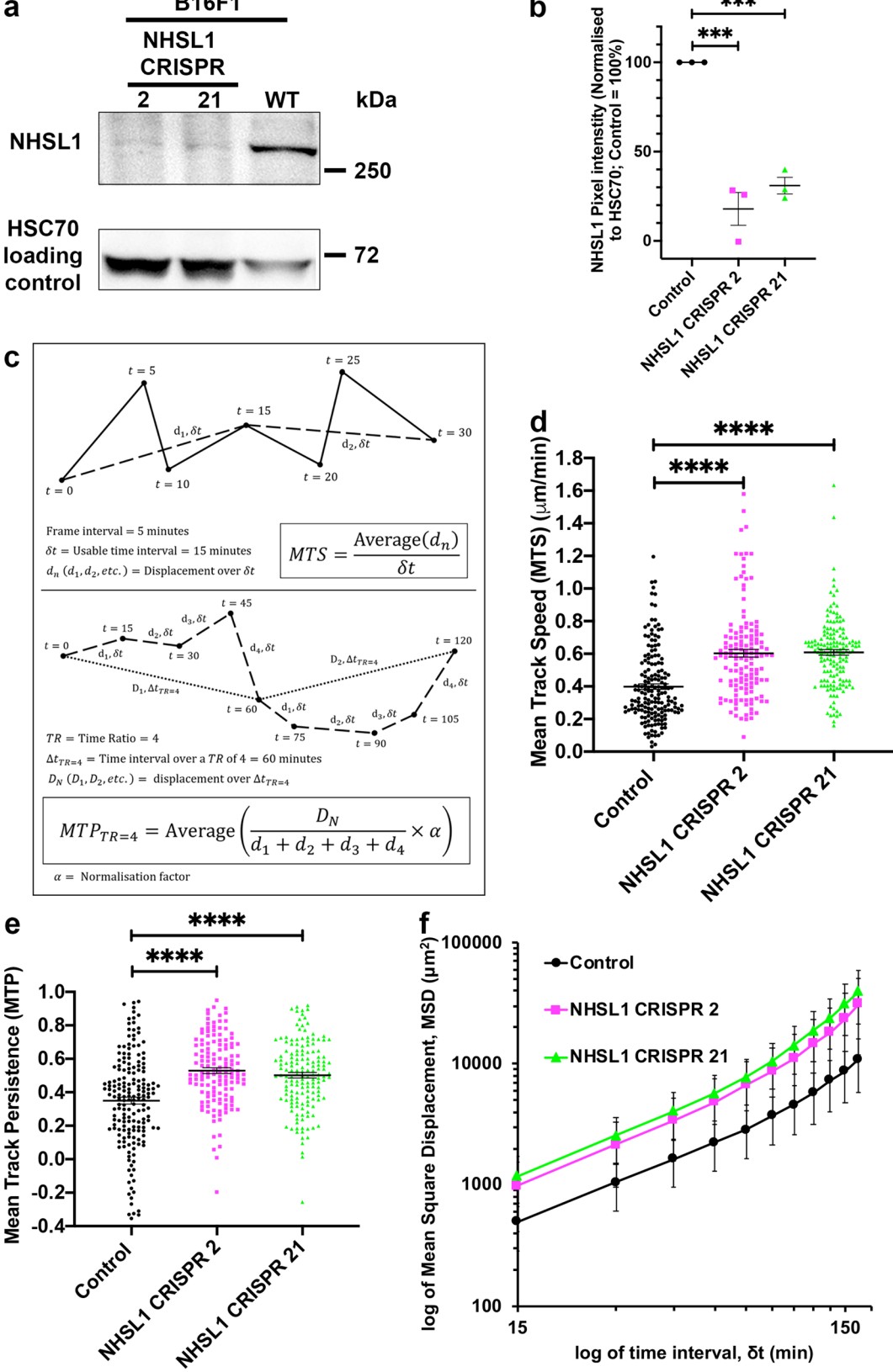

**Fig. 2 NHSL1 negatively regulates cell migration and persistence. a** Western blot showing the extent of reduction of NHSL1 expression in the clonal NHSL1 CRISPR B16-F1 cell lines 2 and 21 were probed with polyclonal NHSL1 antibodies and HSC70 as a loading control. Representative blot from three independent biological repeats. **b** Quantification of western blots in **a** normalised to the HSC70 loading control; Results are mean values ± SEM (error bars); one-way ANOVA: $P < 0.0001$; $F_{(2,6)} = 54.85$; Dunnett's multiple comparisons tests: control vs. CRISPR 2: ***$P = 0.0001$; control vs. CRISPR 21: ***$P = 0.0003$ from three independent biological repeats. **c** Schematic diagram of the definition of Mean Track Speed (MTS) and Mean Track Persistence (MTP). **d** Cell migration speed of both NHSL1 CRISPR clones 2 and 21 in random cell migration assays was significantly increased by an average of ~52% compared to wild-type control B16-F1 cells; Mean Track Speed (MTS) ($\delta t = 15$mins, TR = 4; see 'Methods' for calculation). One-way ANOVA: $P < 0.0001$; $F_{(2,469)} = 41.87$; Dunnett's multiple comparisons test: Results are mean ($\mu$m/min) ± SEM (error bars): control: 0.398 ± 0.017 $\mu$m/min; CRISPR 2: 0.603 ± 0.024 $\mu$m/min; control vs. CRISPR 2: ****$P < 0.0001$; CRISPR 21: 0.609 ± 0.017 $\mu$m/min; control vs. CRISPR 21: ****$P < 0.0001$. **e** Cell migration persistence was significantly increased by 53% or 51% for the NHSL1 CRISPR 2 or 21 cell lines, respectively. Mean Track Persistence (MTP) ($\delta t = 15$mins, TR = 4; see 'Methods' for calculation). One-way ANOVA: $P < 0.0001$; $F_{(2,469)} = 28.01$; Dunnett's multiple comparisons test: control: 0.350 ± 0.021; CRISPR 2: 0.529 ± 0.017; control vs. CRISPR 2: ****$P < 0.0001$; CRISPR 21: 0.502 ± 0.016; control vs. CRISPR 21: ****$P < 0.0001$; results are mean (this is dimensionless, see panel **c** for explanation of MTP) ± SEM (error bars); **f** log-log plot of Mean Square Displacement (MSD) of control B16-F1 cells, NHSL1 CRISPR clones 2 and 21 in random cell migration assays from (d,e); Results are mean values ± SEM (error bars). **d**–**f** $n = 177$ (wild type), 140 (NHSL1 clones 2) and 156 (NHSL1 clones 21) cells from four independent biological repeats. Source data are provided as a Source Data file.

We again expressed the four EGFP-tagged NHSL1 fragments to determine which fragment may mediate the interaction with Myc-tagged dominant active Rac1 and pulled down Myc-DA-Rac1 with Myc-trap beads. A western blot against EGFP showed robust pulldown of EGFP-NHSL1 fragment 2 and weak pulldown of fragment 3 by active Rac (Fig. 3h, i). To further delineate the Rac binding sites in NHSL1, we generated nine overlapping subfragments covering fragments 2 and 3 (Supplementary Fig. 7a). Again, Myc-trap pulldown of Myc-DA-Rac1 from HEK lysates co-expressing EGFP-tagged NHSL1 subfragments 1 to 9 revealed that subfragments 1 and 6, which overlap with each other at the N terminus of fragment 2, and subfragment 9, which is centrally located in fragment 3, are in complex with active Rac (Supplementary Fig. 7b).

Taken together, our data suggest that NHSL1 may be recruited to the leading edge by membrane-associated active Rac binding directly or indirectly to two regions in NHSL1 located at amino acid 410-563 in fragment 2 and at amino acid 946-1099 in fragment 3.

**NHSL1 interacts with the Scar/WAVE complex via Abi.** As the Scar/WAVE complex localises to the leading edge of cells[2], we tested whether NHSL1 co-localises with it at this site. Using live-cell imaging, we observed that NHSL1-EGFP co-localises with mScarlet-tagged Abi1 and Nap1, two components of the Scar/WAVE complex at the very edge of protruding lamellipodia (Fig. 4a, b and Supplementary Movie 6). In agreement, we also found that endogenous NHSL1 co-localises with Abi1, Scar/WAVE1 and Scar/WAVE2 at this site (Fig. 4c–e; see line scans in Supplementary Fig. 8a–h).

To explore a potential interaction with the Scar/WAVE complex, Myc-tagged components of the Scar/WAVE complex and EGFP-NHSL1 were expressed in HEK cells. Immunoprecipitation of EGFP-NHSL1 revealed that NHSL1 interacts with the Scar/WAVE complex (Fig. 5a and Supplementary Fig. 9a, b). To confirm whether the respective endogenous proteins can be found in a complex in cells we immunoprecipitated endogenous NHSL1 and observed co-immunoprecipitation with Abi1 and Scar/WAVE2 from B16-F1 and MCF10A cell lysates (Fig. 5b, c).

Since NHSL1 contains several putative SH3 binding sites we hypothesised that the SH3 domain of Abi, which is part of the Scar/WAVE complex, may mediate the interaction with NHSL1. The interaction between NHSL1 and Abi was tested by GST-pulldown experiments with purified Abi1-SH3 and c-Abl-SH3 from HEK lysates containing EGFP-NHSL1 or EGFP only as control. c-Abl-SH3 was used as another control since binding to Abi may be mediated indirectly by binding with the Abi ligand c-Abl which also contains a SH3 domain[20,21]. We found that while the EGFP negative control did not bind (Supplementary

Fig. 9c), EGFP-NHSL1 specifically bound to the SH3 domain of Abi and not to the SH3 domain of c-Abl (Fig. 5d).

We next explored whether the SH3 domain of Abi is necessary for the interaction between NHSL1 and the Scar/WAVE complex. NHSL1 was immunoprecipitated from HEK cell lysates containing Myc-NHSL1 and all five Myc-tagged components of the Scar/WAVE complex including either full-length Abi1 (Abi1-full-length) or an Abi1 construct that lacked the SH3 domain (Abi1-delta-SH3). Co-immunoprecipitation was detected with an antibody against the Myc-tag (Fig. 5e). Quantification of co-immunoprecipitation of NHSL1 with the Scar/WAVE complex revealed that expression of Abi1-delta-SH3 disrupted the interaction between NHSL1 and the Scar/WAVE complex (Fig. 5e, f). This suggests that the SH3 domain of Abi mediates the interaction of NHSL1 with the Scar/WAVE complex.

To verify that the Scar/WAVE complex properly assembles in cells when all components of the Scar/WAVE complex are expressed as tagged proteins, we expressed Myc-tagged PIR121, Nap1, HSPC300 and Scar/WAVE2 together with either EGFP-Abi1-full-length, EGFP-Abi1-delta-SH3, or EGFP in HEK cells. After GFP-trap pulldown from cell lysates, we detected bound Myc-tagged proteins with an antibody against the Myc-tag. This analysis revealed that indeed all tagged components of the Scar/WAVE complex faithfully associate with each other as a complex (Supplementary Fig. 9d).

To test whether the interaction between Abi and NHSL1 is direct and to map the SH3 binding site in NHSL1, we performed a far-western blot experiment. We purified from *E. coli* eight GST-fusion proteins covering the entire length of NHSL1 (Fig. 1e and Supplementary Fig. 10a, b), which were separated on SDS-PAGE, followed by blotting onto the membrane. We overlaid this membrane with purified MBP-tagged full-length Abi1 (MBP-Abi1-full-length) or an MBP fusion protein with Abi1 in which the SH3 domain had been deleted (MBP-Abi1-delta-SH3) or MBP as control. The far-western overlay showed that only fragments 4 and 5 of NHSL1 directly interacted with wild-type Abi but neither with Abi missing the SH3 domain nor MBP on its own (Supplementary Fig. 10a). In agreement, fragments 4 and 5 contain three putative SH3 binding sites suggesting that Abi binds directly via its SH3 domain to NHSL1.

Next, we explored whether these putative SH3 binding sites were sufficient for the interaction with Abi. We mutated SH3 binding sites 1 and 2 (site 1 + 2), or sites 2 and 3 (site 2 + 3) or all three sites together (site 1 + 2 + 3) in full-length NHSL1 and expressed the EGFP-tagged mutant and wild-type cDNA's together with Myc-tagged Abi1 in HEK cells. After GFP-trap pulldown from lysates, western blot against the Myc-tag revealed that only EGFP-NHSL1 (site 2 + 3) and NHSL1 (sites 1 + 2 + 3) showed loss of interaction

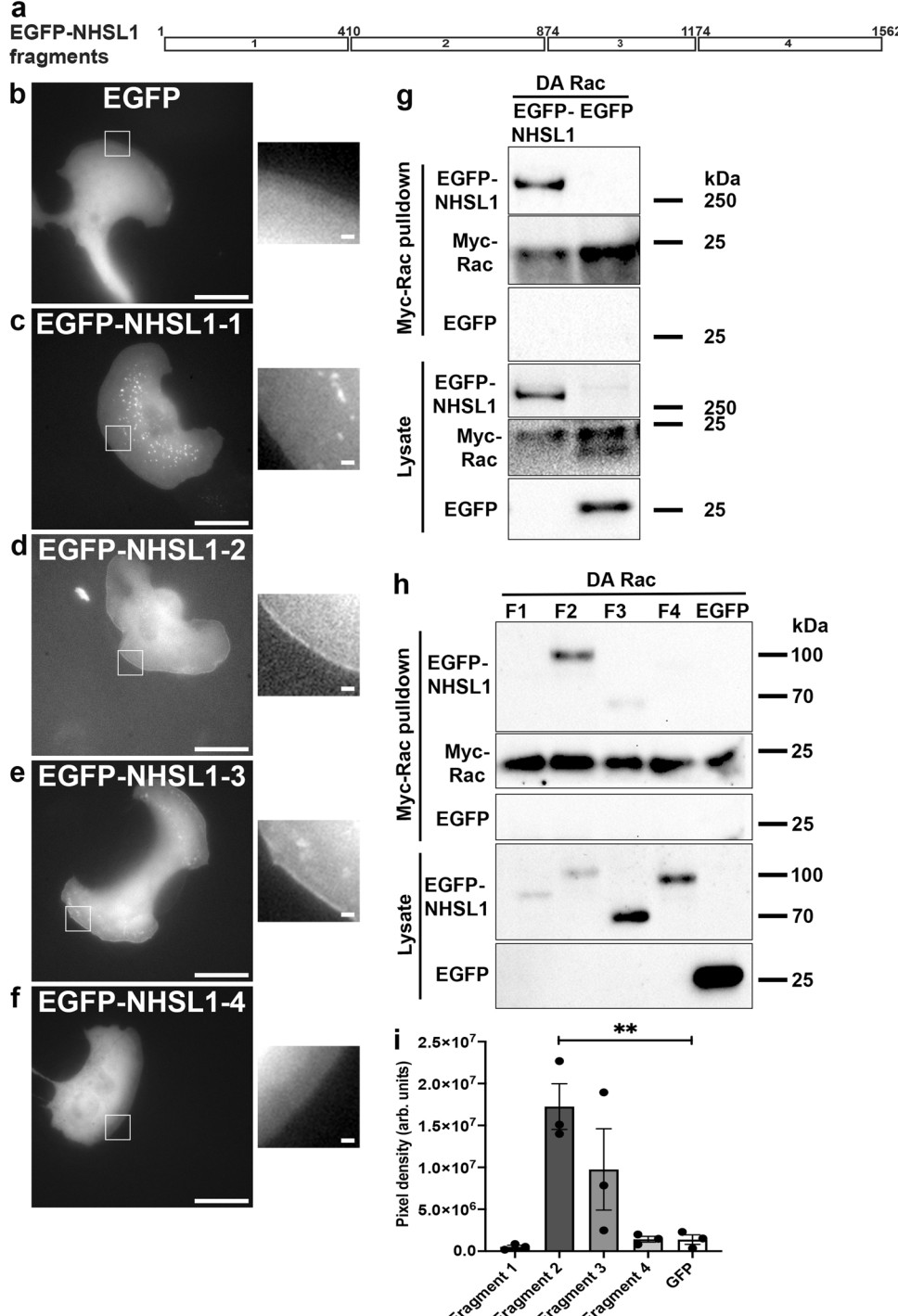

**Fig. 3 NHSL1 is a binding partner of active Rac. a** Scheme of NHSL1 protein: EGFP-fusion proteins 1 to 4 of NHSL1 are indicated (white rectangles). **b–f** Four EGFP-tagged fragments covering the entire length of NHSL1 (Figs. 1e and 3a) were generated and expressed in B16-F1 cells (**c–f**) along with EGFP as a negative control (**b**). Insets show enlarged pictures of the boxed areas from stills taken from live-cell imaging. Fragments 2 and 3 localise to the very edge of lamellipodia similar to full-length NHSL1. Representative images from three independent experiments. Scale bar: 20 μm. Inset represents a magnified view of the white box. Scale bar in inset: 1.25 μm. See also related Supplementary Movie 5. **g** Western blot showing that dominant active (DA) Rac pulls down NHSL1 using Myc-trap beads from HEK cell lysates expressing Myc-tagged DA Rac co-expressed with EGFP-tagged NHSL1 or EGFP only as control. Representative blots from three independent biological repeats (see Supplementary Fig. 6a for full western blots). **h** Western blot showing Myc-tagged DA Rac pulldowns of the four NHSL1 fragments (see panels **a** and **b–e**) from HEK cells expressing the EGFP-tagged fragments covering the entire length of NHSL1 or EGFP only as control co-expressed with Myc-tagged DA Rac. Representative blots from three independent experiments. **i** Quantification of band intensity of chemiluminescence from **h** imaged with a charge-coupled device (CCD) camera shows that Myc-tagged DA Rac binds significantly to fragment 2 compared to GFP. The amounts of EGFP-NHSL1 fragments were normalised to the amount of Myc-Rac that was pulled down. Bars indicate mean ± SEM (error bars), $n = 3$ biological repeats. One-way ANOVA: $P = 0.003$, $F(4,10) = 8.473$; and Dunnett's multiple comparisons tests: **$P = 0.004$. Source data are provided as a Source Data file.

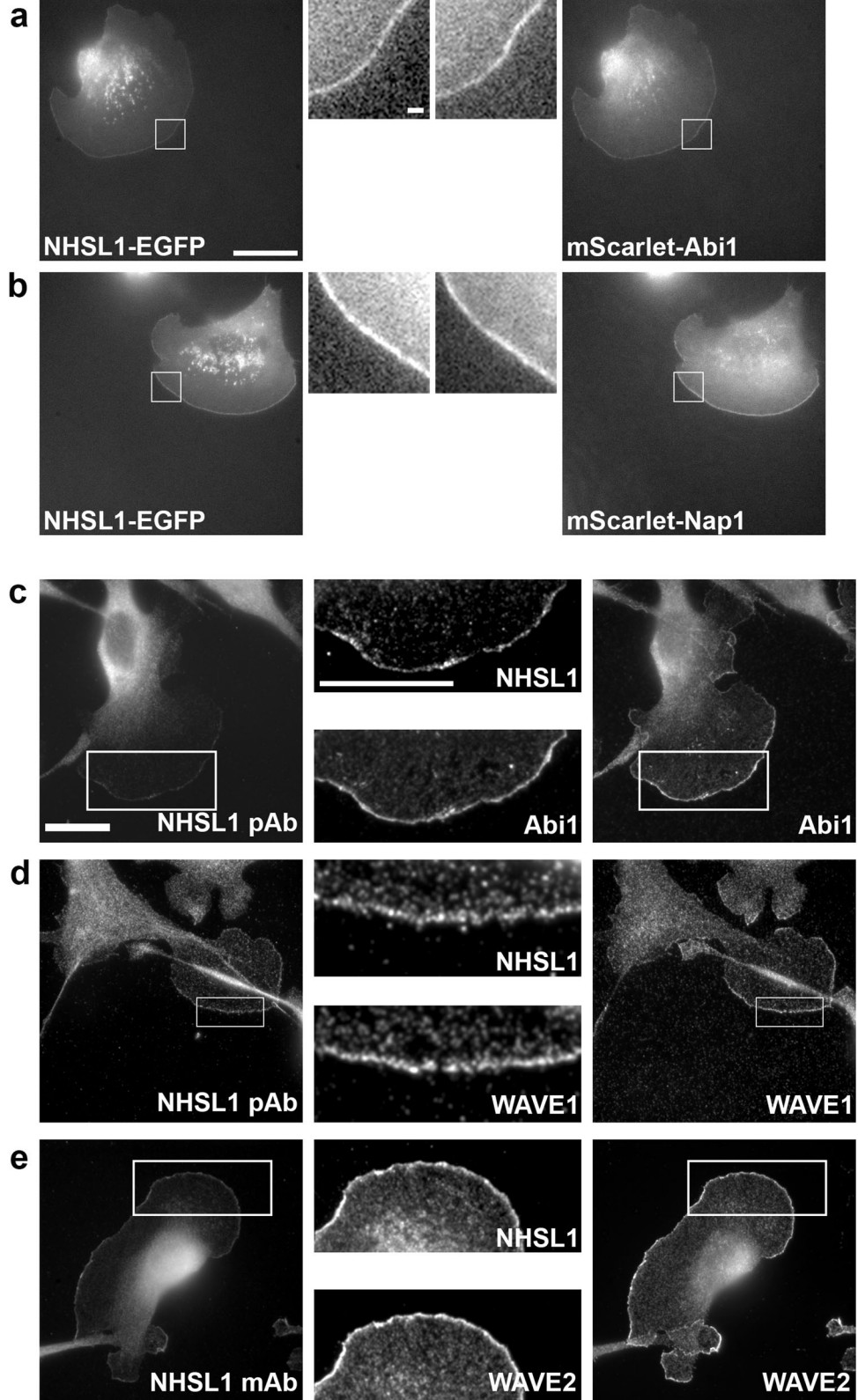

**Fig. 4 NHSL1 co-localises with Abi and the Scar/WAVE complex. a** Still images from live-cell imaging showing NHSL1-EGFP co-expressed with mScarlet-I-tagged Abi1 and **b** Nap1 in B16-F1 cells. Representative images shown from three independent experiments. Scale bar represents 20 μm. Inset is a magnified view of the white box. Scale bar in inset **a** applies to both insets **a**, **b**: 1.25 μm. See also related Supplementary Movie 6. **c–e** Endogenous NHSL1 co-localises with Abi1 (**c** NHSL1 pAb) Scar/WAVE1 (**d** NHSL1 pAb), and Scar/WAVE2 (**e** NHSL1 mAb) at the very edge of lamellipodia in B16-F1 mouse melanoma cells. Scale bar in **c** applies also to **d**, **e**: 20 μm. Inset represents a magnified view of the white box. Scale bar in the inset in **c** applies also to inset for **d**, **e**: 20 μm. Representative images shown from three independent experiments. See Supplementary Fig. 8 for line scans of co-localisations in **c** and **e**.

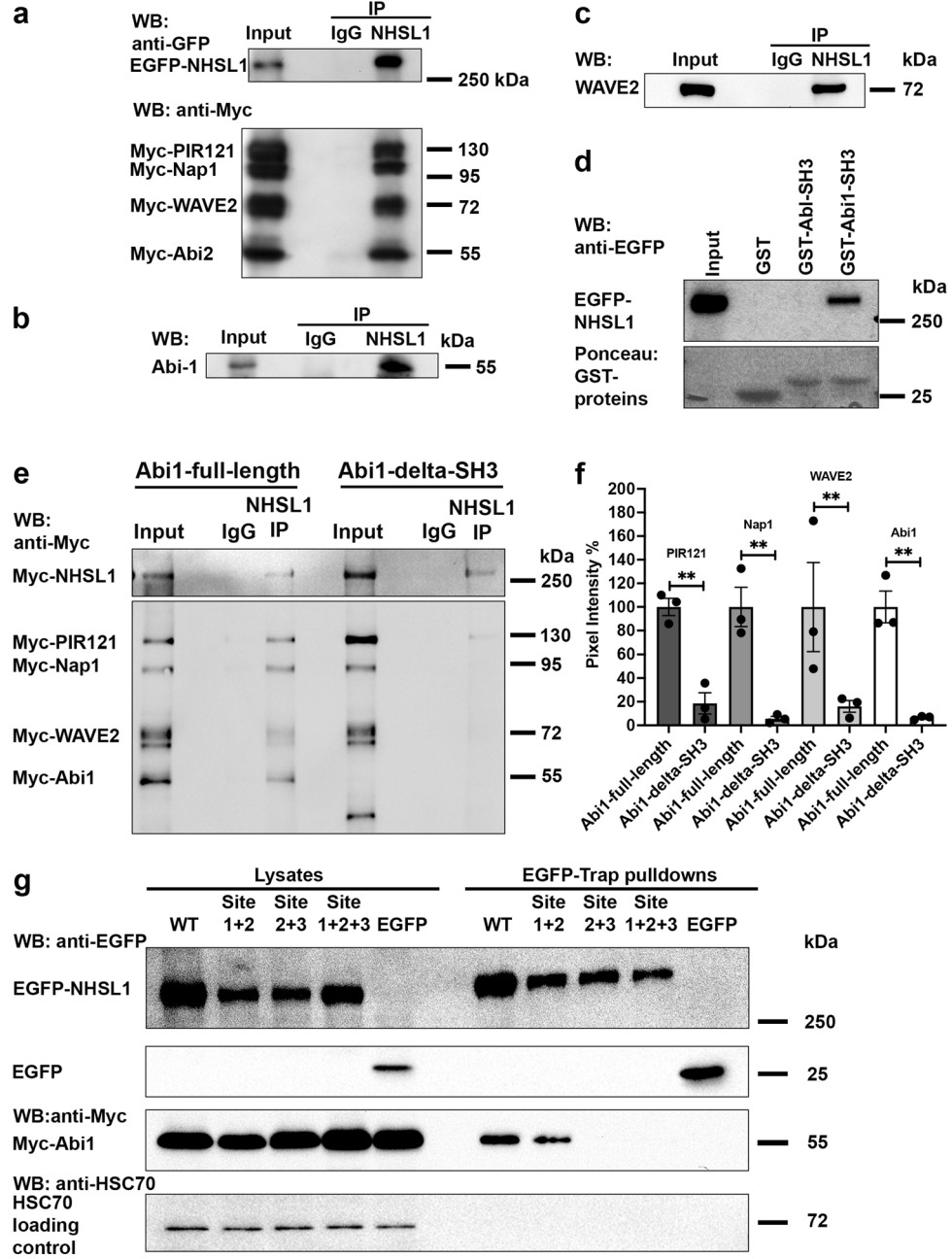

**Fig. 5 NHSL1 interacts with Abi and the Scar/WAVE complex. a** The Scar/WAVE complex co-immunoprecipitates with NHSL1. HEK cells were transfected with EGFP-NHSL1, and Myc-Pir121, -Nap-1, -WAVE2, -Abi2. NHSL1 was immunoprecipitated (pAb 4457) from lysates and co-immunoprecipitation tested on a western blot with Myc and EGFP antibodies. (see Supplementary Fig. 9a, b for full western blots). **b, c** Western blot showing endogenous NHSL1 pulled down with polyclonal NHSL1 (4457) antibody followed by western blotting with **b** Abi1 and **c** Scar/WAVE2 antibodies detecting endogenous co-immunoprecipitation of these proteins in MCF10A (Abi1) and B16-F1 (Scar/WAVE2) cell lysates. Immunoprecipitation using non-immune rabbit IgG served as a negative control. **d** GST-pulldowns using purified Glutathione-sepharose coupled GST-fusion proteins of Abi1 and c-Abl-SH3 domains or GST alone from HEK cell lysates that were transfected with EGFP-NHSL1. Following GST-pulldown EGFP-NHSL1 was detected in a western blot with anti-EGFP antibodies. Ponceau staining of membrane reveals GST or GST-tagged Abi- or Abl-SH3 domains used. **e** Western blot showing immunoprecipitation using NHSL1 polyclonal (4457) antibody or non-immune rabbit IgG control from HEK cell lysates expressing Myc-NHSL1 and all Myc-tagged components of the Scar/WAVE complex, including Myc-Abi1-full-length (**e**, left), Myc- Abi1-delta-SH3 (**e**, right). The western blot shows co-immunoprecipitation between NHSL1 and all components of the Scar/WAVE complex only when the Abi SH3 domain is present (Myc-HSPC300 is not shown). **a–e** Representative blots from three independent biological experiments. **f** Quantification of band intensity from chemiluminescence from **e** imaged with a CCD camera. Co-immunoprecipitation is reduced by >80%. Bars indicate mean ± SEM (error bars); $n = 3$ independent biological experiments; one-way ANOVA: $P = 0.0002$; $F(7,16) = 8.855$; Sidak's multiple comparisons tests: PIR121 **$p = 0.0093$; Nap1 **$P = 0.0028$; WAVE2 **$P = 0.0074$; Abi1 **$P = 0.0030$. **g** HEK cells were transfected with EGFP-tagged NHSL1 or NHSL1-SH3 binding mutants or EGFP only as negative control and Myc-Abi1. After GFP-trap pulldown of wild-type (WT) NHSL1 or different NHSL1-SH3 binding mutants, co-precipitation was detected in a western blot with Myc antibody. Representative blots from five independent biological experiments. Source data are provided as a Source Data file.

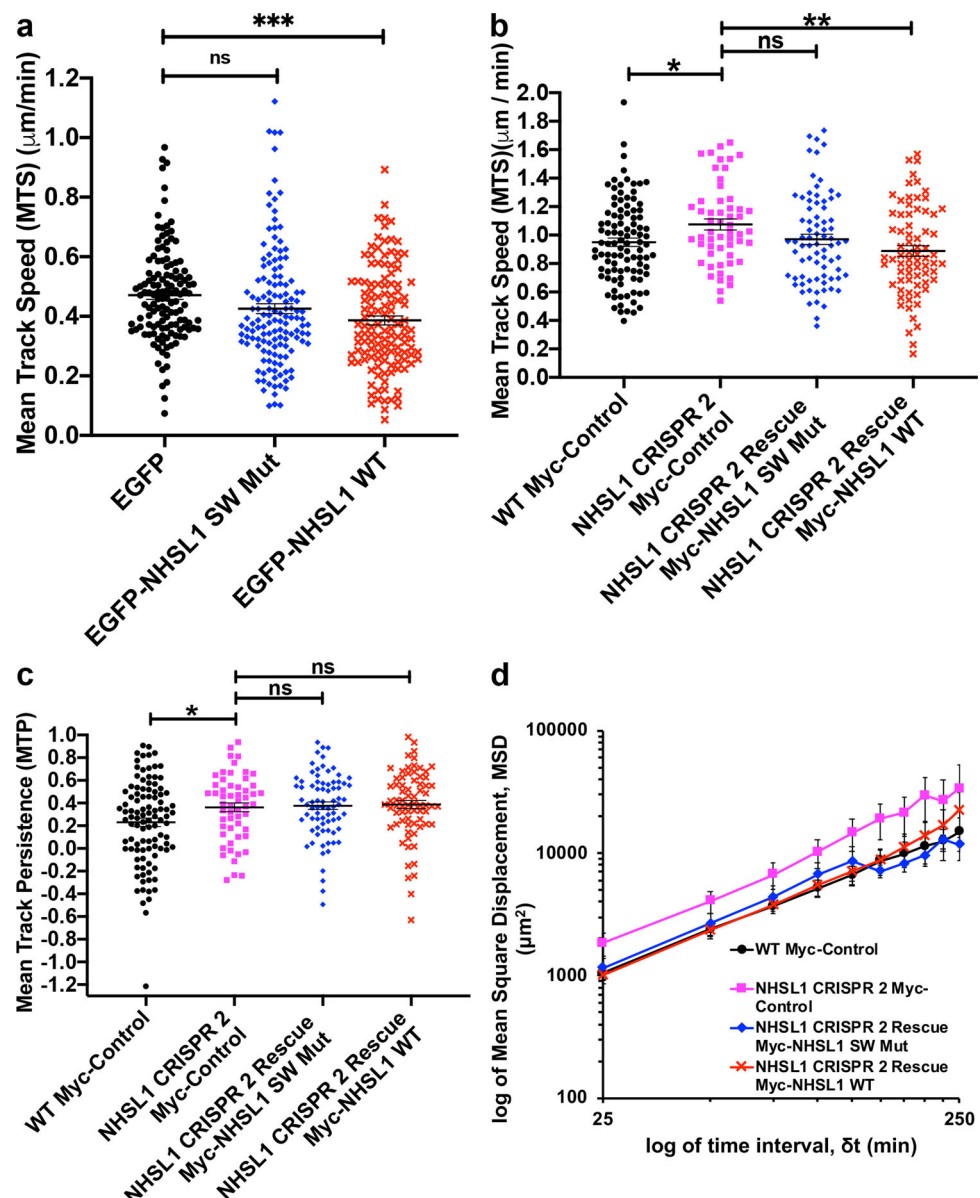

**Fig. 6 NHSL1 negatively regulates cell migration via the Scar/WAVE complex. a** Speed of randomly migrating B16-F1 cells overexpressing either wild-type NHSL1 (NHSL1 WT, black circles) or the NHSL1 mutant in the Scar/WAVE complex binding sites (NHSL1 SW Mut, blue diamonds) or EGFP alone (red crosses) as control plated on fibronectin after selection using a bicistronic puromycin expression plasmid ensuring all cells analysed overexpressed NHSL1. Mean track speed (dt = 3, TR = 4; see 'Methods' for calculation). Results are mean ± SEM (error bars), of four independent biological experiments. Each data point represents the mean speed of a cell from a total number of 106, 104, 108 cells for control, NHSL1 WT and NHSL1 SW Mut, respectively. One-way ANOVA: $P = 0.0007$; $F_{(2,395)} = 7.385$; and Dunnett's multiple comparisons tests: $***P = 0.0003$; $^{ns}P = 0.0655$; **b, c** speed and persistence of randomly migrating wild-type B16-F1 cells expressing Myc alone as control (black circles) or CRISPR 2 cells expressing either the NHSL1 mutant in the Scar/WAVE complex binding sites (NHSL1 SW Mut, blue diamonds) or NHSL1 (NHSL1 WT, red crosses) or Myc alone as control (pink squares) plated on laminin after selection using a bicistronic blasticidin expression plasmid ensuring all cells analysed expressed NHSL1. Mean track speed (**b**) and persistence (**c**) (dt = 5, TR = 2; see 'Methods' for calculation). Each data point represents the mean speed of a cell. Results are mean ± SEM (error bars). **b** One-way ANOVA: $P = 0.0077$; $F_{(3,302)} = 4.044$; and Dunnett's multiple comparisons test: $**P = 0.0018$; $*P = 0.0364$; $^{ns}P = 0.1215$. **c** One-way ANOVA: $P = 0.0034$; $F_{(3,302)} = 4.648$; and Dunnett's multiple comparisons tests: CRISPR 2 vs. WT control: $*P = 0.0397$; NHSL1 CRISPR2 vs. Rescue Myc-NHSL1 SW Mut: $^{ns}P = 0.9877$; NHSL1 CRISPR2 vs. Rescue Myc-NHSL1 WT: $^{ns}P = 0.9442$. **d** Mean Square Displacement analysis (log-log plot) of data shown in **b**, **c**. Results are mean values ± SEM (error bars). **b–d** $n = 102$ (wild-type cells Myc only), 55 (CRISPR 2 cells Myc only), 77 (CRISPR 2 Rescue Myc-NHSL1 SW Mut), 72 (CRISPR 2 Rescue Myc-NHSL1 WT) from five independent biological experiments. Source data are provided as a Source Data file.

with Abi1 (Fig. 5g). Taken together, these data indicate that Abi binds via its SH3 domain to two sites in NHSL1.

**NHSL1 reduces cell migration via the Scar/Wave complex.** We observed that loss of NHSL1 resulted in increased cell migration

speed and persistence (Fig. 2 and Supplementary Figs. 3, 4). To examine the consequences of increasing NHSL1 expression, we overexpressed EGFP-tagged wild-type NHSL1 (EGFP-NHSL1 WT) or the NHSL1 cDNA which cannot interact with the Abi SH3 domain and hence cannot interact with the Scar/WAVE complex (Fig. 5g) (EGFP-NHSL1 SW Mut) in B16-F1 cells

(Supplementary Fig. 11a). We quantified random cell migration behaviour after plating the cells on fibronectin and observed a moderate but significant reduction in cell migration speed (Fig. 6a) and a moderately reduced mean square displacement (Supplementary Fig. 11b) for cells overexpressing wild-type EGFP-NHSL1 compared to EGFP control. This is consistent with the result from the NHSL1 CRISPR cells, which displayed the opposite effect (Fig. 2c–f). Cell migration persistence was increased upon overexpression of NHSL1 (Supplementary Fig. 11c–e). Since CRISPR knockout of NHSL1 also increases persistence, this suggests that optimal expression levels of NHSL1 may finetune cell migration persistence.

The NHSL1 Scar/WAVE binding mutant localises to the very edge of lamellipodia (Supplementary Fig. 11f and Supplementary Movie 7) like wild-type EGFP-NHSL1 (Fig. 1f, g and Supplementary Movie 1). In contrast, the NHSL1 Scar/WAVE binding mutant did not reduce migration speed (Fig. 6a) suggesting that NHSL1 negatively regulates cell migration speed via an interaction with the Scar/WAVE complex. To verify this and to test whether the observed phenotypes in the NHSL1 CRISPR knockout clones were not due to off-target effects, we re-expressed Myc-tagged wild-type NHSL1 (Myc-NHSL1 WT) or the NHSL1 Scar/WAVE complex binding mutant (Myc-NHSL1 SW Mut) in B16-F1 cells. After plating the cells on laminin we again observed an increase in random cell migration speed and persistence between wild-type control and NHSL1 CRISPR2 knockout cells (Fig. 6b, c) confirming our previous results (Fig. 2c–e). Re-expression of neither wild-type Myc-NHSL1 nor Myc-NHSL1 SW Mut in CRISPR 2 cells rescued cell migration persistence (Fig. 6c, d and Supplementary Fig. 12a, b). This result and the increase in cell migration persistence upon NHSL1 overexpression (Supplementary Fig. 11c–e) and NHSL1 knockout (Fig. 2e) together suggest that optimal expression levels of NHSL1 may be required for the lower cell migration persistence observed in the wild-type cells. However, re-expression of wild-type Myc-NHSL1 but not Myc-NHSL1 SW Mut in CRISPR 2 cells resulted in a significant reduction in cell migration speed (Fig. 6b). This indicates that wild-type NHSL1 but not NHSL1 SW Mut (that cannot bind to the Scar/WAVE complex) can rescue the NHSL1 knockout phenotype and suggests that NHSL1 negatively regulates cell migration speed via an interaction with the Scar/WAVE complex.

**NHSL1 CRISPR knockout increases Arp2/3 activity in cells.** As the Scar/WAVE complex is the major activator of the Arp2/3 complex, which induces branched F-actin nucleation at the leading edge of cells, we sought a direct method to investigate whether NHSL1 controls Arp2/3 activity. In order to directly quantify Arp2/3 activity in cells, we developed a FRET-FLIM Arp2/3 biosensor. We designed this biosensor based on in vitro FRET probes for the Arp2/3 complex[22,23], which were successfully used to measure the activity of the purified complex and relies on the conformational change of the Arp2/3 complex upon binding to nucleation promotion factors such as the Scar/WAVE complex[23–29]. Here we tagged the Arp2/3 subunits ARPC3 with mTurq2 as the FRET donor and ARPC1B with mVenus as the FRET acceptor and expressed them from a bicistronic construct (ARPC1B-mVenus-P2A-ARPC3-mTurq2). When the Arp2/3 complex is activated by Scar/WAVE, ARP2 and ARP3 and also ARPC3 and ARPC1B move closer to each other[22–29] allowing FRET between mTurq2 and mVenus to occur. To evaluate whether this biosensor faithfully reports changes in Arp2/3 activity in cells, we expressed our Arp2/3 biosensor together with Myc-DA-Rac1 or empty Myc vector as control and measured FRET efficiency by Fluorescence Lifetime Imaging (FLIM). This analysis revealed that FRET efficiency and hence Arp2/3 activity was significantly increased upon expression of dominant active Rac compared to

control (Fig. 7a–c and Supplementary Fig. 13a–e). We then expressed our biosensor in control B16-F1 cells or in the NHSL1 CRISPR 2 and 21 cell lines and again measured FRET by FLIM. We found that FRET efficiency and hence Arp2/3 activity was significantly increased in both NHSL1 CRISPR 2 and 21 cell lines compared to control B16-F1 cells (Fig. 7d–f and Supplementary Fig. 14a–f).

To test whether the observed phenotypes in the NHSL1 CRISPR knockout clones were not due to off-target effects, we re-expressed Myc-tagged wild-type NHSL1 (Myc-NHSL1 WT) or the NHSL1 Scar/WAVE complex binding mutant (Myc-NHSL1 SW Mut) in the NHSL1 CRISPR 2 cell line and also expressed our Arp2/3 biosensor and quantified FRET by FLIM. Again, we observed a significant increase in Arp2/3 activity in the NHSL1 CRISPR 2 cell line compared to control B16-F1 cells, which was rescued with the wild-type NHSL1. This confirmed that indeed the observed increase in Arp2/3 activity in the NHSL1 CRISPR 2 cells was due to loss of NHSL1 and not due to off-target effects (Supplementary Fig. 14g). We also found a significant rescue with the NHSL1 Scar/WAVE complex binding mutant (Supplementary Fig. 14g) suggesting that in whole cells NHSL1 affects Arp2/3 activity through other mechanisms in addition to its interaction with the Scar/WAVE complex.

As an additional control, we performed immunofluorescence analysis on wild-type B16-F1 cells or the NHSL1 CRISPR 2 and 21 cells and quantified whole-cell Arp2/3 intensity. We found that total Arp2/3 levels were not increased but rather reduced in NHSL1 CRISPR cells (Supplementary Fig. 15a, c) and this excludes the possibility that the observed increase in Arp2/3 activity is due to increased Arp2/3 levels when NHSL1 is knocked out. Similarly, western blot analysis revealed that both total cellular Arp2/3 complex (subunit ARPC2) and Scar/WAVE2 levels were reduced in the NHSL1 CRISPR 2 line (Supplementary Fig. 15d–f), suggesting that persistent overactivation of the Scar/WAVE and Arp2/3 complexes in the absence of NHSL1 may lead to their proteasomal degradation[30,31]. Consistent with a reduction in total Arp2/3 levels, we observed that the total cellular F-actin content was significantly reduced in NHSL1 CRISPR 2 cells compared to wild-type cells. This phenotype was rescued by re-expressing wild-type Myc-NHSL1 but not Myc-NHSL1 SW Mut in the NHSL1 CRISPR 2 cells (Fig. 8c and Supplementary Fig. 16a) suggesting that NHSL1 controls cellular F-actin levels by binding to the Scar/WAVE complex. Similarly, we observed a significant reduction in total cellular F-actin content upon knockdown of NHSL1 and, conversely, an increased total cellular F-actin content upon overexpression of EGFP-NHSL1 in HEK293 cells (Supplementary Fig. 16b, c).

**Loss of NHSL1 increases lamellipodial F-actin content.** To gain a deeper insight on how NHSL1 affects cell motility, we quantified lamellipodium parameters. We transfected our stable NHSL1 CRISPR B16-F1 cell lines or control wild-type B16-F1 cells with LifeAct-EGFP. We found that NHSL1 CRISPR 2 and 21 cells had a larger cell area (Supplementary Fig. 17a) and thus we normalised the lamellipodia length to the total cell area. This analysis revealed that the NHSL1 CRISPR 2 and 21 cells have a significantly reduced ratio of lamellipodia length to cell area (Fig. 8a, b and Supplementary Fig. 17a, b). However, both lamellipodia width and the number of microspikes per length of lamellipodium were unchanged (Fig. 8a and Supplementary Fig. 17c, d).

To quantify Arp2/3 activity in lamellipodia we sought to estimate Arp2/3 biosensor FRET efficiency at the leading edge by manually outlining lamellipodia. This approach is limited by the resolution of the FRET-FLIM microscope. Nevertheless, we observed a change in FRET efficiency from 0.01 to 3.97% for

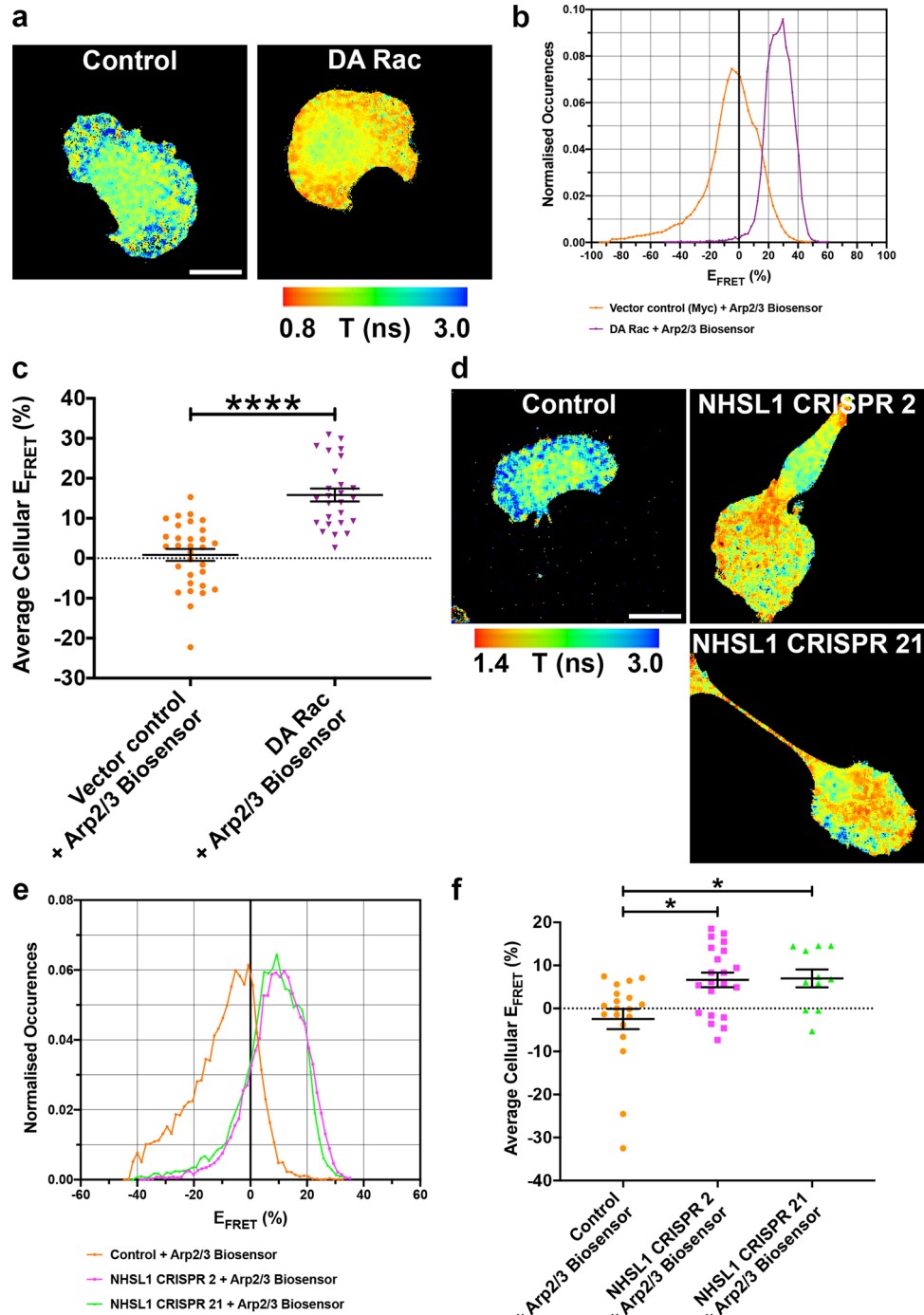

Control vs. CRISPR 2 ($P = 0.6914$) and 0.01 to 8.87% for Control vs. CRISPR 21 ($P = 0.2502$) for Arp2/3 activity in the lamellipodium. Neither change is significant (Supplementary Fig. 14h).

As an alternative method, endogenous Arp2/3 staining intensity in the lamellipodium can be used as an approximation of Arp2/3 activity since Arp2/3, after activation at the leading edge, is incorporated into the lamellipodial F-actin meshwork. We found that relative Arp2/3 complex (ARPC2) intensity was significantly increased in the lamellipodium in the NHSL1 CRISPR 2 cells (the mean changed from 0.9789 to 1.149). The mean of the NHSL1 CRISPR 21 cells also changed from 0.9789 for the control wild-type cells to 1.104 but the latter change was not significant (Fig. 8d and Supplementary Fig. 15a). The

different results from these two CRISPR KO clones are consistent with our finding that the NHSL1 CRISPR 21 cell line represents a partial knockout whereas the NHSL1 CRISPR 2 cell line represents a full knockout (Supplementary Fig. 2). This suggests that NHSL1 functions to reduce Arp2/3 activity in the lamellipodium. When we quantified Alexa488-phalloidin labelled F-actin intensity in the lamellipodium, we observed an increase in intensity per area in the NHSL1 CRISPR cells (Fig. 8e and Supplementary Fig. 15b). Conversely, we found a significant reduction of lamellipodia LifeAct-EGFP intensity per area upon wild-type NHSL1 overexpression but not with the NHSL1 Scar/WAVE binding mutant (Fig. 8f and Supplementary Fig. 17e) suggesting that NHSL1 may inhibit F-actin nucleation in the lamellipodium by binding to Scar/WAVE.

**Fig. 7 NHSL1 negatively regulates Arp2/3 activity. a** Lifetime images of B16-F1 expressing the Arp2/3 biosensor and either empty Myc- control plasmid or Myc-tagged dominant active Rac (DA Rac). Warm colours indicate short lifetimes of the donor mTurq2, which represent high FRET efficiency and Arp2/3 activity. Representative images from 4 independent biological experiments are shown. Scale bar: 20 μm. **b** FRET efficiency histograms from the same representative wild-type B16-F1 (orange circles) and DA Rac (lilac upside-down triangles) cells expressing the Arp2/3 biosensor as shown in **a**. **c** Quantification of average cellular FRET efficiency that represents Arp2/3 activity from 4 independent biological repeats from control: 31 cells (orange circles); DA Rac: 26 cells (lilac upside-down triangles). Data points are the weighted average means for each cell, calculated from the FRET efficiency histogram and used instead of the normal (unweighted) mean in order to better represent the true FRET efficiency. Results are the average weighted average mean ± SEM (error bars), ****$P = 0.0000000065$; $t = 7.379$; df $= 54$. Unpaired, two-tailed $t$-test. **d** Lifetime images of wild-type B16-F1, NHSL1 CRISPR clone 2 and NHSL1 CRISPR clone 21 cells expressing the Arp2/3 biosensor. Warm colours indicate short lifetimes of the donor mTurq2, which represent high FRET efficiency and Arp2/3 activity. Representative images from 4 independent biological experiments are shown. Scale bar: 20 μm. **e** FRET efficiency histogram from the same representative wild-type B16-F1 (orange circles), NHSL1 CRISPR clone 2 (magenta squares) and NHSL1 CRISPR clone 21 (green triangles) cells expressing the Arp2/3 biosensor as shown in **d**. **f** Quantification of average cellular FRET efficiency which represents Arp2/3 activity from 4 independent biological repeats from control: 19 cells (orange circles); NHSL1 CRISPR 2: 21 cells (magenta squares); NHSL1 CRISPR 21: 11 cells (green triangles). Data points are the weighted average means for each cell, calculated from the FRET efficiency histograms. Data points represent the average weighted mean ± SEM (error bars), Kruskal–Wallis test: $P = 0.0091$; Kruskal–Wallis statistic: 9.407 and Dunn's multiple comparisons test: control versus CRISPR 2 *$P = 0.0136$; control versus CRISPR 21 *$P = 0.0260$. Source data are provided as a Source Data file.

**NHSL1 knockout decreases lamellipodial F-actin assembly rate.** To explore the effects of NHSL1 on F-actin retrograde flow in the lamellipodium we again used our NHSL1 CRISPR 2 cell line and rescued it with Myc-NHSL1 WT or the Myc-NHSL1 SW Mut and transfected them with LifeAct-EGFP (Fig. 9a and Supplementary Movie 8). The protrusion speed of lamellipodia was not significantly changed by NHSL1 knockout (Supplementary Fig. 18a). However, particle image velocimetry analysis revealed a significant reduction of F-actin retrograde flow in the NHSL1 CRISPR 2 lamellipodia compared to wild-type control lamellipodia (Fig. 9a, b). Re-expression of Myc-NHSL1 WT in the NHSL1 CRISPR 2 cells changed the mean values of the retrograde flow speed in lamellipodia from 1.34 to 1.43 μm/min and re-expression of Myc-NHSL1 SW Mut changed the mean values of the flow speed from 1.34 to 1.51 μm/min but these changes were not significant (Fig. 9a, b).

A more intuitive measure of effective actin polymerisation at the lamellipodium is the F-actin assembly rate which takes the forward protrusion of the plasma membrane and the retrograde actin flow into account and can be calculated in this case as the sum of the magnitude of the vectors of the flow speed and the protrusion speed[32]. Even though we measured increased Arp2/3 activity in the NHSL1 CRISPR 2 cells (Fig. 7d–f), we surprisingly observed a significant reduction of F-actin assembly rate in NHSL1 CRISPR 2 lamellipodia compared to wild-type control lamellipodia (Fig. 9a, c). Re-expression of Myc-NHSL1 WT or the Myc-NHSL1 SW Mut changed the mean values of the F-actin assembly rate for both from 4.04 to 4.30 μm/min compared to the NHSL1 CRISPR knockout lamellipodia but this change was not significant (Fig. 9a, c). This seemingly counterintuitive observation that NHSL1 CRISPR knockout causes increased Arp2/3 activity yet decreases F-actin retrograde flow and assembly rate in the lamellipodium may be explained by the effects of higher filament densities on membrane tension and the tension gradient across the lamellipodium[32,33]. Our observations are consistent with results from modelling of actin filament density in the lamellipodium and calculating resulting F-actin retrograde flow, protrusion speed and assembly rate[32,33] (see 'Discussion' section).

**NHSL1 reduces stability of lamellipodia protrusion.** Since NHSL1 reduced Arp2/3 activity (Fig. 7), we further explored whether NHSL1 affects lamellipodia dynamics and/or lamellipodia stability. We quantified leading-edge morphodynamics (Supplementary Movies 9 and 10). This revealed that protrusion speed did not change significantly in the NHSL1 CRISPR cells compared to control (Fig. 10e) in agreement with lamellipodia protrusion speed measurements during flow analysis

(Supplementary Fig. 18a). In addition, we did not observe a significant change in lamellipodia protrusion speed when we re-expressed wild-type or the Scar/WAVE binding mutant of NHSL1 in the NHSL1 CRISPR 2 knockout cell line (Fig. 10e).

Lamellipodia stability can be defined both temporally, as the duration of lamellipodium protrusion until the next retraction, as well as spatially: lamellipodia can be characterised by protruding segments interrupted by small retractions. Stable lamellipodia with longer sections of protrusions may more likely increase cell migration speed and persistence than lamellipodia with many retractions. We thus analysed lamellipodia stability and observed that the NHSL1 CRISPR 2 cells displayed a significantly increased uninterrupted lamellipodium while this parameter was not changed in the NHSL1 CRISPR 21 cell line (Fig. 10a, c and Supplementary Movie 9). The different results from these two CRISPR KO clones are consistent with our finding that the NHSL1 CRISPR 21 cell line represents a partial knockout whereas the NHSL1 CRISPR 2 cell line represents a full knockout (Supplementary Fig. 2). Conversely, cells overexpressing Myc-NHSL1 (Fig. 10b, d and Supplementary Movie 10) showed a significantly decreased uninterrupted lamellipodium compared to controls. When we compared the standard deviation (S.D.) of the lamellipodia protrusion speed from all frames of a movie as a readout for the temporal stability of protrusions over time, we observed a change from S.D. $= 0.376$ μm/min for WT Myc-Control to S.D. $= 0.265$ μm/min for NHSL1 CRISPR 2 knockout cells. This standard deviation of the lamellipodia protrusion speed of the NHSL1 CRISPR 2 knockout cells was significantly increased in the NHSL1 CRISPR 2 cells re-expressing wild type but not the NHSL1 Scar/WAVE complex binding mutant (Fig. 10f). Taken together, this indicates that NHSL1 reduces the stability of lamellipodia protrusions potentially via its interaction with the Scar/WAVE complex.

## Discussion

The Scar/WAVE complex, which is activated by several signals including Rac, is essential for lamellipodia formation and promotes mesenchymal cell migration[2]. We postulate that the Scar/WAVE complex needs also to be negatively regulated at the leading edge to tightly control lamellipodial protrusion dynamics and cell steering. Yet, so far, no direct inhibitor of the Scar/WAVE complex has been described. Here we reveal that NHSL1 is a direct, negative regulator of the Scar/WAVE complex, which consequently reduces Arp2/3 activity. This reduction in Scar/WAVE-Arp2/3 activity may cause a decrease in lamellipodia stability, and thus also cell migration efficiency.

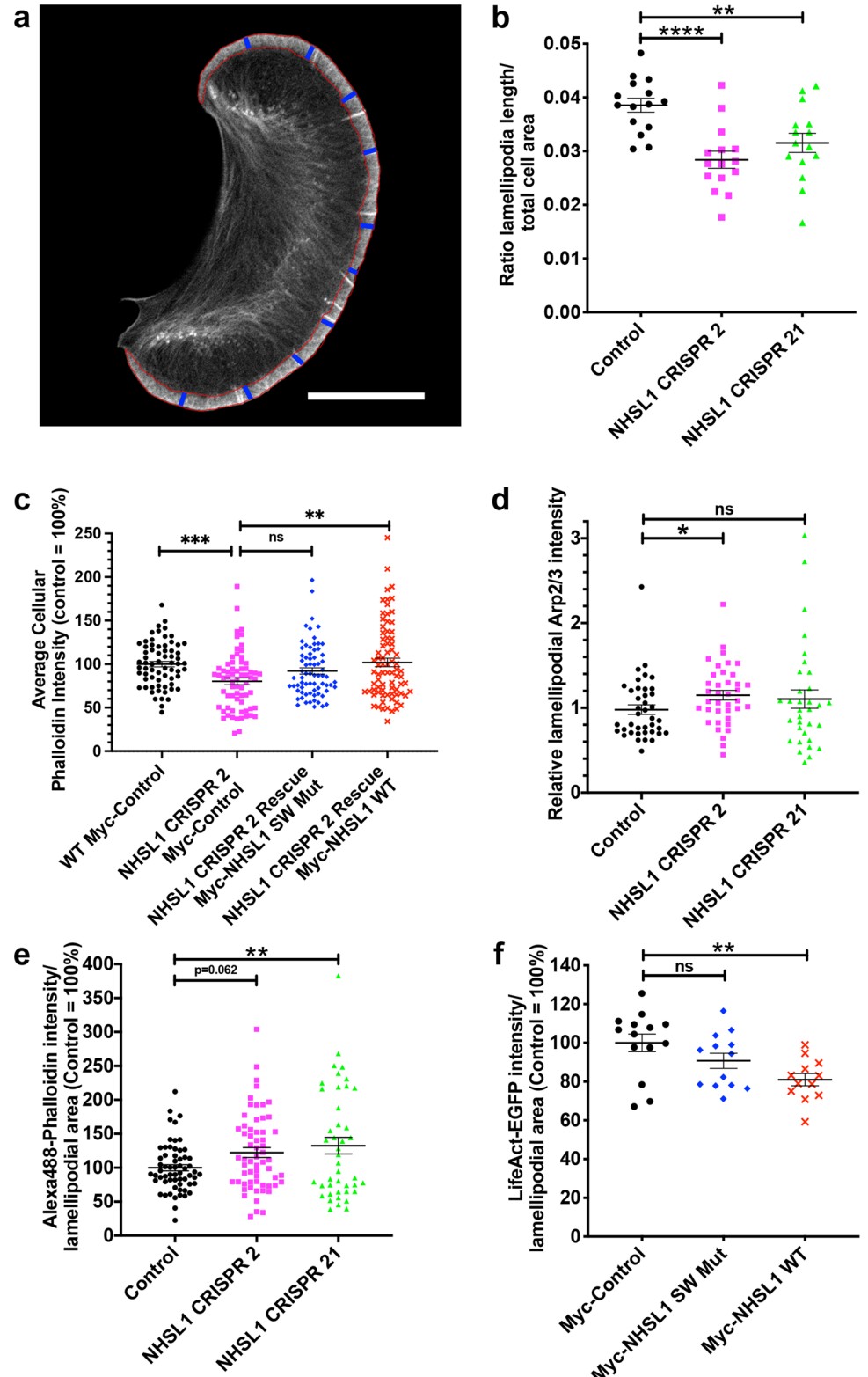

NHSL1 knockout cells also displayed reduced retrograde F-actin flow speed which coincided with a reduced F-actin assembly rate (Fig. 9). Actin polymerisation at the very edge of lamellipodia creates the force for plasma membrane protrusion. How the assembly rate is divided between the protrusion and retrograde flow controls the speed of protrusions. Dolati et al. (2018) modelled the relationship between actin filament density at the leading edge of cells and the resulting retrograde flow, lamellipodia protrusion and F-actin assembly rates[32]. The model, which agrees best with experimental results, revealed that protrusion and assembly rates increase with actin filament density at the leading edge up to a certain threshold. This threshold corresponds to the density measured for wild-type B16-F1 lamellipodia. When actin filament density is further increased the protrusion rate then plateaus and the actin assembly rate again decreases[32]. This is most likely due to membrane tension which counteracts the

**Fig. 8 NHSL1 negatively regulates Arp2/3 and F-actin content in the lamellipodium. a** First frame of a movie of LifeAct-EGFP expressing B16-F1 cells showing how length, width and area was quantified (see 'Methods' section) in (**b**; S17a–d): red line: area; blue lines: width of lamellipodium. Bar: 20 μm. **b** Ratio of lamellipodia length/total cell area. $n = 15$ cells each. One-way ANOVA, Dunnett's test; ****$P < 0.0001$; **$P = 0.0059$; $F(2,42) = 10.84$. **c** Total cellular F-actin (Alexa568-phalloidin) intensity/area ratio in wild-type B16-F1 cells expressing Myc-tag (WT Myc-Control, black circles) or CRISPR 2 cells expressing the NHSL1 mutant in the Scar/WAVE complex binding sites (Myc-NHSL1 SW Mut, blue diamonds) or NHSL1 (Myc-NHSL1 WT, red crosses) or Myc-tag as control (pink squares) plated on laminin. One-way ANOVA: $P = 0.0009$; Kruskal–Wallis statistics 16.44; ***$P = 0.0003$; **$P = 0.0083$; $^{ns}P = 0.1518$; outliers removed: ROUT method ($Q = 1\%$) ($n$ analysed/$n$ outliers $= n_o$ removed); WT control: $n = 67$ cells/$n_o = 0$; NHSL1 CRISPR2: $n = 71$ cells/$n_o = 3$; NHSL1 CRISPR2 + Myc-NHSL1 SW Mut Rescue: $n = 76$ cells/$n_o = 5$; NHSL1 CRISPR2 + Rescue Myc-NHSL1 WT: $n = 82$ cells/$n_o = 4$. **d** Relative lamellipodial Arp2/3 intensity: Arp2/3 intensity in the lamellipodium normalised cell-by-cell against whole-cell Arp2/3 intensity. Kruskal–Wallis test ($P = 0.0514$), Dunn's test: *$P = 0.0353$; $^{ns}P > 0.9999$; wild type ($n = 40$ cells), NHSL1 CRISPR 2 ($n = 38$ cells) and NHSL1 CRISPR 21 ($n = 33$ cells) B16-F1 cells on laminin, stained with anti-ARPC2 antibodies. **e** Lamellipodial F-actin (Alexa488-phalloidin) intensity/area ratio in B16-F1 wild-type control cells or NHSL1 CRISPR 2 and NHSL1 CRISPR 21 B16-F1 cells plated on laminin. $n =$ control (61), CRISPR 2 (61), CRISPR 21 (43) cells; one-way ANOVA: $P = 0.0127$; $F(2,162) = 4.486$; Dunnett's multiple comparisons test: **$P = 0.0099$. **f** Lamellipodial F-actin (LifeAct-EGFP) intensity/area ratio in B16-F1 cells transfected with a tri-cistronic plasmid expressing Myc-tagged NHSL1 (wild type or Scar/WAVE binding mutant), puromycin resistance and LifeAct-EGFP or empty plasmid (Myc-IRES-Puro-T2A-LifeAct-EGFP) after selection with puromycin plated on laminin. $n =$ control (14), WT (12), NHSL1-SW mut (13) cells. One-way ANOVA: $P = 0.0074$; $F(2,36) = 5.636$; Dunnett's multiple comparisons test: **$P = 0.0036$; $^{ns}P = 0.1810$. **b–f** Results are mean ± SEM (error bars) from three (**b**, **d**) or four (**c**, **f**) or five (**e**) biologically independent experiments. Source data are provided as a Source Data file.

---

protrusive force[32,33]. In NHSL1 knockout cells Arp2/3 activity is increased (Fig. 7) and in the lamellipodium, Arp2/3 and actin filament densities (Fig. 8) are higher compared to wild-type B16-F1 cells. Thus, our observed experimental values of reduced retrograde flow coinciding with reduced F-actin assembly rates are consistent with the predicted rates from the model[32,34]. We did not observe an increase in protrusion speed at higher Arp2/3 activity. This may seem counterintuitive but agrees with the predictions of the model. Consequently, the faster cell migration speed observed in the NHSL1 knockout cells cannot be explained by a faster lamellipodial protrusion speed. The increase in cell migration speed may therefore be due to an increase in lamellipodial stability. Taken together, the observed reduction in retrograde flow and assembly rates fit very well with the increased Arp2/3 activity in the NHSL1 CRISPR cells. The resulting increased actin filament branching density in the NHSL1 CRISPR cells may exert more force to counteract the membrane tension and thereby increase lamellipodial stability and consequently increase cell migration efficiency.

We found that NHSL1 knockout cells display a reduction in relative overall lamellipodia length (Fig. 8) while the longest uninterrupted lamellipodia length increases (Fig. 10). How the overall length of lamellipodia is controlled and why some cell types such as fibroblasts have smaller lamellipodia than others, for example, B16-F1 cells or fish keratocytes is unknown. The Scar/WAVE complex appears to form higher-order linear oligomers[35] but how a lateral extension of such oligomers is controlled is not known. Our finding that NHSL1 interacts with the Scar/WAVE complex and affects lamellipodia length and continuity opens the possibility that NHSL1 may be involved in this control of lateral lamellipodia propagation.

Several isoforms of NHSL1 have been predicted from the human genome sequence, which results from alternative splicing of several different exon 1 sequences (Ensembl[13,14,16]). Only one out of several alternative exon 1 sequences includes a Scar homology domain (SHD) also found in Scar/WAVE family proteins which in Scar/WAVE proteins mediates the interaction with Abi, HSPC300, Nap1 and Sra1[2]. The NHSL1 isoform that we cloned harbours an exon 1 which does not include the SHD. A similar exon-intron structure has been described for the related Nance–Horan Syndrome (NHS) and NHSL2 proteins[13,14,36] but overall conservation between family members is low at the amino acid level (NHSL1 versus NHS: 30.3% and NHSL1 versus NHSL2: 25.1% see Supplementary Fig. 18b). The function of NHS in cell migration is not known but NHS has been reported to localise to lamellipodia in MTLn3 breast cancer cells[14] which is consistent with our observation that NHSL1 localises to the edge of protruding lamellipodia (Fig. 1).

The SHD domain of NHS has been shown to co-immunoprecipitate with the Scar/WAVE complex components Abi, HSPC300, Nap1 and Sra1[14]. Here we show that an NHSL1 isoform without the SHD domain interacts with the entire Scar/WAVE complex via the SH3 domain of Abi. Interestingly, the two Abi binding sites which mediate binding to NHSL1 (site 2 + 3; Fig. 5g) are highly conserved in the NHS family (PFAM PF15273 HHM logo, (Supplementary Fig. 18c, d)) even though overall conservation is low (Supplementary Fig. 18b)[13,36] highlighting the importance of Scar/WAVE complex interaction at these particular sites. Furthermore, this suggests that all isoforms of NHS may also directly interact with the Scar/WAVE complex via the Abi SH3 domain-mediated by the two Abi binding sites identified here. We thus postulate that at least some functions such as regulation of cell migration may be conserved between the family members. Therefore, deregulated cell migration may play a role in the pathogenesis of Nance–Horan syndrome.

Overexpression of an isoform of NHS harbouring the SHD domain has been reported to inhibit the ability of MTLn3 breast cancer cells to respond to EGF by forming lamellipodia, whilst overexpression of an isoform of NHS without the SHD domain did not affect lamellipodia extension[14]. In agreement, we show that overexpression of the NHSL1 isoform without the SHD domain does not inhibit lamellipodia extension (Supplementary Movie 1). Furthermore, we found that knockout of all isoforms of NHSL1 increases lamellipodia stability, cell migration speed and persistence (Figs. 2, 6 and 10). However, only lamellipodia stability and cell migration speed but not persistence were rescued in the NHSL1 CRISPR knockout cells when we re-expressed wild-type NHSL1. Since overexpression of NHSL1 also increased cell migration persistence, optimal expression levels of NHSL1 may be required for low cell migration persistence.

The zebrafish orthologue of NHSL1 had been identified in a genetic screen for defects in a specific developmental migration of facial branchiomotor neurons suggesting a critical role of NHSL1 in neuronal migration[37]. Here, we found that NHSL1 negatively regulates random 2D cell migration by reducing lamellipodia stability (Figs. 2, 6 and 10). We hypothesise that this reflects a role in the guidance of NHSL1 during developmental migration as cells must be able to change their path. In addition, the neurons need to integrate additional inputs in this mode of migration since zebrafish NHSL1 functions together with the planar cell polarity components Scrib and Vangl cell autonomously in facial branchiomotor neurons, which involves the interactions between the neurons and their planar-polarised environment of the surrounding tissue[37].

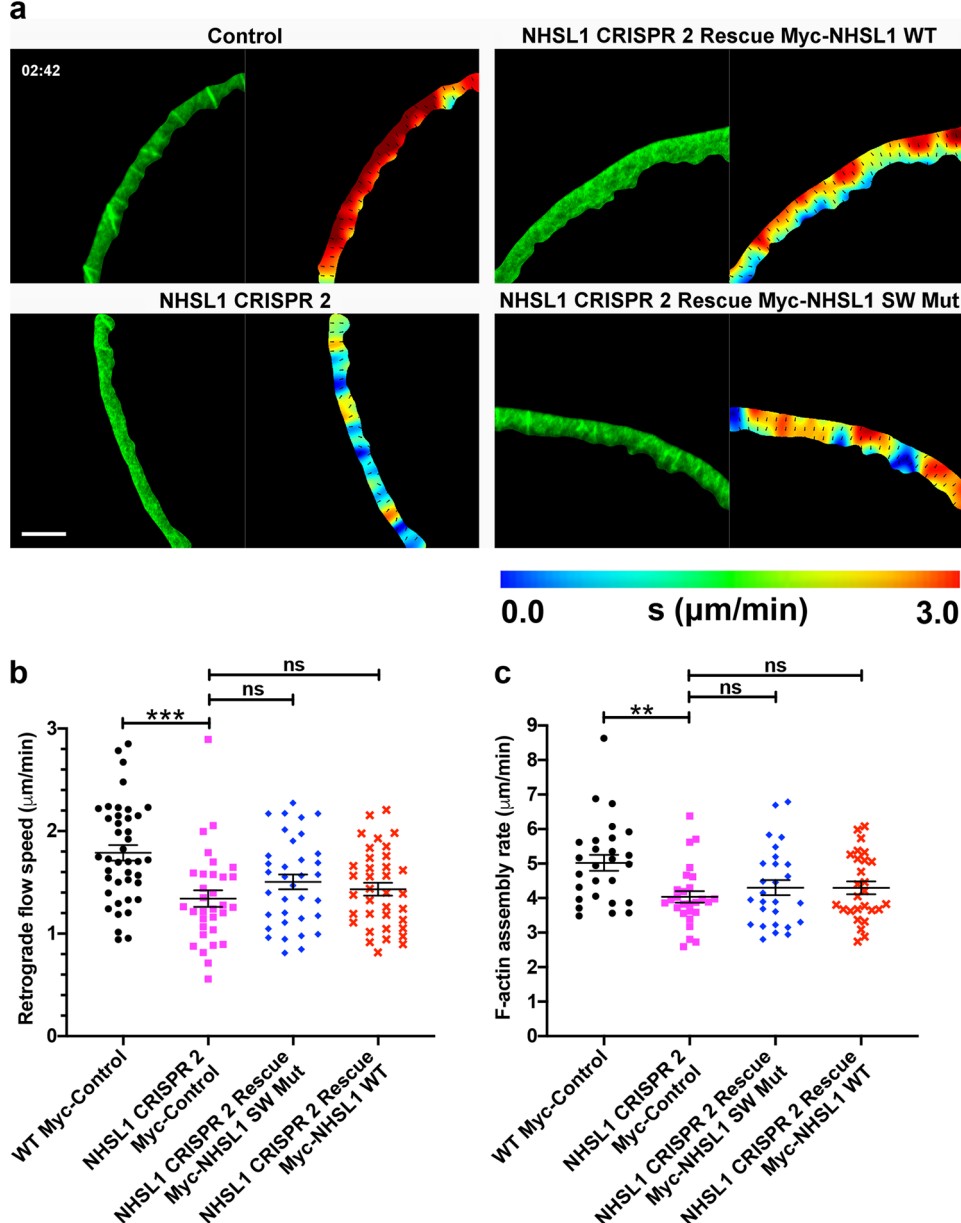

**Fig. 9 NHSL1 increases actin retrograde flow speed and F-actin assembly rate. a** A representative snapshot in time (02:42 min:s) of a segmented lamellipodium from a wild-type cell (WT), NHSL1 knockout cell (NHSL1 CRISPR 2) and NHSL1 knockout cells rescued with a wild-type NHSL1 construct (NHSL1 CRISPR 2 Rescue Myc-NHSL1 WT) and with a mutant NHSL1 construct (NHSL1 CRISPR 2 Rescue Myc-NHSL1 SW Mut) using a bicistronic expression plasmid also conferring resistance to blasticidin to ensure that all cells analysed expressed NHSL1 and also expressing lifeAct-EGFP. Left panels show a frame from segmented Airyscan movies; right panels show the corresponding frame from the PIV interpolated movie. Each colourmap has the same range (0–3 µm/min) and distance between vector arrows (1 µm); warmer colours represent faster flow, vector arrows are all of the unit length. Scale bar: 10 µm. Representative images shown from six independent biological experiments. **b** Quantification of the average speed of actin retrograde flow for each condition over all frames for each cell. Flow is significantly reduced in NHSL1 knockout cells but cannot be fully rescued by either the WT or mutant NHSL1 constructs. Results are mean ± SEM; $n = 41, 32, 35$ and 36 for the respective conditions (left to right on x axis), from 6 independent biological repeats. One-way ANOVA: $P = 0.0001$; $F(3,140) = 7.312$; Tukey's multiple comparisons tests: ***$P = 0.0002$, ns = not significant: CRISPR 2 versus rescue Myc-NHSL1 SW Mut: [ns]$P = 0.4285$; CRISPR 2 versus rescue Myc-NHSL1 WT: [ns]$P = 0.8275$. **c** Quantification of the F-actin assembly rate for each condition over all frames for each cell. The F-actin assembly rate is significantly reduced in NHSL1 knockout cells but cannot be rescued by either the WT or mutant NHSL1 constructs. Results are mean ± SEM; $n = 41, 32, 35$ and 36 for the respective conditions (left to right on x axis), from 6 independent biological repeats. One-way ANOVA: $P = 0.0059$; $F(3,104) = 4.400$; Tukey's multiple comparisons test: **$P = 0.0044$, ns = not significant: CRISPR 2 versus rescue Myc-NHSL1 SW Mut: [ns]$P = 0.7874$; CRISPR 2 versus rescue Myc-NHSL1 WT: [ns]$P = 0.7967$. Source data are provided as a Source Data file.

Active Rac binds to NHSL1 via two sites and thus Rac may recruit NHSL1 to the leading edge of cells (Fig. 3 and Supplementary Figs. 6, 7). Whether this interaction is direct or not needs to be investigated in future studies. We show here that NHSL1 is a direct interactor of the Scar/WAVE complex and co-localises with it at the very edge of lamellipodia. In addition, an EGFP-NHSL1 Scar/WAVE binding mutant localises at lamellipodia like wild-type NHSL1 (Supplementary Fig. 11f). Finally, NHSL1 knockout cells display increased Arp2/3 activity (Fig. 7). Taken together this suggests that NHSL1 may act in a Rac-NHSL1-Scar/

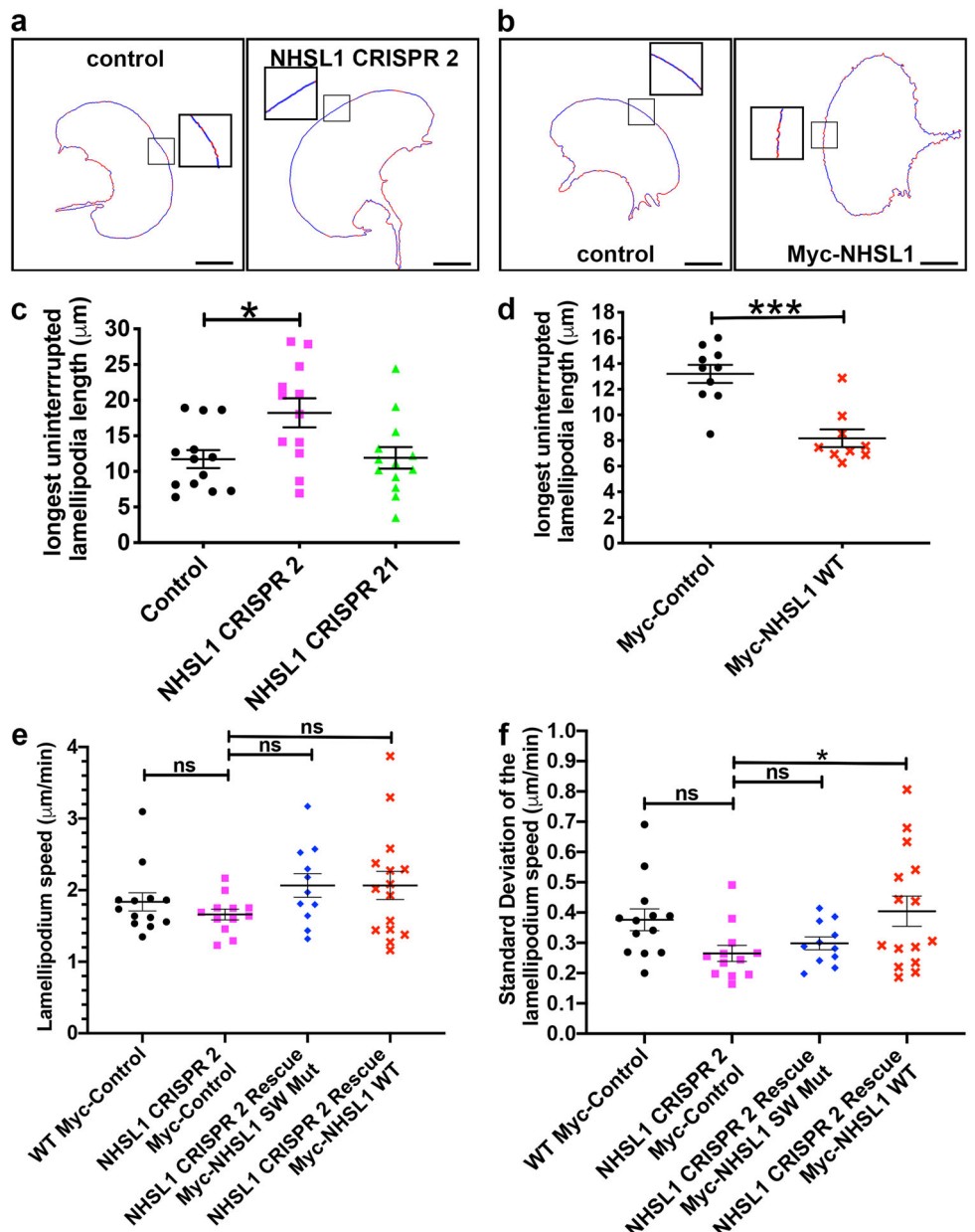

**Fig. 10 NHSL1 reduces the stability of lamellipodia protrusion. a**, **b** Movie stills showing protrusion (blue) and retraction (red) dynamics of wild-type B16-F1, NHSL1 CRISPR 2 and B16-F1 cells overexpressing Myc-tagged NHSL1. Scale bar: 20 μm. **c**, **d** NHSL1 reduces the length of the longest uninterrupted lamellipodium. Each data point represents the mean length of the longest uninterrupted lamellipodium quantified from all frames of a movie of one LifeAct-EGFP expressing B16-F1 cell (5 s intervals, 10 min duration) (see 'Methods' for details). Control (black circles), NHSL1 CRISPR 2 (magenta squares), 21 (green triangles) (**c**) and control (black circles) and Myc-NHSL1 overexpression (red crosses) (**d**). **c** Control: $n = 13$; NHSL1 CRISPR 2: $n = 12$; NHSL1 CRISPR 21: $n = 13$ cells. One-way ANOVA: $P = 0.0111$; $F(2,35) = 5.133$; Dunnett's test *$P = 0.0144$, control versus CRISPR 21: not significant: $^{ns}P = 0.9949$. **d** control $n = 10$; Myc-NHSL1 $n = 9$ cells. ***$P = 0.006$, Mann–Whitney test. **e**, **f** Quantification (see 'Methods' section) of lamellipodia protrusion speed (**e**) and lamellipodia stability (**f**) of randomly migrating wild-type B16-F1 cells or CRISPR 2 cells expressing either NHSL1 (NHSL1 WT) or the NHSL1 mutant in the Scar/WAVE complex binding sites (NHSL1 SW Mut) or Myc alone (control) plated on laminin. WT Control: $n = 13$, NHSL1 CRISPR 2: $n = 12$, Rescue Myc-NHSL1 SW Mut: $n = 11$, Rescue Myc-NHSL1 SW WT: $n = 15$ cells. One-way ANOVA: $P = 0.2097$, $F(3,47) = 1.568$; Dunnett's test: NHSL1 CRISPR 2 vs Rescue SW mut: $^{ns}P = 0.2043$; NHSL1 CRISPR 2 vs Rescue SW WT: $^{ns}P = 0.1573$; NHSL1 CRISPR 2 vs WT control: $^{ns}P = 0.7639$. **f** The standard deviation of the cone speed shows the fluctuation of speeds along the edge and serves as a measure for the stability of lamellipodial protrusions. WT control: $n = 13$, NHSL1 CRISPR 2: $n = 12$, Rescue Myc-NHSL1 SW Mut: $n = 11$, Rescue Myc-NHSL1 SW WT: $n = 15$ cells. One-way ANOVA: $P = 0.0412$, $F(3,47) = 2.971$; Dunnett's test: NHSL1 CRISPR 2 vs Rescue SW mut: $^{ns}P = 0.8865$; NHSL1 CRISPR 2 vs Rescue SW WT: *$P = 0.0299$; NHSL1 CRISPR 2 vs WT control: $^{ns}P = 0.1173$. **c–f** Results are mean ± SEM (error bars), four independent biological repeats. Source data are provided as a Source Data file.

WAVE-Arp2/3 pathway negatively regulating Scar/WAVE and thus also Arp2/3 activity.

Interestingly, the two Rac binding sites in NHSL1 do not resemble any known Rac binding sites such as the CRIB or DUF1394 domains.

The latter domain is found in the Scar/WAVE complex component Sra1/PIR121 but also in CYRI. CYRI binds to active Rac via a DUF1394 domain thereby competing with Sra1/PIR121 in the Scar/

WAVE complex for Rac binding and indirectly counteracting Scar/WAVE recruitment to the leading edge[38].

Surprisingly, we found that NHSL1 reduces cell migration via the Scar/WAVE complex. This is interesting since the Scar/WAVE complex also directly binds to Rac and this increases Arp2/3 activation via the Scar/WAVE complex. A similar paradoxical regulation has been observed at the level of the Arp2/3 complex: Arpin negatively regulates Arp2/3 activity whereas the Scar/WAVE complex positively regulates Arp2/3 activity, and both of these activities occur downstream of active Rac at the same time in a protruding lamellipodium. Through this paradoxical circuitry, called an incoherent feedforward loop[39], Arpin controls cell migration speed and persistence[40].

Similar to NHSL1, Lpd functions downstream of active Rac but in contrast, Lpd promotes cell migration speed and persistence[6,7]. Both Lpd and NHSL1 appear to co-localise at the very edge of lamellipodia during protrusion (Supplementary Fig. 1e and Supplementary Movie 3). We, therefore, propose that NHSL1 together with Lpd forms an incoherent feedforward loop at the level of the Scar/WAVE complex downstream of active Rac at the leading edge of cells in which Lpd promotes whereas NHSL1 inhibits cell migration via Scar/WAVE to enable cell steering.

## Methods

**Molecular biology, plasmids and reagents**. Full-length human NHSL1d cDNA including exon 1d and excluding exon 5 was generated by cloning overlapping EST clones Flj35425 and DKFZp686P1949 into pENTR3C (Invitrogen). Fragments of NHSL1 (Figs. 1e, 3a and Supplementary Fig. 7a) were cloned into pENTR3C (Invitrogen). NHSL1 cDNA in pENTR was mutated using Quikchange (Agilent) to create double (sites 1 + 2 or sites 2 + 3) and triple (sites 1 + 2 + 3) SH3 binding sites: Site 1: aa518–524(R > A)AS(P > A)LK(P > A); Site 2: aa625–633KKPPLP(P > A)S(R > A); Site 3: aa836–844 KPK(P > A)KV(P > A)ER.

ARPC1B (HsCD00370547), ARPC3 (HsCD00288065), Sra1 (CYFIP1; HsCD00042136), PIR121 (CYFIP2; HsCD00045545), Nap1 (NCKAP1; HsCD00045562), Abi2 (HsCD00042752) and HSPC300 (C3orf10; HsCD00045008) cloned into pENTR233 or pDONR221 (Harvard Institute of Proteomics). pDONR221-WAVE-2 (German Resource Centre for Genome Research). hsAbi1d (BC024254; Geneservice) full-length and Abi1d-Δ- SH3 (aa 1–417) were cloned into pENTR11. Human Lpd (AY494951) was amplified and cloned into pENTR3C (Invitrogen)[6]. ARPC1B-mVenus-P2A-ARPC3-mTurq2 was generated by Gibson assembly (NEB Hifi, New England Biolabs, Inc.) in pENTR3C (Invitrogen). pENTR3C-ARPC1B-mVenus-P2A-ARPC3-mTurq2 was transferred to pRK5-Myc-DEST for CMV driven expression in mammalian cells.

cDNAs in pENTR or pDONR were transferred into tagged mammalian expression vectors using Gateway® recombination (Invitrogen): pCAG-DEST-EGFP, pCAG-DEST- EGFP-T2A-Puro, pCAG-EGFP-DEST-IRES-Puro, pRK5-Myc-DEST, pCAG-Myc-DEST- IRES-Puro, pCAG-Myc-DEST-IRES-BLAST, pCAG-Myc-DEST-IRES-Puro-T2A-LifeAct-EGFP, pCDNA3.1-mScarlet-I-DEST, pV16-gateway (MBP), which were generated by traditional restriction-based cloning or Gibson assembly (NEB Hifi, New England Biolabs, Inc.) cloning.

The SH3 domains of Abi1d, c-Abl and eight fragments covering the entire length of NHSL1 were cloned by PCR and restriction cloning into pGEX-6P1 and purified from E. coli on Glutathione-sepharose (Amersham). Dominant active Myc-tagged Rac1-Q61L in pRK5-Myc was kindly provided by Laura Machesky (Beatson Institute, Glasgow, UK).

The NHSL1 shRNA constructs (shRNA A: TGCCCATTCTGTGATGATTA shRNA B: TGGATAAATCCCTATCAAGA), Control-shRNA: GCCGATAACCGAGAATACC were cloned into pLL3.7Puro which is derived from pLL3.7 with the addition of a puromycin selection. All constructs were verified by sequencing.

The sequences of all primers used are supplied as Supplementary Data 1.

**Northern blot and qPCR**. The murine multiple tissue northern blot (Ambion) was probed with a 650 bp NHSL1 fragment covering GST-NHSL1-6 (Fig. 1a, e) according to the manufacturer's instructions.

For Fig. 2b, RNA was extracted from cells using the RNeasy Mini Qiagen kit. cDNA was prepared using Superscript IV (Invitrogen) and expression levels of NHSL1 were quantified using standard curves relative to GusB using gene-specific primers (PrimeTime qPCR primer assays, IDT):

GusB: Mm.PT.39a.22214848 (primer 1: ACCACACCCAGCCAATAAAG; primer 2: AGCAATGGTACCGGCAG) NHSL1 (detecting all isoforms): Mm.PT.58.41480072 (primer 1: TCTTCATCTTGATAATCGTCGCA; primer 2: CAAGCTCAACCTCAAATCAGTG) in combination with Bioline Sensimix on a Roche lightcycler 480II qPCR machine.

For Supplementary Fig. 5b, mRNA levels were measured using the Cells to Ct Kit (Applied Biosystems) and an NHSL1-specific Taqman assay (Hs00325291_m1, Applied Biosystems). The amount of remaining mRNA was calculated with the deltaCt method.

**Generation of CRISPR cell lines**. Stable CRISPR NHSL1 knockout B16-F1 cell lines were made by knocking in a stop codon into exon 2, which is common to all isoforms. We utilised a CRISPR-Cas9 nickase approach with two sgRNA's, which minimises potential off-target effects. To generate the donor plasmid for homology-directed repair, genomic DNA from B16-F1 cells (QiaAmp genomic DNA kit, Qiagen) was amplified with primers adding a stop codon at the beginning of exon 2 of NHSL1 common to all isoforms and cloned into the targeting plasmid pHR110PA-1(5′MCS-EF1-RFP + Puro-3′MCS) (SBI). Two closely spaced NHSL1-specific sgRNA(sgRNA1: caccgTC-GACTCTCCTCGTCCAAGT; sgRNA2: caccgCTGTCCACTACACGGCACCA) were designed using http://crispr.mit.edu and cloned into pX330S-2 (Addgene 58778) and pX330A_D10A_x2 (Addgene 58772) harbouring the Cas9 nickase and combined into one plasmid using a golden gate reaction[33]. B16-F1 cells were transiently transfected with both targeting plasmid and nickase plasmid including 2 sgRNA's and after selection with puromycin, clonal cell lines were generated by limited dilution and reduction in expression of NHSL1 tested in a western blot. Cells were transiently transfected with a plasmid driving Cre recombinase expression from a PGK promoter and FACS sorted for the absence of RFP. Genomic DNA was extracted using Qiagen QIAamp DNA mini kit and knockout cell lines genotyped using primers amplifying insertion of the selection cassette into the correct locus (primer 1: aggtctacactctgccgttctggg; primer 2: CGGCCGCtgTCTAGATTTTTGAa) or to detect WT or KO alleles after Cre mediated excision of the selection cassette: Primer pair 1: (gccccagcctcgcaatg; ccgtccagcaagcccagaga); primer pair 2: (aatcctcgtgccccagcctc; gcggacgtgcaagccagtta).

**Antibodies**. Immobilised GST-NHSL1-6, 7, 8 (mAb) or GST-aa387-546NHSL1 (pAb) (Fig. 1f) were digested with Prescission protease (Amersham Pharmacia Biotech) and used to produce monoclonal antibodies by PEG induced fusion of primary B cells isolated from popliteal lymph nodes with the mouse myeloma cell line P3-X63-Ag-8. The NHSL1 monoclonal antibody was subcloned twice (clone C286F5E1; IgG1) and used for western blot: undiluted hybridoma supernatant, immunofluorescence: 50 µg/ml. GST-aa387-546NHSL1 (Fig. 1f) after digestion with Prescission protease (Amersham Pharmacia Biotech) was used to raise polyclonal rabbit antiserum #4457 (Eurogentec) and used for western blot: 1:2000, immunofluorescence: 1:400. Commercial primary antibodies: EGFP, western blot: 1:2000 (11814460001, Roche); Myc, western blot: 1:5000 (M5546, 9E10, Sigma); MBP, western blot: 1:10,000 (E8032S, New England Biolabs); Abi1, western blot: 1:1000 (MBL, clone 1B9, D147-3), immunofluorescence: 1:100; Scar/WAVE1, immunofluorescence: 1:50 (BD 612276); Scar/WAVE2 rabbit mAb, western blot: 1:1000, immunofluorescence: 1:50 (D2C8, CST 3659), ARPC2, western blot: 1:1000, immunofluorescence: 1:100 (07-227-I-100UG, Millipore). Secondary antibodies: HRP-goat anti-rabbit, western blot: 1:2000 (P044801, Agilent-Dako); HRP-goat anti-mouse, western blot: 1:2000 (P044701, Agilent-Dako).

**Cell culture and transient transfections**. HEK 293FT cells (ThermoFisher Scientific R70007) and B16-F1 mouse melanoma cells (ATCC CRL-6323) were cultured in Dulbecco's modified Eagle's medium containing penicillin/streptomycin, L-glutamine and 10% fetal bovine serum. MCF10A cells (ATCC CRL- 10317) were cultured[41] in DMEM/F12, 5%(v/v) horse serum, 20 ng/ml EGF, 0.5 µg/ml hydrocortisone, 100 ng/ml Choleratoxin, 10ug/ml insulin, and penicillin/streptomycin. Ag-8 cell (ATCC CRL-6323) were cultured in OptiMEM. Transient transfections in HEK 293FT cells were carried out using Lipofectamine 2000 (Invitrogen) according to the manufacturer's instructions. Transient transfections with B16-F1 cells were carried out with X-tremeGene 9 (Roche) and replaced with normal growth media 4–6 h after transfection. All cells were maintained at 37 °C in 5% (MCF10A) or 10% (B16-F1, HEK 293FT) CO$_2$.

**Immunoprecipitations, pulldowns and western blotting**. Cells were lysed in glutathione S-transferase (GST) buffer (50 mM Tris-HCL, pH 7.4, 200 mM NaCl, 1% NP-40, 2 mM MgCl$_2$, 10% glycerol, NaF +Na$_3$VO$_4$, and complete mini tablets without EDTA, Roche). Lysates were incubated on ice for 15 min and centrifuged at 17,000 × g at 4 °C for 10 min. Protein concentration was then determined (Pierce BCA protein assay kit; ThermoFisher Scientific). For pulldowns, the lysate was incubated with either glutathione beads (Amersham), GFP-trap or Myc-trap beads (Chromotek). For immunoprecipitations, Protein A bead precleared lysates were incubated with primary antibody or non-immune control IgG followed by 1% BSA blocked protein A beads (Pierce) or protein A/G beads (Alpha Diagnostics). For GFP-trap or Myc-trap pulldowns, beads were blocked with 1% BSA before incubating with lysates for 1–2 h or 30 min, respectively. Following bead incubation, all beads were washed with lysis buffer, separated on SDS- PAGE gels and transferred onto Immobilon-P membranes (EMD Millipore). Western blotting was performed by transferring at 100 V, 350 mA, 50 W for 1.5 h before blocking in 5%(w/v) BSA, 5%(w/v) or 10%(w/v) milk in TBS-T (20 mM Tris-base, 154 mM NaCl, 0.1% (v/v) Tween-20, pH7.6), overnight followed by 1 h incubation with the indicated primary antibodies followed by HRP conjugated secondary antibodies (Dako) for 1 h at

room temperature with three washes of TBS-T, TBS-T + 0.5 M NaCl and TBS-T + 0.5% (v/v) TX-100. Blots were developed with the Immun-Star WesternC ECL kit (Bio-Rad Laboratories) using the Bio-Rad Imager and ImageLab software.

**Far-western blot**. GST and MBP fusion proteins were purified from BL21-CodonPlus (DE3)-RP *E. coli* (Stratagene) using glutathione (GE Healthcare) or amylose (New England Biolabs, Inc.) beads. Purified GST-NHSL1 fragments were separated on SDS-PAGE and transferred onto PVDF membranes, blocked in 5% BSA/TBS-T and overlayed with 10 µg/ml of purified MBP, MBP-Abi full-length or MBP-Abi-delta-SH3 in 5% (w/v) BSA, TBS-T and MBP were detected with 1:10,000 MBP antibodies in 5% BSA/TBS-T, E8032S (New England Biolabs).

**Immunofluorescence and live-cell imaging**. For immunofluorescence analysis cells were plated on nitric acid-washed coverslips (Hecht-Assistant), coated with 25 µg/ml laminin (L2020, Sigma) and fixed 10 min with 4% paraformaldehyde-PHEM (60 mM PIPES, 25 mM HEPES, 10 mM EGTA, 2 mM MgCl₂, 0.12 M sucrose). For NHSL1 mAb and ARPC2 pAb (07-227-I-100UG, Millipore): cells were permeabilised for 2 min with 0.1% Triton-X-100/TBS and quenched for 10 min with 1 mg/ml Sodium Borohydride; For NHSL1 pAb: cells were permeabilised with 0.05% saponin, 2 h at RT. For all antibodies: cells were blocked with 10% normal goat serum and 10% BSA, TBS. Secondary antibodies: goat anti-rabbit, or anti-mouse Alexa488 or 568, 1:500 (Invitrogen) and mounted in Prolong Diamond (Invitrogen). For Alexafluor 488 or 568-conjugated Phalloidin staining: cells were fixed 20 min with 4% paraformaldehyde-PHEM at room temperature. Cells were permeabilised for 2 min with 0.1% Triton-X-100/TBS and washed in a TBS solution. Cells were stained with Alexa fluor 488 or 568-conjugated Phalloidin (Invitrogen) diluted 1:250 in 10% normal goat serum/TBS for 1 h at room temperature, then washed three times in a TBS solution, and mounted in a Prolong Diamond (Invitrogen). Cells were imaged on an IX81 Olympus microscope (see below). Equal exposure times were used for all cells and all conditions for a given experiment. Intensity values were then normalised for each experiment to the mean of the control condition for that experiment.

Cell profiler analysis of F-actin content in HEK293 cells was analysed using Cellprofiler 2.0 software (http://www.cellprofiler.org, Broad Institute). Briefly, cells were identified based on nuclear DAPI and phalloidin staining. F-Actin content was measured using the MeasureObjectIntensity module. Data was exported into a text file and analysed with Graphpad Prism. Values obtained from control samples (GFP or scrambled shRNA) were set to 100%.

For low magnification phase contrast and high magnification imaging, cells were plated on 12-well tissue culture dishes or glass-bottom dishes (Ibidi; 81218-200) coated with 10 µg/ml fibronectin (F1141, Sigma) or coated with 25 µg/ml laminin (L2020, Sigma). For immunofluorescence and live imaging, an IX 81 microscope (Olympus), with a Solent Scientific incubation chamber, filter wheels (Sutter), an ASI X-Y stage, Cascade II 512B camera (Photometrics), and 4× UPlanFL, 10× UPlanFL, ×60 Plan-Apochromat NA 1.45, or 100× UPlan-Apochromat S NA 1.4 objective lenses controlled by MetaMorph software or an LSM 880 Airyscan confocal microscope (Zeiss) with an environmental chamber (37 ℃; 10% CO₂) driven by Zen Black software with a ×63 oil (NA 1.4) Plan-Apochromat objective (Zeiss) was used. Supplementary movies were prepared in Metamorph 6.1 and Fiji 1.53c. Adobe Photoshop CS6 and Creative Cloud for the presentation of the images in the figures.

**Scratch assays and quantification of random cell migration speed and persistence**. Confluent control or shRNA MCF10A cells were scratched with a P200 pipette tip and treated with mitomycin C to inhibit cell proliferation. Movies were acquired for 12 h with a frame every 5 min. The scratch area was measured at 0 and 12 h with Fiji.

For random migration, B16-F1 cells were plated onto fibronectin or laminin-coated 12-well dishes for 24 h before imaging for 24 h every 5 min. Cells were manually tracked by their nuclear position using the Manual Tracking plugin (FIJI), and the cell track coordinates were imported into Mathematica for analysis using the Chemotaxis Analysis Notebook v1.6β (G. Dunn, King's College London, UK[6]).

Speed and persistence measurements from tracking data are susceptible to positional error (due to manual tracking) as well as biological noise (from cell morphology changes or nuclear repositioning). To address the former, we estimated the positional error by tracking the same cell multiple times (each time blinded to the previous track), and then calculated the time interval (as an integer multiple of the frame interval) over which this error fell to at least below 10% of the average displacement measurement. We selected the smallest multiple for which this condition was true to avoid error from sampling too infrequently to measure the true path length. This time interval is termed the 'usable time interval' and denoted $\delta t$ (see Fig. 2c, upper panel). In order to be consistent over data sets, we adopted a usable time interval that was common to all data within a dataset. This was $\delta t = 15$ (equivalent to a multiple of 3 frame intervals or taking a displacement measurement every 3 frames) for the CRISPR knockout and overexpression experiments on fibronectin and $\delta t = 25$ min (equivalent to a multiple of 5 frame intervals or taking a displacement measurement every 5 frames) for the CRISPR 2 knockout and rescue experiments on laminin. This 'usable time interval' approach also helps to reduce the impact of biological noise on the data.

Cell speed is the displacement ($d_n$) over $\delta t$, where $n$ denotes which interval of the track we are measuring (for example $n = 1$ would indicate the first 15 min,

$n = 2$ the second 15 min etc.). The Mean Track Speed (MTS) is then the average cell speed over the whole track length (Fig. 2c, upper panel, boxed equation).

Cell persistence has traditionally been defined by the directionality ratio: the net (straight line) displacement divided by the total track length. This measurement comes with additional issues to noise, namely that it is dependent on interval size, track length and cell speed. The interval size is determined by the usable time interval, $\delta t$ for the same reasons as previously described. To address the dependence on track length, we measured persistence over another time interval, denoted $\Delta t$, which is an integer multiple of $\delta t$ (see Fig. 2c, lower panel). This multiple is called the Time Ratio (TR, equal to $\Delta t / \delta t$). This approach allows tracks of different total length to be compared equally, without skewing our results.

Persistence is defined by the Mean Track Persistence (MTP), equal to the average directionality ratio over the intervals $\Delta t$ ($\Delta t = 60$ min for TR = 4), multiplied by a normalisation factor, $\alpha$, which sets the maximum persistence to 1 (a perfectly straight migration path) and the minimum persistence to that of a purely random walk. The directionality ratio is the net displacement, $D_N$, where $N$ is the interval number for each $\Delta t$ (i.e. $N = 1$ indicates the first 60 min, $N = 2$ indicates the second 60 min etc.) divided by the track length over $\Delta t$ (the sum of the displacements $d_n$) (Fig. 2c, lower panel, boxed equation).

Finally, the issue of cell speed dependence can be addressed by selecting an appropriate combination of $\delta t$ and TR. Plotting MTS as a function of $\delta t$ reveals that speed falls off exponentially with increasing time interval, for all intervals which yield displacement values above the level of track noise. The mean persistence profile is a plot of the MTP obtained for all possible choices of $\delta t$ for a given TR. In the mean persistence profile, the persistence normally starts off low at small speed intervals and rapidly increases to a peak before falling off again. To avoid noise, we chose a suitable $\delta t$ and TR combination with the MTP to be measured close to this peak. The same TR must be used across all data sets so they are directly comparable; thus, another compromise must be that the TR chosen is appropriate for all populations. After some investigation, a $\delta t$ of 15 and TR of 4 was chosen, for the CRISPR knockout and overexpression experiments on fibronectin and a $\delta t$ of 25 and TR of 2 was chosen for the CRISPR 2 knockout and rescue experiments on laminin. These combinations ensured in both cases that tracks were long enough to contain multiple intervals.

The direction autocorrelation, directionality ratio over time and mean square displacement was calculated and plotted using excel macros provided in Gorelik and Gautreau, Nature Methods 2014[18] according to the instruction provided.

**Quantification of lamellipodia dynamics**. B16-F1 cells expressing LifeAct-EGFP plated on laminin were imaged for 10 min at 5 s per frame using an LSM 880 Airyscan confocal microscope or IX81 Olympus microscope and cells were automatically segmented and protrusion vectors at each pixel along the cell edge calculated using MATLAB code kindly provided by Andrew Jamieson and Gaudenz Danuser (UT Southwestern, USA): the Windowing-protrusion analysis package from the Danuser lab can be downloaded at: https://github.com/DanuserLab/Windowing-Protrusion.

Tracking the direction of migration of these cells allowed us to automatically quantify membrane protrusion speed only of the lamellipodium in the direction of migration. Custom written MATLAB code and Metamorph journals were used to calculate protrusion speed, the longest uninterrupted lamellipodium and the length distribution of protrusions from all frames of the movie from the leading-edge automatic profiling data. This custom written MATLAB code which has not been published before is available as Supplementary Software 1.

The length of the longest uninterrupted lamellipodium was quantified as follows: Each data point in Fig. 10c, d represents the mean length of the longest uninterrupted lamellipodium quantified from all frames of a movie of one LifeAct-EGFP expressing B16-F1 cell (5 s intervals, 10 min duration). The lamellipodia protrusion speed (Fig. 10e) and lamellipodia stability (standard deviation of the cone speed) (Fig. 10f) of randomly migrating B16-F1 cells plated on laminin were quantified by automatically segmenting the cell outline in each frame of the movies and calculating protrusion vectors at each pixel along the cell edge. Lamellipodia protrusion speed was quantified in a cone of 60° centred about the direction of migration as quantified by tracking the centroid of the cell. The standard deviation of the cone speed shows the fluctuation of speeds along the edge and serves as a measure for the stability of lamellipodial protrusions. Fiji and Excel were used to quantify the LifeAct-EGFP intensity, total cell area, lamellipodia width and length and microspike number per length of lamellipodium in the first frame of each movie. Lamellipodia length was quantified using the Fiji plugin "measure_ROI" (http://www.optinav.info/Measure-Roi.htm: Measure_Roi_Curve.java). This plugin measures the length of curved objects: For each particle, it first finds the points with the largest separation. These are designated as the endpoints. Next, it computes two curves that connect the endpoints along the left and right sides of the object. These curves are parameterised by arc length. The left and right side curves are averaged for each arc length to create a centerline curve. The arc length of the centerline curve is reported as the length of the object. Lamellipodia width was measured using the line tool in Fiji to draw lines at 90° angle relative to the leading edge at ten roughly equal distance points along the lamellipodium and these were averaged. Supplementary movies were prepared in Fiji 1.53c.

**FRET-FLIM analysis of Arp2/3 activity**. B16-F1 cells plated on nitric acid cleaned No 1.5 coverslips (Hecht Assistent), coated with 25 µg/ml laminin (Sigma) were fixed for 20 min with 4% paraformaldehyde-PHEM (60 mM PIPES, 25 mM

HEPES, 10 mM EGTA, 2 mM MgCl₂, 0.12 M sucrose), permeabilised for 2 min with 0.1% TX-100/PBS, background fluorescence quenched with 1 mg/ml sodium borohydride in PBS and mounted in ProLong Diamond (ThermoFisher).

Time-domain FLIM data were acquired via a time-correlated single-photon counting (TCSPC) custom-built, automated, 2-photon microscope. Briefly, this consisted of a Modelocked femtosecond Ti:Sapphire laser (Coherent Vision II; Coherent (UK) Ltd, Scotland) for fluorescence excitation, a dual-axis scanner, a photomultiplier detector (HPM-100-06; Becker & Hickl GmbH, Germany) and TCSPC electronics (SPC830; Becker & Hickl GmbH, Germany) controlled by LabView 2018. Images were acquired using a 1.3 NA ×40 Plan Fluor Oil Immersion objective (Nikon Instruments Ltd, UK). Fluorescence lifetimes were determined for every pixel using a modified Levenberg–Marquardt fitting technique[42]. The FLIM images were batch analysed by running an in-house exponential fitting algorithm (TRI2 software) written in LabWindows/CVI 2017 (National Instruments, Austin, TX)[42], which can be obtained here: https://flimlib.github.io. The fitting parameters for each time-resolved intensity image were recorded in individual output files and used to generate a distribution of lifetime and an average fluorescence lifetime. FLIM/FRET analysis was performed to investigate Arp2/3 activity using the Arp2/3 biosensor (ARPC1B-mVenus-P2A-ARPC3-mTurq2). FRET efficiencies were calculated based on the equation $E = 1 - \tau_{DA}/\tau_D$, where $\tau_D$ and $\tau_{DA}$ are the measured fluorescence lifetimes of the donor in the absence and presence of the acceptor, respectively.

**Analysis of F-actin retrograde flow**. Cells transfected with LifeAct-GFP were harvested and re-plated onto 35 mm glass-bottomed dishes (Ibidi), pre-coated with laminin (L2020, Sigma). The dishes were centrifuged for 2 min at 200 × g and incubated for ≥3 h before imaging. The cells were then counterstained with 0.5 μM SiR-DNA (Spirochrome) for ~30 min prior to imaging. This stain was not used in actin flow experiments but was used for cell tracking in the quantification of lamellipodial dynamics experiments. Videos were captured on a Zeiss LSM880 Airyscan confocal microscope driven by Zen Black edition software at 5× zoom, 0.4–1.0% laser power and frame rate of 3.22 s with at least 20 frames per movie.

**Pre-processing**. Zen Black edition software was used to Airyscan-process the raw data with automatic parameter fitting. Lamellipodia were segmented in Fiji by using a semi-automated method: frame-by-frame normalisation of contrast and Gaussian blur, and then correcting for photobleaching over the entire movie by histogram matching and using the result to construct a binary mask determined by semi-automatic thresholding with manual editing to remove certain features such as vesicles. This mask was then multiplied by the corresponding contrast-enhanced original Airyscan movie (not blurred or bleach-corrected) to obtain a segmented movie which formed the input for PIV. Movies were truncated when loss of focus, lamellipodial collapse or significant translocation of the cell out of the field of view occurred.

**PIV analysis**. PIV analysis was carried out using a custom MATLAB script which can be downloaded here: https://github.com/stemarcotti/PIV. The script utilises a two-dimensional cross-correlation algorithm adapted from classical PIV and is published here in refs. [43,44]. The method relies on searching for a region of interest (source area) in a larger region of a subsequent frame (search area) and finding the best match by cross-correlation. For our analysis we optimised the region size, overlap and correlation threshold using a source box size of 0.3 μm, search box size of 0.5 μm, grid size (defining the overlap) of 0.2 μm and correlation coefficient threshold of 0.5. Subsequently, spatial and temporal convolution are used to interpolate the displacement vectors over all the pixels in the image. Here, the spatial kernel size was 3 μm ($\sigma = 0.5$ μm) and the temporal kernel size was 15 s ($\sigma = 6$ s). All colourmaps presented were generated by setting a maximum flow velocity of 3 μm/min, and the number of vector arrows displayed is arbitrarily defined by the distance between them to represent the interpolation.

**Statistical analyses**. Data were tested for normal distribution by D'Agostino & Pearson and Shapiro–Wilk normality tests. Statistical analysis was performed in Prism v7, v8, v9 (GraphPad Software) using a Student's t-test, One-way ANOVA or non-parametric Kruskal–Wallis test with appropriate post-hoc tests (see figure legends in each case). P values < 0.05 were considered significant.

**Reporting summary**. Further information on research design is available in the Nature Research Reporting Summary linked to this article.

## Data availability
The imaging datasets generated are available from the corresponding author on reasonable request. All quantifications and full western blots from this study are provided in the Source Data file. Source data are provided with this paper.

## Code availability
Custom written MATLAB code which has not been published before is available as supplementary software 1. This zip file contains the additional Matlab codes and instructions for "Quantification of lamellipodia protrusion speed" and for "Analysis of length distribution of lamellipodia" to be used in conjunction with the Windowing-

protrusion analysis package from the Danuser lab which can be downloaded at: https://github.com/DanuserLab/Windowing-Protrusion. The PIV Matlab script can be downloaded here: https://github.com/stemarcotti/PIV. The TRI2 software for FLIM analysis can be downloaded here: https://flimlib.github.io

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

## Acknowledgements

We thank Laura Machesky (Beatson Institute, Glasgow, UK) for reagents. Andrew Jamieson and Gaudenz Danuser (UT Southwestern, USA) for providing the Windowing MATLAB script. William Barrell (King's College London, UK) for qPCR training. F.M. was supported by a Medical Research Council (MRC) studentship. T.P. was supported by an Engineering and Physical Sciences Research Council (EPSRC) studentship. S.J. was supported by a Malaysian Public Service Department (PSD) studentship. This work was supported by grants from the European Research Council (ERC) under the European Union's Horizon 2020 research and innovation programme (grant agreement No 681808) (B.M.S.), CRUK Comprehensive Cancer Centre block grant (S.M.A.-B), Medical Research Council (MRC) (S.M.A.-B.), Richard Dimbleby Cancer Trust (S.M.A.-B.), Wellcome Trust (107859/Z/15/Z) (B.M.S.) (082907/Z/07/Z) (M.K.), the Biotechnology and Biological Science Research Council, UK (BB/F011431/1; BB/J000590/1; BB/N000226/1; BB/R015953/1) (M.K.) and Cancer Research UK (S.A.-B.) (C22104/A7155) (M.K.).

## Author contributions

A.L.L., S.J., T.P., A.G., S.B., A.J., R.K., G.P. and M.K. performed the experimental studies. A.L.L., S.J., T.P., F.M., L.Y., S.M., J.A.L., S.P.P., M.R.-S., A.J., R.K. and M.K. carried out the analysis. S.M.A.-B, B.M.S. and M.K. supervised the work.

## Competing interests

The authors declare no competing interests.
