## [Peer Review File · Nature Communications]

Reviewers' comments:

Reviewer #1 (Remarks to the Author):

In this manuscript, Law, Krause and colleagues report the detailed characterisation of the role of NHSL1 in cell migration using knock-out, knock-down and overexpression. They identify two binding sites through which NHSL1 binds to the Scar/WAVE complex (SWC), specifically through the SH3 domain of the Abi subunit. They report that the small GTPase Rac1 binds to NHSL1 through two binding sites and that Rac1 regulates the NHSL1-SWC interaction. They also develop a new method to measure the Arp2/3 activity in vivo through the adaptation of an in vitro FRET pair. Overall their conclusion is that NHSL1 is a negative regulator of membrane protrusion and cell migration, through the complex regulation exerted by Rac1 on the SWC.

The authors study NHSL1 because another member of the family, NHS, had been reported to localise at the leading edge of migrating cells. Here they generalise to NHSL1, which appears to be a more ubiquitous isoform. They state several times that NHS was not previously involved in cell migration, which can be discussed. NHS was also previously reported to interact with the SWC and to regulate lamellipodium formation and cell spreading (Brooks et al., Hum Mol Genet 2010), even if proper migration assays were indeed not performed in this previous publication. There is no doubt that the present manuscript goes beyond the previous characterization of NHS role, even if I have several critiques on the way experiments are performed in detail. The development of the FRET Arp2/3 biosensor is a major contribution to the field, which has overall relied on membrane protrusions and cell migration tracking as indirect read-out of Arp2/3 activity. To me, this paper is eligible to a Nature Comm publication, but only after a significant revision is performed.

1 The salient point of this NHS protein family is that the family members contain a Scar/WAVE homology domain in the N-terminus. Here we learn at the end of the manuscript that the whole study was performed on a specific splice isoform of NHSL1 that does not contain the Scar/WAVE homology domain. I would like to see the relative abundance of the splice forms containing the Scar/WAVE homology domain or not in the couple of cell systems that are characterised in the manuscript.

2 CRISPR mediated inactivation of NHSL1 is performed in B16F1. One clone seemingly corresponding to a biallelic KO and one clone seemingly corresponding to a heterozygous KO were selected and characterised further. These clones should be characterised at the genomic DNA level. The remaining protein levels, in both cases, should be evaluated by serial dilution of the control lysate. Finally, even if I appreciate that care was taken to use the dual Cas9 nickase approach, a rescue of the KO clone by expression of NHSL1 would be the ultimate proof that the reported phenotype is specific to the KO.

3 After extensive mapping, a mutant form of NHSL1, which does no longer bind to the SWC, has been isolated. The difference of phenotype between the mutant form of NHSL1 and WT NHSL1, expressed in WT cells, appears moderate (Fig.8D, 8E). The authors should rescue the phenotype of the KO B16F1 cell line with WT and mutant NHSL1. This would validate the specific phenotype of KO cells and should reveal that interaction with SWC is critical for NHSL1, if mutant NHSL1 indeed does not rescue. The implicit assumption that the Scar/WAVE homology domain is not a major determinant of the NHSL1 role in cell migration should also be tested in the rescue system.

4 The authors could go much further in their analysis of movies, which are so far used only to demonstrate colocalisation of GFP-NHSL1 with Lamellipodin and SWC subunits. Since NHSL1 negatively regulates protrusions, contrary to the positive regulators, SWC and lamellipodin, their colocalisation should be analysed over time in relation with protrusions and retractions. Does their

cross-correlation display a characteristic delay ? According to the model, we should expect that NHSL1 arrival/enrichment should predict a retraction of the membrane. Such a result would give an undeniable confirmation of current findings.

5 I was shocked by the superficiality of the Rac1 part, which fully relied on the transfection of GTPase-deficient, dominant active form. This was particularly surprising given that B16F1 cells must already have a high basal level of active Rac1, given their extraordinary lamellipodial activity. Rac1 activity should be inhibited for a comparison. In all these experiments, Rac activity should have been modulated using many ways. DA Rac must be compared to WT and DN Rac1, the nucleotide free form T17N that titrates GEFs. It is a bit old-fashioned but that works. In addition, there are Rac chemical inhibitors. Rac1 can also be down-regulated using siRNAs. Some toxins activate or inactivate GTPases and Rac1 in particular. Rac1 KO MEFs have been reported. The authors may choose the most appropriate ways of modulating Rac activity, but the manuscript cannot simply rely on the overexpression of active Rac throughout.

6 The main finding concerning Rac1, which is that it modulates the NHSL1-SWC interaction, is surprisingly only documented when all proteins are overexpressed (Fig.8B). This cannot be justified, given that the interaction between endogenous NHSL1 and endogenous SWC can be detected (Fig.4F, 4G). A major question, which should receive a clear answer, is whether the NHSL1-SWC interaction requires Rac1 activity.

7 This experiment alone where both Rac and NHSL1 are overexpressed would not be sufficient to convince that Rac1 binds to NHSL1, even if appropriately controlled. It should be complemented by a GST pull-down using GST-Rac1 Q61L (remaining GTP bound) compared to GST-Rac1 WT (mainly associated with GDP, because of GTP hydrolysis), or using GST-Rac1 WT loaded with GDP or GTP γ S. The SWC can provide a useful positive control for GTP-Rac specific binding.

8 Is the binding of Rac1 to NHSL1 direct ? The question is especially important, since no established Rac1 binding site has been detected. Given that the binding sites are mapped, this experiment is easy to perform. One binding site appears much stronger than the other one, which is at the limit of detection. This point should be mentioned in the text.

9 Single cell migration assays are central to this paper. They should not be characterized by the best parameters only. I believe that MSD is missing from Fig.2. Then to characterise the mutant form that does no longer bind to the SWC, the full set should be displayed: trajectories, MSD, Speed and directional persistence.

10 The wound healing assay is not very convincing (Fig.S2, video S3). The NHSL1 shRNA A condition displays borders, which are not sharp. Many cells are rolled up. The wound appears less wide in this shRNA A condition than in the other conditions. Ibidi inserts that can be removed offer useful alternative to scratches.

11 Fig.6F: Myc tagged Rac1 is cotransfected with GFP-NHSL1 or GFP. Myc is immunoprecipitated, WB anti-GFP. Of course, NHSL1 cannot coprecipitate if it is not expressed ! GFP alone is the appropriate control only if GFP is precipitated first, followed by a WB anti-Myc. As presented here, the appropriate control is GFP-NHSL1, with no Myc tagged protein or an irrelevant Myc tagged protein.

12 The logic of several coIPs appears strange to me. Fig4D and Fig.5AB: GFP-NHSL1 is overexpressed, but endogenous NHSL1 is precipitated with anti-NHSL1 antibodies... GFP trap would be more logical in these cases. Similarly, FigS3C is not convincing. Myc-NHSL1 is overexpressed, endogenous NHSL1 is immunoprecipitated. Strangely, no Myc tagged site mutant is detected in the IP ! What can we conclude from such an experiment ?

13 Fig.5D is very important, since it defines the NHSL1 mutant that does no longer bind to the SWC. The lack of coIP with SWC is not convincing, because less of the NHSL1 mutants that should not bind to the SWC than controls are precipitated. Overall the GFP immunoprecipitations appear very inefficient, because it does not concentrate the proteins seen in the lysates. Again I don't see why

the SWC has to be overexpressed in this case.

14 Fig8B. It is required to show the amount of NHSL1 that is immunoprecipitated, in order to compare the amount of SWC coprecipitated in this experiment.

15 Fig.10EF is really not clear. What is the unit of the x-axis ? Why this strange normalisation of each bin in the control case ?

16 The Palmitoylation experiment appear superfluous to me. I don't know why it is presented here. It is not well justified. I also doubt that the simple experiment reported here specifically reveals a palmitoyl modification, i.e. an acyl modification of a certain length. Then such an experiment immediately raises many questions that his manuscript cannot address in addition to all the rest. What is the residue modified ? Is this modification important for the activity ? The authors can keep this point for future studies.

Miscellaneous

17 Typo: occurrence -> occurrence in all FRET figures (Fig.9, Fig.10, Fig.S4)

18 FigS5: indicate on the conservation logo diagrams the residues that were mutated

19 The nomenclature for the mouse gene would be *Nhsl1* and not NHSL1, which is the human form. Then for the protein, it has to be clearly defined. It can be useful to state in the results section that the constructs were derived from the mouse protein.

Reviewer #2 (Remarks to the Author):

In this manuscript, Law et al. demonstrate that NHSL1 is a novel direct binding partner of the Scar/WAVE complex, which localizes to the edge of lamellipodia. Moreover, NHSL1, which binds to the active Rac, negatively regulates cell migration speed and persistence by restricting Arp2/3 activity. Unfortunately, this is not a well-written manuscript, and the way that data are presented (i.e., organization of figures) is at best subpar. As described below in detail, certain experimental controls are missing; individual bands instead of whole blots are presented; and the scientific rigor is not thorough in the present form of this manuscript. This reviewer also makes key suggestions, which could potentially bring the manuscript to publication quality.

Major Comments:

1. The authors claim that NHSL1 localizes to the lamellipodia edge and inhibits Arp2/3. However, based on FLIM FRET Lifetime images (Fig. 9D), Arp2/3 is shown to be activated inside the cell upon NHSL1 knockdown. How do the authors interpret this finding?

2. The authors indicate that NHSL1 inhibits Arp2/3 by binding with the SCAR/WAVE complex. However, authors do not demonstrate this directly. At minimum, the authors should show that NHSL1 regulates Arp2/3 activity via SCAR/WAVE by comparing Arp2/3 FLIM FRET results with NHSL1 knockdown to FLIM FRET results with SCAR/WAVE inhibition and simultaneous NHSL1/SCAR/WAVE inhibition.

3. The authors should also use NHSL1 SW-mut construct for lamellipodia quantification assays (Fig. 10).

4. The authors should quantify additional parameters relevant to lamellipodia dynamics such as

retrograde actin flow.

5. The far Western blot experiments (Fig. 5C) should be verified using in vitro pull-down assays. Also, in Fig. 5C, the authors should blot for GST to verify the presence of each of the 8 fragments.

6. The scientific rigor of this study is at best problematic. For instance, in Figs. 8D and 8E, are the differences between NHSL1-WT and NHSL1-SW mut statistically different?

7. Key experiments should be verified with multiple cell lines (Figs. 2, 8D-E, 9). Along these lines, the authors should show data with several CRISPR cell lines.

8. Migration data are provided using old-fashioned 2D assays. The authors should verify their 2D findings using different ECMs (e.g., collagen type I), and also perform 3D migration assays.

9. What is the role of NHSL1 in vivo?

Minor Comments:

1. In Fig. 1, the authors should also show that endogenous NHSL1 is localized at the very edge of lamellipodia (which is actually shown in Fig. 4A for B16F1 mouse melanoma cells).

2. The entire blot of Figs. 1D and 1E should be provided as Supplemental data.

3. Quantification of Western blots in Fig. 2A should be provided as a Supplemental Figure.

4. Fig. 3 should be merged with Figures 4 and 5.

5. The authors should show the entire blot of Fig. 4D.

6. The authors should present the entire blot of Fig. 6F.

7. Figs 6 and 7 should be combined.

8. In Figures 6G, 7B, the amount of NHSL1 fragments loaded is unequal. The authors should repeat these experiments to ensure closer to equal loading of each fragment.

9. In the introduction, the authors state that the SCAR/WAVE complex is autoinhibited. However, to the best of our knowledge, the WASP complex is autoinhibited but the SCAR/WAVE complex is not. The authors should provide specific references for this claim (no citation currently).

Reviewer #3 (Remarks to the Author):

Nance-Horan Syndrome (NHS) is an X-linked developmental disorder with symptoms of congenital cataract, dental anomalies, characteristic dysmorphic features and developmental delay. Nance-Horan protein (NHS) has been linked to regulate actin remodeling and cell morphology. The authors in this manuscript characterize another member of this family of proteins, NHSL1 and show that it

serves as a negative regulator of cell migration. The authors also show that depletion of this protein induces higher migration speed and activation of the Arp complex thereby increasing stability of lamellipodia. Though the study is interesting, and the authors do elucidate a novel regulator of the wave complex and Arp activity, there are some inconsistencies that should be addressed. Broadly the inconsistencies are on the following front:

1. Reading through the texts, it seems many of the results are over-interpreted and the authors could be a bit milder in their approach.
2. There are several technical problems specially in the biochemical results which will be pointed out in my comments.
3. The figures can be shortened and clubbed together to make the data a lot tighter.
4. Some studies lack proper controls.

Major Comments:

1. Since NHS has been a well characterized member of this family and has been shown to have similar localization, a thorough comparison needs to be done for NHS and NHSL1 in figure 1 based on its domains. This will reveal how similar or dis-similar these two members are.
2. Throughout the western blot figures (Fig 1D, 2A, S2C), the authors have not provided any loading controls. These need to be provided.

The authors need to state whether the images in Fig 1D are from the same or different blots. If they are from different blots, they need to have different MW marker lanes.

Also, for fig 1D, it will be a good idea to characterize the antibody through specific KD of NHSL1 and evaluating NHSL1 along with other members of the family. Since the authors also use the antibody for IF in later figures, it will be good to show how specific the antibody is in IF staining. The authors do need to provide the full blots for Fig 1D to show how specific the antibody is.

Another inconsistency with the manuscript is the placement of the molecular weight markers. While in fig 1D, 2A, 4D, the endogenous NHSL1 line up with the 250 Kd marker, Fig 6F has EGFP tagged NHSL1 running lower than 250 Kd.

In another instance while endogenous Abi has been shown to run at 55 Kd (Fig S3B), EGFP tagged Abi has also been shown to run at 55 Kd (Fig S3A) which should run at around 80 Kd. The authors need to clarify these technical issues.

3. As mentioned earlier the authors need to provide loading controls for Fig 2A.

Figure 2B: The average random migration speed for B16-F1 cells determined by the authors was 0.3 $\mu\text{m}/\text{min}$. This appears to be very slow compared to other reports for this cell line (Kage F et al., Nat Commun. 2017 Mar 22; 8:14832) One possible explanation for that could be that the authors have performed these random migration assays on fibronectin, as opposed to all other experiments in which they have been using laminin-coating. What is the reason for that? I would suggest repeating random migration assays using laminin as substrate.

Since the authors show a significant effect of NHSL1 KO in cell migration speeds, the authors need to characterize the KO cells better. To this end the authors can report the following parameters:

- a. Perform Rhodamine-phalloidin staining to characterize lamellipodia in KO cells.
- b. Report initial Lamellipodial width in KO cells vs WT cells.
- c. Evaluate the levels of the lev of WAVE complex proteins and Arp2/3 in the KO cells.
- d. Evaluate the levels of NHS in the NHSL1 KO cells.

4. Fig 5A does not match the description as mentioned in the text. The authors need to clear if they used myc-NHSL1 or EGFP-NHSL1 and alter the text/image accordingly.

The authors need to show a Coomassie gel for the GST fragments isolated in Fig 5C. Can the authors state why they see smears instead of single bands in Fig 5C?

Figure 5D: The pulldown efficiency is overall very weak, which makes the result that mutated NHSL1 (site 2+3, site 1+2+3) binds less efficiently to Abi1 questionable. I suggest optimizing the pulldown

efficiency by refraining from overexpression of all WAVE complex components simultaneously and instead focusing on the interaction of mutated versions of NHSL1 and Abi1 by sole ectopic expression of those.

5. The authors need to have similar expression of the various fragments in Fig 7B in order to conclude that Fragment 1 of NHSL1 is the major interactor of Rac1.

Also, it would be good to express Rac1 K17N (DN mutant) to show that it hampers localization and interaction of NHSL1 in membrane and with Abi respectively.

6. Fig 8A seems incomplete and do not contribute significantly to the major conclusions of the paper. The authors can put it in the supplementary figure or figure out the site and better characterize this PTM of NHSL1.

Fig 8B: The authors show here that expression of dominant-active Rac increases binding between NHSL1 and the WAVE complex. This interpretation, especially in consideration of the small difference between DA Rac and vector control, is not valid unless the authors show respective expression levels and pulldown efficiencies of NHSL1 in both conditions. At this stage it cannot be excluded that the pulldown of NHSL1 was less efficient in case of vector control, which would explain reduced binding to the WAVE complex independent of RAC activity. Moreover, in fig 4D the authors have had a descent pull down of WAVE2 in NHSL1 IP samples while in this fig (8B), it seems to be dependent on the activity of DA Rac? Can the authors comment on this?

Also, it would be good to have a DN Rac control for Fig 8B.

It is confusing whether Fig 8D, E were done in KO background or WT background? It would be good to do these experiments in KO background.

Figure 8E: Instead of lamellipodia intensity/area I would rather suggest using the term Lifeact intensity/ lamellipodial area, because it is the Lifeact signal that was measured. Moreover, it would be good repeating those measurements on glutaraldehyde-fixed and phalloidin-stained cells. As the Lifeact expression level itself impacts on the outcome of the measurement, the authors should preferentially use phalloidin to visualize endogenous F-actin in lamellipodia of cells with diverse genetic backgrounds.

7. Figure 9D and Figure S5: Both NHSL1 CRISPR clones appear to have an untypically elongated cell shape compared to B16-F1 controls. Furthermore, activated A

Response to reviewers comments:

We would like to thank the referees for their constructive criticism which we have taken on board. We have done many additional experiments and have revised the manuscript accordingly. We respond to each comment and suggestion in detail below.

Reviewer Comments:

Reviewer #1

In this manuscript, Law, Krause and colleagues report the detailed characterisation of the role of NHSL1 in cell migration using knock-out, knock-down and overexpression. They identify two binding sites through which NHSL1 binds to the Scar/WAVE complex (SWC), specifically through the SH3 domain of the Abi subunit. They report that the small GTPase Rac1 binds to NHSL1 through two binding sites and that Rac1 regulates the NHSL1-SWC interaction. They also develop a new method to measure the Arp2/3 activity in vivo through the adaptation of an in vitro FRET pair. Overall their conclusion is that NHSL1 is a negative regulator of membrane protrusion and cell migration, through the complex regulation exerted by Rac1 on the SWC.

The authors study NHSL1 because another member of the family, NHS, had been reported to localise at the leading edge of migrating cells. Here they generalise to NHSL1, which appears to be a more ubiquitous isoform. They state several times that NHS was not previously involved in cell migration, which can be discussed. NHS was also previously reported to interact with the SWC and to regulate lamellipodium formation and cell spreading (Brooks et al., Hum Mol Genet 2010), even if proper migration assays were indeed not performed in this previous publication.

Indeed, as this referee correctly pointed out Brooks et al., 2010 did not investigate the function of NHS in cell migration. It is also correct that Brooks et al identified a role for NHS in lamellipodium formation and cell spreading. However, even though lamellipodium formation is the first step in mesenchymal cell migration, any effect on lamellipodia may or may not result in altered whole cell migration. Any effect on cell migration needs to be properly tested in cell migration assays. Brooks et al 2010 did not perform any such assays and this is what we are pointing out and thus the role of NHS in cell migration is not known.

We show in the work presented that NHSL1 interacts with the Scar/WAVE complex via the SH3 domain of Abi. Interestingly, the two Abi binding sites which mediate binding to NHSL1 (site 2+3; Fig. 5G) are highly conserved in the NHS family (PFAM PF15273 HHM logo, (Fig. S14C,D)) even though overall conservation between NHSL1 and NHS is just 30.3% and between NHSL1 and NHSL2 just 25.1% on the amino acid level (Fig. S14B) highlighting the importance of Scar/WAVE complex interaction at these particular sites. Furthermore, this suggests that NHS may also directly interact with the Scar/WAVE complex via the Abi SH3 domain mediated by the two novel Abi binding sites. We thus postulate that at least some functions such as regulation of cell migration may be conserved between the family members. Therefore, deregulated cell migration may play a role in the pathogenesis of Nance-Horan syndrome.

There is no doubt that the present manuscript goes beyond the previous characterization of NHS role, even if I have several critiques on the way experiments are performed in detail. The development of the FRET Arp2/3 biosensor is a major contribution to the field, which has overall relied on membrane protrusions and cell migration tracking as indirect read-out of Arp2/3 activity. To me, this paper is eligible to a Nature Comm publication, but only after a significant revision is performed.

1 The salient point of this NHS protein family is that the family members contain a Scar/WAVE homology domain in the N-terminus. Here we learn at the end of the manuscript that the whole

study was performed on a specific splice isoform of NHSL1 that does not contain the Scar/WAVE homology domain. I would like to see the relative abundance of the splice forms containing the Scar/WAVE homology domain or not in the couple of cell systems that are characterised in the manuscript.

We agree with this referee that the NHSL1 isoform containing the Scar homology domain (SHD) is very interesting. However, NHSL1 is alternatively spliced for exon 1 and there are several different variants, only one containing the SHD. Thus, the NHSL1 isoform with the SHD may have evolved a different function from the other isoforms. Indeed, our NHSL1 CRISPR knockout cell lines are knockout for all isoforms since we knocked-in a stop codon into exon 2, which is common to all isoforms. Therefore, in this first study we are asking the general question of what the function of all isoforms of NHSL1 are in mesenchymal cell migration and we found that NHSL1 negatively regulates cell migration. We have now added rescue experiments both with our wild type NHSL1 isoform without the SHD as well as this NHSL1 cDNA mutated for binding to the entire Scar/WAVE complex. Our results show that the observed negative regulatory effect of all isoforms of NHSL1 is not due to off target effects of the CRISPR sgRNA and in addition that it also depends on the interaction of NHSL1 with the entire Scar/WAVE complex via two Abi SH3 binding sites present in all isoforms of NHSL1. Therefore, we conclude that all isoforms of NHSL1 negatively regulate cell migration.

2 CRISPR mediated inactivation of NHSL1 is performed in B16F1. One clone seemingly corresponding to a biallelic KO and one clone seemingly corresponding to a heterozygous KO were selected and characterised further. These clones should be characterised at the genomic DNA level.

We have now validated the CRISPR knockout cell lines on the genomic level (Fig. S2A,B), by qPCR and western blot showing that the NHSL1 CRISPR 2 clone lacks all wild type alleles and displays an absence of mRNA suggesting that the CRISPR 2 line represents a full NHSL1 knockout. NHSL1 CRISPR 21 cells exhibit reduced mRNA and protein levels consistent with Cas9 induced indels only in some alleles (Fig. 2A,B; S3A) suggesting that the CRISPR 21 clone represents a heterozygous knockout.

The remaining protein levels, in both cases, should be evaluated by serial dilution of the control lysate.

We have utilised qPCR to quantify NHSL1 mRNA abundance in the CRISPR 2 and 21 cell lines and found that the CRISPR 2 cell line contained negligible amounts of NHSL1 mRNA (reduced by 98%) whereas in CRISPR 21 cells NHSL1 mRNA was only reduced by 19% compared to expression of the housekeeping gene GusB. Both values are an average from 3 independent biological repeats (new Fig. 2B).

These results are in line with our genomic characterisation that all wild type alleles of NHSL1 are absent in CRISPR 2 cells whereas the reduction in NHSL1 mRNA in CRISPR 21 cells is caused by Cas9 induced indels in only some alleles of NHSL1. Furthermore, this result is in line with the results of our western blot analysis of our knockout clones in which we see a much greater reduction in NHSL1 antibody signal in the lysates of our CRISPR 2 clone compared to the CRISPR 21 clone (new Fig. 2A; S3A).

Finally, even if I appreciate that care was taken to use the dual Cas9 nickase approach, a rescue of the KO clone by expression of NHSL1 would be the ultimate proof that the reported phenotype is specific to the KO.

We agree with this referee that the ultimate proof for absence of off target effects in CRISPR approaches is rescue by re-expression of the wild type cDNA and rescue of phenotypes. We have thus done rescue experiments, and in addition, have compared the re-expression of wild type NHSL1 cDNA with re-expression of our NHSL1 cDNA which cannot interact with the Scar/WAVE complex (see below). We have performed these experiments and analysed these cells in random migration

assays (new Fig. 6), lamellipodial dynamics assays (new Fig. 10) and newly added F-actin retrograde flow assays (new Fig. 9) and thus greatly extended our analysis.

3 After extensive mapping, a mutant form of NHSL1, which does no longer bind to the SWC, has been isolated. The difference of phenotype between the mutant form of NHSL1 and WT NHSL1, expressed in WT cells, appears moderate (Fig.8D, 8E). The authors should rescue the phenotype of the KO B16F1 cell line with WT and mutant NHSL1. This would validate the specific phenotype of KO cells and should reveal that interaction with SWC is critical for NHSL1, if mutant NHSL1 indeed does not rescue.

We agree with this referee and referee 3 that this would be informative and thus have performed these experiments and analysed these cells in random cell migration assays (new Fig. 6) and found that the increase in cell migration speed can be reduced and the phenotype is thus rescued by re-expression of wild type NHSL1 but not by the NHSL1 mutant cDNA which cannot interact with the Scar/WAVE complex.

The implicit assumption that the Scar/WAVE homology domain is not a major determinant of the NHSL1 role in cell migration should also be tested in the rescue system.

We apologise if our explanation of the putative role of the SHD for NHSL1 function in the previous manuscript has not been clear. Since the human genome contains several alternative first exons for NHSL1 and only one of these harbours the SHD, this isoform may have a more specific function compared to other isoform of NHSL1. In addition, we know now from this study that all isoforms of NHSL1 have the two Abi binding sites and through these they interact with the entire Scar/WAVE complex. The isoform with the SHD domain may in addition have the ability to interact with components of the Scar/WAVE complex through the SHD domain, thus complicating the analysis. Therefore, we can only now analyse the NHSL1 isoform with the SHD domain after having carefully analysed the NHSL1 isoform without the SHD domain. We will thus carefully analyse the role of the NHSL1 isoform with the SHD domain in a separate project.

4 The authors could go much further in their analysis of movies, which are so far used only to demonstrate colocalisation of GFP-NHSL1 with Lamellipodin and SWC subunits. Since NHSL1 negatively regulates protrusions, contrary to the positive regulators, SWC and lamellipodin, their colocalisation should be analysed over time in relation with protrusions and retractions. Does their cross-correlation display a characteristic delay ? According to the model, we should expect that NHSL1 arrival/enrichment should predict a retraction of the membrane. Such a result would give an undeniable confirmation of current findings.

At the current level of analysis, we observed that NHSL1 only localised to protruding lamellipodia where it co-localised with components of the Scar/WAVE complex and also Lpd. We agree with this referee that an in-depth cross correlation would provide additional insights on how NHSL1 may function compared to Lpd. However, this would require much higher frame rates as well as custom written image analysis software.

An alternative hypothesis to the mechanism proposed by this referee that Lpd and NHSL1 function sequentially during lamellipodium protrusion and retraction, respectively is that both could localise to protrusions at the same time and function to control lamellipodia protrusions in a novel incoherent feedforward loop as highlighted in our discussion (amended from the previous version in order not to overstate):

“Similar to NHSL1, Lpd functions downstream of active Rac but in contrast, Lpd promotes cell migration speed and persistence^{6,7}. Both Lpd and NHSL1 appear to co-localise at the very edge of lamellipodia during protrusion (Fig. S1, video S3). We therefore propose that NHSL1 together with Lpd forms a novel incoherent feedforward loop at the level of the Scar/WAVE complex downstream of active Rac at the leading edge of cells in which Lpd promotes whereas NHSL1 inhibits cell migration via Scar/WAVE to enable cell steering during

migration.”

Therefore, we focussed our efforts on experiments that strengthen the core of our arguments in this manuscript and may revisit this more detailed analysis in future studies.

5 I was shocked by the superficiality of the Rac1 part, which fully relied on the transfection of GTPase-deficient, dominant active form. This was particularly surprising given that B16F1 cells must already have a high basal level of active Rac1, given their extraordinary lamellipodial activity. Rac1 activity should be inhibited for a comparison. In all these experiments, Rac activity should have been modulated using many ways. DA Rac must be compared to WT and DN Rac1, the nucleotide free form T17N that titrates GEFs. It is a bit old-fashioned but that works. In addition, there are Rac chemical inhibitors. Rac1 can also be down-regulated using siRNAs. Some toxins activate or inactivate GTPases and Rac1 in particular. Rac1 KO MEFs have been reported. The authors may choose the most appropriate ways of modulating Rac activity, but the manuscript cannot simply rely on the overexpression of active Rac throughout.

6 The main finding concerning Rac1, which is that it modulates the NHSL1-SWC interaction, is surprisingly only documented when all proteins are overexpressed (Fig.8B). This cannot be justified, given that the interaction between endogenous NHSL1 and endogenous SWC can be detected (Fig.4F, 4G). A major question, which should receive a clear answer, is whether the NHSL1-SWC interaction requires Rac1 activity.

We agree with this referee and referee 3 that the regulation of interaction between NHSL1 and the Scar/WAVE complex by active Rac needs to be more thoroughly investigated. However, we believe that it is more important to strengthen the core arguments in this manuscript that NHSL1 acts as a negative regulator of Scar/WAVE-Arp2/3 complex activity, lamellipodial stability and cell migration, as requested by all referees. Therefore, we will revisit the detailed analysis of the regulation of the interaction between NHSL1 and the Scar/WAVE complex by active Rac later and this will form a large part of a future study.

7 This experiment alone where both Rac and NHSL1 are overexpressed would not be sufficient to convince that Rac1 binds to NHSL1, even if appropriately controlled. It should be complemented by a GST pull-down using GST-Rac1 Q61L (remaining GTP bound) compared to GST-Rac1 WT (mainly associated with GDP, because of GTP hydrolysis), or using GST-Rac1 WT loaded with GDP or GTPgammaS. The SWC can provide a useful positive control for GTP-Rac specific binding.

8 Is the binding of Rac1 to NHSL1 direct? The question is especially important, since no established Rac1 binding site has been detected. Given that the binding sites are mapped, this experiment is easy to perform. One binding site appears much stronger than the other one, which is at the limit of detection. This point should be mentioned in the text.

We agree with this referee that the interaction between active Rac and NHSL1 needs to be more thoroughly investigated. We have now added a control experiment (also in response to point 11 of this referee) to show specificity of the co-precipitation between active Rac and NHSL1: We have verified the interaction in a reciprocal experiment in which Myc-NHSL1 was co-expressed with EGFP-(DA)-Rac1-Q61L or EGFP only as a control in HEK cells. EGFP-Rac1-Q61L or EGFP was pulled down with GFP-trap beads and the interaction with Myc-NHSL1 was evaluated by western blot against Myc (Fig. S6B). Again, we observed a specific interaction between dominant active Rac and NHSL1 suggesting that NHSL1 is a novel binding partner of active Rac. With the support of this additional data, we believe that we now have sufficient data suggesting that active Rac and NHSL1 may be in complex in cells to include this data into the current manuscript.

We have now stated in our discussion that: “Active Rac binds to NHSL1 via two sites and thus Rac may recruit NHSL1 to the leading edge of cells (new Fig.3; S6,S7). Whether this interaction is direct or not needs to be investigated in future studies.”

9 Single cell migration assays are central to this paper. They should not be characterized by the best parameters only. I believe that MSD is missing from Fig.2. Then to characterise the mutant form that does no longer bind to the SWC, the full set should be displayed: trajectories, MSD, Speed and directional persistence.

We agree with this referee that it is essential to properly characterise parameters of single cell migration analysis which includes cell migration speed and persistence. We had already in our first submission of this manuscript used a method originally developed by Graham Dunn and published in our Law et al., 2013 JCB paper (DOI: 10.1083/jcb.201304051) which allows accurate analysis of cell migration persistence and overcomes the pitfalls of the traditional method (influence by different track lengths and speed) associated with measurements of directionality ratio at the endpoint of the movie. The Dunn method quantifies the directionality ratio over short intervals of the movie and then averages these measurements over the entire length of the movie. We realise now that it was not obvious how we measured persistence from just reading the results section, though it was described in the methods. We have thus added a more detailed description in the results section and an explanatory schematic diagram to new Fig. 2C. In addition, we added MSD (new Fig. 2F) (which is a dependent on speed and persistence), directionality ratio over time and direction autocorrelation (which is independent of speed) (new Fig. S4A,B).

10 The wound healing assay is not very convincing (Fig.S2, video S3). The NHSL1 shRNA A condition displays borders, which are not sharp. Many cells are rolled up. The wound appears less wide in this shRNA A condition than in the other conditions. I did insert that can be removed offer useful alternative to scratches.

We very carefully conducted this experiment and performed all necessary controls: The presented result is the mean of four independent biological repeats. In addition, we always stopped this analysis before cells of opposing fronts touched each other to avoid any effects from contact inhibition of locomotion. We also added mitomycin C (see materials and methods) before each experiment to prevent any effect of cell proliferation on this admittedly complex cell migration assay. In addition, slight variations in the width of scratches do not skew the data since we quantified the percentage change in area of the scratch wound over time and thus our analysis ensures this is independent of initial scratch size. Out of caution and to address the concerns of this referee, we have removed the NHSL1 shRNA A condition and renamed the remaining two NHSL1 specific shRNA's as shRNA A and B (formerly shRNA B and C). Furthermore, as can be seen in the supplemental movie (Video S4), the borders are clearly visible underneath the rolled-up cells when they emerge and therefore we believe this can be accurately quantified (new Fig. S5 and video S4).

11 Fig.6F: Myc tagged Rac1 is cotransfected with GFP-NHSL1 or GFP. Myc is immunoprecipitated, WB anti-GFP. Of course, NHSL1 cannot coprecipitate if it is not expressed ! GFP alone is the appropriate control only if GFP is precipitated first, followed by a WB anti-Myc. As presented here, the appropriate control is GFP-NHSL1, with no Myc tagged protein or an irrelevant Myc tagged protein.

We agree with this referee that this needs to be better controlled: We have verified the interaction in a reciprocal experiment in which Myc-NHSL1 was co-expressed with EGFP-(DA)-Rac1-Q61L or EGFP only as control in HEK cells. EGFP-Rac1-Q61L or EGFP was pulled down with GFP-trap beads and the interaction with Myc-NHSL1 was evaluated by western blot against Myc (new Fig. S6B). Again, we observed a specific interaction between dominant active Rac and NHSL1 suggesting that NHSL1 is a novel binding partner of active Rac.

12 The logic of several colPs appears strange to me. Fig4D and Fig.5AB: GFP-NHSL1 is overexpressed, but endogenous NHSL1 is precipitated with anti-NHSL1 antibodies... GFP trap would be more logical in these cases.

We agree with this referee that GFP-trap experiments have advantages, but we believe that it is for this experiment equally acceptable to precipitate EGFP-NHSL1 (including endogenous NHSL1) with our NHSL1 antibodies since in these experiments we expressed wild type (and not mutated) EGFP-NHSL1 and thus it does not affect the outcome of the experiment if also endogenous NHSL1 is precipitated alongside the overexpressed EGFP-NHSL1.

Similarly, FigS3C is not convincing. Myc-NHSL1 is overexpressed, endogenous NHSL1 is immunoprecipitated. Strangely, no Myc tagged site mutant is detected in the IP ! What can we conclude from such an experiment ?

We are sorry that our original western blots were not clear and agree that the experiment in the Fig S3C in the old manuscript was not optimal in which we explored the interaction between three different Myc-NHSL1 constructs mutated in single Abi-SH3 binding sites and the tagged Scar/WAVE complex. We have therefore replaced this experiment with another more detailed analysis showing that only the loss of the Abi-SH3 domain binding sites 2 and 3 in NHSL1 together results in absence of interaction between NHSL1 and Abi, which is now clearly shown in new Fig. 5G.

13 Fig.5D is very important, since it defines the NHSL1 mutant that does no longer bind to the SWC. The lack of coIP with SWC is not convincing, because less of the NHSL1 mutants that should not bind to the SWC than controls are precipitated. Overall the GFP immunoprecipitations appear very inefficient, because it does not concentrate the proteins seen in the lysates. Again I don't see why the SWC has to be overexpressed in this case.

We agree with this referee that the EGFP-NHSL1 pulldown bands appear weak compared to the EGFP-NHSL1 lysate lanes. The reason is that efficient blotting of large size proteins above 200 KDa is not trivial. Since then we have improved our blotting technique to obtain more consistent results and replaced the pulldown experiment (old Fig 5D) with a new Fig 5G and followed the advice of this referee and referee 3 and only tested absence of interaction between NHSL1 and Abi.

14 Fig8B. It is required to show the amount of NHSL1 that is immunoprecipitated, in order to compare the amount of SWC coprecipitated in this experiment.

We agree with this referee and referee 3 that the regulation of interaction between NHSL1 and the Scar/WAVE complex by active Rac needs to be more thoroughly investigated. However, we believe that it is more important to strengthen the core arguments in this manuscript that NHSL1 acts as a negative regulator of Scar/WAVE-Arp2/3 complex activity, lamellipodial stability and cell migration, as requested by all referees. Therefore, we will revisit the detailed analysis of the regulation of the interaction between NHSL1 and the Scar/WAVE complex by active Rac later and this will form a large part of a future study.

15 Fig.10EF is really not clear. What is the unit of the x-axis ? Why this strange normalisation of each bin in the control case ?

Since the presentation of the data in old Fig 10 EF appears confusing and conveys a similar message as shown in Fig. 10 A-D (old and new manuscript) we have replaced this data with new data: In addition to cell migration assays we have also performed lamellipodia dynamics analysis comparing wild type with NHSL1 CRISPR 2 cells re-expressing wild type or NHSL1 cDNA mutated in the Scar/WAVE binding sites. This data shows that only the NHSL1 wild type but not the NHSL1 cDNA mutated in the Scar/WAVE binding sites can rescue the effect of NHSL1 on lamellipodium stability (new Fig. 10 F).

16 The Palmitoylation experiment appear superfluous to me. I don't know why it is presented here. It is not well justified. I also doubt that the simple experiment reported here specifically reveals a palmitoyl modification, i.e. an acyl modification of a certain length. Then such an experiment immediately raises many questions that his manuscript cannot address in addition to all the rest.

What is the residue modified ? Is this modification important for the activity ? The authors can keep this point for future studies.

Whilst this aspect is interesting, we agree with this referee that the regulation of NHSL1 by S-palmitoylation does not fit well with our current story. Thus, we removed the data from this manuscript and will revisit the analysis of the regulation of NHSL1 by S-palmitoylation in future studies.

Miscellaneous

17 Typo: occurrence -> occurrence in all FRET figures (Fig.9, Fig.10, Fig.S4)

We would like to thank the referee for spotting this typo which we have corrected.

18 FigS5: indicate on the conservation logo diagrams the residues that were mutated

We have now included the residues that were mutated underneath the conservation logo diagram in the new Fig. S14C,D.

19 The nomenclature for the mouse gene would be *Nhsl1* and not NHSL1, which is the human form. Then for the protein, it has to be clearly defined. It can be useful to state in the results section that the constructs were derived from the mouse protein.

In our manuscript we have used the human cDNA of NHSL1 throughout and made this more explicit in the materials and methods.

+++++

Reviewer #2

In this manuscript, Law et al. demonstrate that NHSL1 is a novel direct binding partner of the Scar/WAVE complex, which localizes to the edge of lamellipodia. Moreover, NHSL1, which binds to the active Rac, negatively regulates cell migration speed and persistence by restricting Arp2/3 activity. Unfortunately, this is not a well-written manuscript, and the way that data are presented (i.e., organization of figures) is at best subpar. As described below in detail, certain experimental controls are missing; individual bands instead of whole blots are presented; and the scientific rigor is not thorough in the present form of this manuscript. This reviewer also makes key suggestions, which could potentially bring the manuscript to publication quality.

Major Comments:

1. The authors claim that NHSL1 localizes to the lamellipodia edge and inhibits Arp2/3. However, based on FLIM FRET Lifetime images (Fig. 9D), Arp2/3 is shown to be activated inside the cell upon NHSL1 knockdown. How do the authors interpret this finding?

We have shown that NHSL1 localises to the leading edge of cells by expressing EGFP-tagged NHSL1 as well as endogenous NHSL1 protein using two different antibodies (monoclonal as well as polyclonal). We have added new data indicating that the observed localisation of the EGFP tagged NHSL1 is not due to space filling at potential ruffles at the leading edge (new Fig S1D; video S2). In addition, both immunofluorescence analysis as well as EGFP tagged live cell imaging suggests that NHSL1 also localises to vesicles. FRET-FLIM is currently a very slow technique with each image taking 5 minutes to capture thus precluding fast live cell imaging. In this FRET-FLIM technique the activity of a biosensor like our Arp2/3 biosensor is measured as an average over the entire cell. We agree with this referee that looking at the lifetime images, it appears that Arp2/3 is activated on vesicles in

addition to the leading edge of cells (new Fig. 8D). We attempted to quantify the activation of the Arp2/3 complex using our biosensor at the leading edge of the NHSL1 CRISPR cell lines, though the accuracy of this data is currently limited by the resolution of our FRET-FLIM microscope. Nevertheless, we observed a trend for an increase of Arp2/3 activity in the NHSL1 CRISPR 2 and 21 cell lines compared to control which did not reach significance (new Fig. S13H). Thus, this non-significant trend agrees with a potential role for NHSL1 to inhibit Scar/WAVE-Arp2/3 activity at the leading edge. In addition, NHSL1 may inhibit Scar/WAVE-Arp2/3 activity at vesicles. For good reasons (technical limitations) most labs using FRET-FLIM do not attempt to measure activity at sub-cellular resolutions and thus a more definitive answer awaits technological advances in FRET-FLIM.

Further evidence for a role of NHSL1 at the leading edge comes from our new data that Arp2/3 and F-actin intensity at the leading edge is increased in the NHSL1 CRISPR KO cells (new Fig. 7D,E). In addition, we have presented data showing that overexpression of wild-type NHSL1 but not overexpression of NHSL1 cDNA, which cannot interact with the Scar/WAVE complex, reduces lifeAct-EGFP intensity at the leading edge (new Fig. 7F). Taken together, this suggest that NHSL1 may function to inhibit Arp2/3 mediated actin nucleation at the leading edge of cells.

2. The authors indicate that NHSL1 inhibits Arp2/3 by binding with the SCAR/WAVE complex. However, authors do not demonstrate this directly. At minimum, the authors should show that NHSL1 regulates Arp2/3 activity via SCAR/WAVE by comparing Arp2/3 FLIM FRET results with NHSL1 knockdown to FLIM FRET results with SCAR/WAVE inhibition and simultaneous NHSL1/SCAR/WAVE inhibition.

Indeed, a direct proof that NHSL1 inhibits the Scar/WAVE complex which then causes reduced Arp2/3 complex activity would be desirable. However, we don't believe that the experiments suggested by this referee would provide this proof: As far as we know there is no drug available targeting Scar/WAVE complex activity and therefore this cannot be achieved acutely but only by knockdown of components of the Scar/WAVE complex. However, any knockdown of the Scar/WAVE complex would be accompanied by an increase in N-WASP induced Arp2/3 activity as a compensatory mechanism (see Tang et al. 2013 Curr Biol: (doi: 10.1016/j.cub.2012.11.059)). Therefore, the difference in Arp2/3 activity between knockdown of the Scar/WAVE complex alone, compared to knockout of NHSL1 alone and simultaneous knockout of NHSL1 and knockdown of the Scar/WAVE complex would be confounded by the compensatory mechanisms and would therefore likely not be informative.

However, to answer this question we have rescued the NHSL1 CRISPR 2 knockout cells with either wild-type NHSL1 or NHSL1 mutated in the Scar/WAVE binding sites. We have analysed these in random cell migration assays and lamellipodia dynamics assays. The key results from these experiments show that we can rescue the increased random cell migration speed only with the wild-type but not the NHSL1 Scar/WAVE binding mutant. Similarly, the increased lamellipodial stability observed upon NHSL1 knockout can only be rescued with the wild type but not the NHSL1 Scar/WAVE binding mutant. Taken together, this suggest that NHSL1 may functions via the Scar/WAVE complex in reducing cell migration speed by decreasing lamellipodial stability. The lamellipodial Arp2/3 and F-actin intensity quantification data (new Fig. 7D-F) indicates that NHSL1 may do this by decreasing branched F-actin polymerisation in the lamellipodium via binding to Scar/WAVE.

3. The authors should also use NHSL1 SW-mut construct for lamellipodia quantification assays (Fig. 10).

We have now rescued the NHSL1 CRISPR 2 knockout cells with either wild type NHSL1 or NHSL1 mutated in the Scar/WAVE binding sites and analysed these in lamellipodia dynamics assays. These experiments show that the increased lamellipodial stability observed upon NHSL1 knockout can only be rescued with the wild-type but not the NHSL1 Scar/WAVE binding mutant suggesting

that NHSL1 may function via the Scar/WAVE complex in decreasing lamellipodial stability (new Fig. 10F).

4. The authors should quantify additional parameters relevant to lamellipodia dynamics such as retrograde actin flow.

We have now rescued the NHSL1 CRISPR 2 knockout cells with either wild-type NHSL1 or NHSL1 mutated in the Scar/WAVE binding sites. We have analysed these in actin retrograde flow assays and quantified the movies by particle image velocimetry. This analysis revealed a significant reduction of F-actin retrograde flow and assembly rate in the NHSL1 CRISPR 2 lamellipodia compared to wild-type control lamellipodia (new Fig. 9A-C). Re-expression of wild-type NHSL1 (EGFP-NHSL1 WT) or the NHSL1 Scar/WAVE complex binding mutant (EGFP-NHSL1 SW mut) appeared to increase flow speed and the actin assembly rate compared to the NHSL1 CRISPR knockout lamellipodia but this did not reach significance (new Fig. 9A-C).

5. The far Western blot experiments (Fig. 5C) should be verified using in vitro pull-down assays. Also, in Fig. 5C, the authors should blot for GST to verify the presence of each of the 8 fragments.

We do not believe that in vitro pulldown assays would provide any additional insights since they in effect provide the same information as the far western blot experiments: both provide evidence of a direct interaction between NHSL1 and the Abi SH3 domain. The more important experiment here is the GFP-trap pulldown experiment of the full length EGFP-NHSL1 cDNA mutated in the Abi-SH3 domain binding sites to verify that the interaction between NHSL1 and Abi is mediated via these SH3 domain binding sites in NHSL1 (see new Fig. 5G). Since the GST fragments have a relatively high abundance on the blots, clean detection by western blot is not straightforward. Therefore, we have added a Coomassie gel showing equivalent amounts of each fragment and GST (see new Fig. S9B) used in the Far Western experiment (see new Fig. S9A).

6. The scientific rigor of this study is at best problematic. For instance, in Figs. 8D and 8E, are the differences between NHSL1-WT and NHSL1-SW mut statistically different?

In the revised manuscript we have now added exact p values and provided the F-statistics for ANOVA. We are also providing the raw data in the source data file allowing the referee to examine the validity of our statistical results herself/himself.

7. Key experiments should be verified with multiple cell lines (Figs. 2, 8D-E, 9). Along these lines, the authors should show data with several CRISPR cell lines.

We have verified the key result that NHSL1 is a negative regulator of cell migration using the MCF10A cell line (now new Fig. S5). We also show for all experiments data from two CRISPR cell lines (CRISPR 2 and CRISPR 21). For this revision, we have added the rescue of our CRISPR 2 cell line with wild type NHSL1 or our NHSL1 mutant cDNA which cannot interact with the Scar/WAVE complex. The observed rescue by wild-type but not mutant NHSL1 further supports our conclusions that NHSL1 is indeed a negative regulator of cell migration and functions via the Scar/WAVE complex.

8. Migration data are provided using old-fashioned 2D assays. The authors should verify their 2D findings using different ECMs (e.g., collagen type I), and also perform 3D migration assays.

For this revision, we have added the rescue of our NHSL1 CRISPR 2 cell line with wild type NHSL1 and our NHSL1 mutant cDNA which cannot interact with the Scar/WAVE complex. These 2D cell migration assays were carried out with cells plated on laminin in addition to the original experiments where cells were plated on fibronectin. We obtained the same results on both substrates: that NHSL1 negatively regulates cell migration (see new Fig. 2, 6, S10, S11).

2D migration assays are well established assays which provide important information. These have been widely used as the only cell migration readout in many current studies which uncovered

fundamental mechanisms of cell migration. For examples see

Maiuri et al., 2015, Cell, DOI: 10.1016/j.cell.2015.01.056;
Mueller et al., 2017 Cell, DOI: 10.1016/j.cell.2017.07.051,
Kage et al., 2017, Nature Comm. DOI: 10.1038/ncomms14832,
Nordenfelt et al., Nature Comm, DOI: 10.1038/s41467-017-01848-y
Azoitei et al., 2019, JCB, DOI: 10.1083/jcb.201812073
Kumari et al., 2020, Curr Biology; DOI: /10.1016/j.cub.2019.12.049

Nevertheless, we agree with this referee that 3D cell migration experiments would be informative, but these are beyond the scope of the current study.

9. What is the role of NHSL1 *in vivo*?

As we have pointed out in our discussion: The zebrafish orthologue of NHSL1 had been identified in a genetic screen for defects in a specific developmental migration of facial branchiomotor neurons suggesting a positive role of NHSL1 in neuronal migration³². However, we found that NHSL1 negatively regulates 2D cell migration (new Fig. 2,6). This is interesting and we hypothesise that this reflects different modes of migration since zebrafish NHSL1 functions together with the planar cell polarity components Scrib and Vangl cell autonomously in facial branchiomotor neurons. This specific migration mode involves the interactions between the neurons and their planar-polarised environment of the surrounding tissue³².

32. Walsh, G. S., Grant, P. K., Morgan, J. a & Moens, C. B. Planar polarity pathway and Nance-Horan syndrome-like 1b have essential cell-autonomous functions in neuronal migration. *Development* **138**, 3033–3042 (2011).

Depending on the migratory cells studied *in vivo*, NHSL1 may have different roles depending on whether the cells migrate collectively or as single cells, whether the cells migrate as mesenchymal or epithelial cells or whether they migrate in mesenchymal or amoeboid modes. Therefore, we decided to first ask fundamental questions on how NHSL1 affects mesenchymal migration in a simplified 2D model of cell migration, before we can in future studies address the different roles that NHSL1 may play *in vivo* in different developmental migration processes and during homeostasis or disease.

Minor Comments:

1. In Fig. 1, the authors should also show that endogenous NHSL1 is localized at the very edge of lamellipodia (which is actually shown in Fig. 4A for B16F1 mouse melanoma cells).

As the referee pointed out, we already show the localisation of endogenous NHSL1 at the very edge of lamellipodia in old and new Fig 4.

2. The entire blot of Figs. 1D and 1E should be provided as Supplemental data.

We have now provided blots equivalent to old Fig. 1D and 1E (new Fig. 1B-D) as full blots in supplemental Fig. S1A-C.

3. Quantification of Western blots in Fig. 2A should be provided as a Supplemental Figure.

We have utilised qPCR to quantify NHSL1 mRNA abundance in the CRISPR 2 and 21 cell lines and found that the CRISPR 2 cell line contained negligible amounts of NHSL1 mRNA (reduced by 98%) whereas in CRISPR 21 NHSL1 mRNA was only reduced by 19% compared to expression of the housekeeping gene GusB. Both values are an average from 3 independent biological repeats. We have added this new data as new Fig. 2B.

4. Fig. 3 should be merged with Figures 4 and 5.

If we would merge Fig. 3 with figures 4 and 5 the panels would become so small that readers would not be able to see the immunofluorescence data or the western blots properly. In the revised version we have extensively rearranged the figures to make the best use of space.

5. The authors should show the entire blot of Fig. 4D.

Old Fig. 4D, new Fig. 5A: The full anti-Myc blot is shown in the new Fig. S8A and the full anti-EGFP blot is now shown in the new Fig. S8B.

6. The authors should present the entire blot of Fig. 6F.

Old Fig. 6F, new Fig. 3G: The full blots are now shown in new Fig S6A.

7. Figs 6 and 7 should be combined.

If we would combine both figures it would be impossible to see the details of the blots and images. We have now moved the old Fig. 7 into supplemental as new Fig. S7.

8. In Figures 6G, 7B, the amount of NHSL1 fragments loaded is unequal. The authors should repeat these experiments to ensure closer to equal loading of each fragment.

The reason is unequal expression or stability of fragments and not unequal loading. This is consistent through all repeats of this experiment. Therefore, we took expression levels into account when drawing conclusions on interaction strength:

In old Fig. 6G and new Fig. 3H the fragment 2 is more weakly expressed yet robustly pulled down whereas the fragment 3 is more strongly expressed but weakly pulled down by dominant active Rac suggesting that fragment 2 is binding more strongly than fragment 3 to dominant active Rac. Since this conclusion takes expression levels of fragments into account we believe that the conclusions are valid despite differences in expression levels.

In old Fig. 7B and new Fig. S7B fragments 2 and 9 are more weakly expressed than all other fragments. Taking expression levels into account fragments 1, 6, and 9 are pulled down to roughly equal extent by dominant active Rac. Since this conclusion takes expression levels of fragments into account we believe that the conclusions are valid despite differences in expression levels.

9. In the introduction, the authors state that the SCAR/WAVE complex is autoinhibited. However, to the best of our knowledge, the WASP complex is autoinhibited but the SCAR/WAVE complex is not. The authors should provide specific references for this claim (no citation currently).

It has been documented in many publications that the Scar/WAVE complex is autoinhibited, see:

Eden et al., Nature 2002; doi: [10.1038/nature00859](https://doi.org/10.1038/nature00859)

Derivery et al., Cell Motility and the Cytoskeleton, 2009, doi: [10.1002/cm.20342](https://doi.org/10.1002/cm.20342)

Ismail et al., 2009, Nature Structural & Molecular Biology; doi: [nsmb.1587](https://doi.org/10.1038/nsmb.1587) [pii] [10.1038/nsmb.1587](https://doi.org/10.1038/nsmb.1587)

Lebensohn and Kirschner, Mol Cell 2009, doi: [10.1016/j.molcel.2009.10.024](https://doi.org/10.1016/j.molcel.2009.10.024)

Chen et al., 2010 Nature; doi: [10.1038/nature09623](https://doi.org/10.1038/nature09623)

Chen et al., 2017 elife; doi: [10.7554/eLife.29795](https://doi.org/10.7554/eLife.29795)

In the introduction we have cited our review: (Krause and Gautreau, Nature Reviews MCB 2014; DOI: [10.1038/nrm3861](https://doi.org/10.1038/nrm3861)) and now added a review by Giorgio Scita and colleagues (Bisi et al., 2013 Curr. Opin. Cell Biol; doi: [10.1016/j.ceb.2013.04.001](https://doi.org/10.1016/j.ceb.2013.04.001)). Both of these reviews summarise this wealth of knowledge about the autoinhibition of the Scar/WAVE complex.

+++++

Reviewer #3

Nance-Horan Syndrome (NHS) is an X-linked developmental disorder with symptoms of congenital cataract, dental anomalies, characteristic dysmorphic features and developmental delay. Nance-Horan protein (NHS) has been linked to regulate actin remodeling and cell morphology. The authors in this manuscript characterize another member of this family of proteins, NHSL1 and show that it serves as a negative regulator of cell migration. The authors also show that depletion of this protein induces higher migration speed and activation of the Arp complex thereby increasing stability of lamellipodia. Though the study is interesting, and the authors do elucidate a novel regulator of the wave complex and Arp activity, there are some inconsistencies that should be addressed. Broadly the inconsistencies are on the following front:

1. Reading though the texts, it seems many of the results are over-interpreted and the authors could be a bit milder in their approach.

We apologise if our writing was overly interpreted at times. We have now toned down our conclusions appropriately.

2. There are several technical problems specially in the biochemical results which will be pointed out in my comments.

We have addressed all of these technical problems and have given specific answers in the detailed comments (see below).

3. The figures can be shortened and clubbed together to make the data a lot tighter.

In this revised manuscript we have extensively rearranged the figures to make the best use of space.

4. Some studies lack proper controls.

We apologise that we did not perform all necessary controls. We have now added all additionally requested controls (see below).

Major Comments:

1. Since NHS has been a well characterized member of this family and has been shown to have similar localization, a thorough comparison needs to be done for NHS and NHSL1 in figure 1 based on its domains. This will reveal how similar or dis-similar these two members are.

Only five cell biology papers have been published about NHS and so to date there is very limited information about the structure and function of NHS. Therefore, apart from the SHD, which is only found in one specific isoforms, there are no known domains present in the NHS family. As pointed out in the discussion, the conservation between NHS and NHSL1 is low (30.3%) at the amino acid level (new Fig. S14B).

2. Throughout the western blot figures (Fig 1D, 2A, S2C), the authors have not provided any loading controls. These need to be provided.

We apologise for omitting these and we have now provided western blots with loading controls to replace old Fig. 1D,E; 2A and these are now new Fig. 1B-D; 2A; S1A-C;S3A.

The authors need to state whether the images in Fig 1D are from the same or different blots. If they are from different blots, they need to have different MW marker lanes.

We have replaced the blot in old Fig. 1D with new separate western blots for lysates from B16-F1 (new Fig. 1B) or MCF10A (new Fig. 1C) with different molecular weight marker lanes and appropriate loading controls (please find the full blots in new Fig. S1A,B).

Also, for fig 1D, it will be a good idea to characterize the antibody through specific KD of NHSL1 and evaluating NHSL1 along with other members of the family. Since the authors also use the antibody for IF in later figures, it will be good to show how specific the antibody is in IF staining.

We have characterised our NHSL1 antibody on lysates of NHSL1 CRISPR KO cell lysates compared to control wild-type B16F1 lysates (see new Fig. 2A and full blots in Fig. S3A). Several bands of lower molecular weight can be seen in addition to a 260 kDa band. These may represent smaller isoforms of NHSL1 or degradation products of the 260 kDa band. However, all additional bands of lower molecular weight are equally reduced in the NHSL1 CRISPR 2 KO cell line (see full blots in new Fig. S3A) suggesting that our antibody is specific.

The authors do need to provide the full blots for Fig 1D to show how specific the antibody is.

The replacement blot for old Fig. 1D are now the blots in new Fig. 1B,C. The full western blots of new Fig. 1B,C are provided in Fig. S1A,B.

Another inconsistency with the manuscript is the placement of the molecular weight markers. While in fig 1D, 2A, 4D, the endogenous NHSL1 line up with the 250 Kd marker, Fig 6F has EGFP tagged NHSL1 running lower than 250 KD.

We would like to apologise and thank this referee for pointing out this mistake. Indeed, endogenous NHSL1 has an apparent molecular weight of 260 kDa and when tagged with EGFP it has an apparent molecular weight of 290 kDa. We have double checked all molecular weight makers in all figures and rectified the mistake.

In another instance while endogenous Abi has been shown to run at 55 Kd (Fig S3B), EGFP tagged Abi has also been shown to run at 55 Kd (Fig S3A) which should run at around 80 Kd. The authors need to clarify these technical issues.

We would like to apologise and thank this referee for pointing out this mistake. We have rectified the mistake and the molecular weight marker in new Fig. S8D (old Fig. S3A) indicates that EGFP-Abi1 has an apparent molecular weight of 80kDa.

3. As mentioned earlier the authors need to provide loading controls for Fig 2A.

We have now provided blots with loading controls to replace old Fig. 2A, this is now new Fig. 2A and the full blots are shown in new Fig. S3A.

Figure 2B: The average random migration speed for B16-F1 cells determined by the authors was 0.3 um/min. This appears to be very slow compared to other reports for this cell line (Kage F et al., Nat Commun. 2017 Mar 22; 8:14832) One possible explanation for that could be that the authors have performed these random migration assays on fibronectin, as opposed to all other experiments in which they have been using laminin-coating. What is the reason for that? I would suggest repeating random migration assays using laminin as substrate.

The reason that the observed random migration speed in Fig. 2 of the B16-F1 cells is relatively slow is indeed due to the fibronectin coating. We have now repeated the random migration experiments on laminin and indeed observe a faster migration speed compared to fibronectin (new Fig. 6B) which is similar to other reports for this cell line on laminin (Kage F et al., Nat Commun. 2017 Mar 22; 8:14832).

Since the authors show a significant effect of NHSL1 KO in cell migration speeds, the authors need to characterize the KO cells better. To this end the authors can report the following parameters:

a. Perform Rhodamine-phalloidin staining to characterize lamellipodia in KO cells.

We agree with this referee and referee 2 that this would be informative and have now quantified Alexa488-phalloidin labelled F-actin intensity in the lamellipodium of control and

NHSL1 CRISPR KO cells and observed an increase in intensity per area in the NHSL1 CRISPR 2 and 21 cells (new Fig. 7E).

b. Report initial Lamellipodial width in KO cells vs WT cells.

We have performed this analysis and report in the revised manuscript that both lamellipodia width and number of microspikes per length of lamellipodium are unchanged in WT vs NHSL1 KO cells (new Fig. 7A,C; S11E).

c. Evaluate the levels of WAVE complex proteins and Arp2/3 in the KO cells.

As a control we performed immunofluorescence analysis on wild type B16-F1 cells or the NHSL1 CRISPR 2 and 21 cells and quantified whole cell Arp2/3 (ARPC2) intensity. We found that total Arp2/3 levels were not increased (new Fig. S13G) and this excludes the possibility that the observed increase in Arp2/3 activity by FLIM is due to increased Arp2/3 levels in the NHSL1 CRISPR cells.

d. Evaluate the levels of NHS in the NHSL1 KO cells.

Currently there are no characterized commercial antibodies against NHS available and thus we are unable to evaluate the level of NHS in NHSL1 KO cells. We cannot exclude a potential compensation by higher expression levels of NHS in the NHSL1 KO cells. However, since we are observing a significant phenotype in the NHSL1 KO cells, we can attribute this phenotype to NHSL1. NHS and NHSL1 only share 30.3% conservation on the amino acid level (new Fig. S14B) and therefore we predict that any potential compensation between these two family members would be only partial.

4. Fig 5A does not match the description as mentioned in the text. The authors need to clear if they used myc-NHSL1 or EGFP-NHSL1 and alter the text/image accordingly.

We would like to apologise and thank this referee for pointing out this mistake. The description in the main results text was correct but not in the figure legend. We indeed used Myc-NHSL1 for this experiment and have changed the text accordingly.

The authors need to show a Coomassie gel for the GST fragments isolated in Fig 5C.

We have now included as new Fig. S9B a Coomassie gel showing the GST fragments used in the far-western experiment in old Fig. 5C (new Fig. S9A).

Can the authors state why they see smears instead of single bands in Fig 5C?

Both fragment 4 and 5 appear to be not very stable fragments and show multiple degradation products as can be seen in the Coomassie blot (new Fig. S9B). NHSL1 is predicted to be mostly disordered and thus fragments without clear domain boundaries are likely to be unstable and degrade. Nevertheless, it can be seen in the far-western overlay experiment in old Fig. 5C (new Fig. S9A) that there is distinct binding to the highest molecular weight bands and several bands with slightly reduced molecular weight which would all still include the Abi SH3 domain binding sites.

Figure 5D: The pulldown efficiency is overall very weak, which makes the result that mutated NHSL1 (site 2+3, site 1+2+3) binds less efficiently to Abi1 questionable. I suggest optimizing the pulldown efficiency by refraining from overexpression of all WAVE complex components simultaneously and instead focusing on the interaction of mutated versions of NHSL1 and Abi1 by sole ectopic expression of those.

We agree with this referee that the GFP-NHSL1 pulldown bands appear weak compared to the GFP-NHSL1 lysate lanes. The reason is that efficient blotting of large size proteins above 200 KDa is not trivial. Since then we have improved our blotting technique to obtain more consistent results

and replaced the pulldown experiment (old Fig 5D) with a new Fig 5G and followed the advice of this referee and referee 1 and only tested absence of interaction between NHSL1 and Abi.

5. The authors need to have similar expression of the various fragments in Fig 7B in order to conclude that Fragment 1 of NHSL1 is the major interactor of Rac1.

The reason is unequal expression or stability of fragments. This is consistent through all repeats of this experiment. This unequal stability of fragments is impossible to rectify. Therefore, we took expression levels into account when drawing conclusions on interaction strength:

In old Fig. 6G and new Fig. 3H the fragment 2 is more weakly expressed yet robustly pulled down whereas the fragment 3 is more strongly expressed but weakly pulled down by dominant active Rac suggesting that fragment 2 is binding more strongly than fragment 3 to dominant active Rac. Since this conclusion takes expression levels of fragments into account we believe that the conclusions are valid despite differences in expression levels.

In old Fig. 7B and new Fig. S7B fragments 2 and 9 are more weakly expressed than all other fragments. When taking expression levels into account fragments 1, 6, and 9 are pulled down to roughly equal extend by dominant active Rac. Since this conclusion takes expression levels of fragments into account we believe that the conclusions are valid despite differences in expression levels.

Also, it would be good to express Rac1 K17N (DN mutant) to show that it hampers localization and interaction of NHSL1 in membrane and with Abi respectively.

In our hands co-expression of tagged-NHSL1 and DN-Rac appears to reduce expression of NHSL1 and thus makes it difficult to assess its effect on interactions quantitatively. We are not sure what we would learn from the expression of DN-Rac on localisation of NHSL1 to lamellipodia since DN-Rac would cause the disappearance of these membrane protrusions.

6. Fig 8A seems incomplete and do not contribute significantly to the major conclusions of the paper. The authors can put it in the supplementary figure or figure out the site and better characterize this PTM of NHSL1.

We agree with this referee and referee 1 that the regulation of NHSL1 by S-palmitoylation does not fit well with the conclusions of this paper. Thus, we removed the data from this manuscript and will revisit the analysis of the regulation of NHSL1 by S-palmitoylation in future studies.

Fig 8B: The authors show here that expression of dominant-active Rac increases binding between NHSL1 and the WAVE complex. This interpretation, especially in consideration of the small difference between DA Rac and vector control, is not valid unless the authors show respective expression levels and pulldown efficiencies of NHSL1 in both conditions. At this stage it cannot be excluded that the pulldown of NHSL1 was less efficient in case of vector control, which would explain reduced binding to the WAVE complex independent of RAC activity. Moreover, in fig 4D the authors have had a descent pull down of WAVE2 in NHSL1 IP samples while in this fig (8B), it seems to be dependent on the activity of DA Rac? Can the authors comment on this? Also, it would be good to have a DN Rac control for Fig 8B.

We agree with this referee and referee 1 that the regulation of the interaction between NHSL1 and the Scar/WAVE complex by active Rac needs to be more thoroughly investigated. However, we believe that it is more important to strengthen the core arguments in this manuscript as requested by all referees. Therefore, we will revisit the detailed analysis of the regulation of the interaction between NHSL1 and the Scar/WAVE complex by active Rac later and this will form a large part of a future study.

It is confusing whether Fig 8D, E were done in KO background or WT background? It would be good to do these experiments in KO background.

These experiments were done in WT background and are thus overexpression experiments. We agree with this referee and referee 1 that it would be informative to repeat them in the KO background and thus have performed these rescue experiments and analysed these cells in random cell migration assays (new Fig. 6 B-D) and found that the increase in cell migration speed can be reduced and the phenotype thus rescued by re-expression of wild type NHSL1 but not by the NHSL1 mutant cDNA which cannot interact with the Scar/WAVE complex.

Figure 8E: Instead of lamellipodia intensity/area I would rather suggest using the term Lifeact intensity/ lamellipodial area, because it is the Lifeact signal that was measured.

We agree with this referee that the term lifeAct-EGFP intensity/lamellipodial area is more accurate and have changed the graph in old Fig. 8E (new Fig. 7F) accordingly.

Moreover, it would be good repeating those measurements on glutaraldehyde-fixed and phalloidin-stained cells. As the Lifeact expression level itself impacts on the outcome of the measurement, the authors should preferentially use phalloidin to visualize endogenous F-actin in lamellipodia of cells with diverse genetic backgrounds.

We agree with this referee and referee 2 that this would be informative and have now quantified Alexa488-phalloidin labelled F-actin intensity in the lamellipodium of control and NHSL1 CRISPR KO cells and observed an increase in intensity per area in the NHSL1 CRISPR 2 and 21 cells (new Fig. 7E).

Since we compare NHSL1 overexpressing cells with control cells (new Fig. 7F), any variability in lifeAct-EGFP expression levels should affect both cell populations equally. Because we are comparing the difference between lifeAct-EGFP intensities in these populations and not absolute intensities, we believe that our data in new Fig. 7F is valid.

7. Figure 9D and Figure S5: Both NHSL1 CRISPR clones appear to have an untypically elongated cell shape compared to B16-F1 controls.

We agree with this referee that the examples that we show appear to be elongated. We have now quantified cell shape parameters comparing wild-type with NHSL1 CRISPR KO cells and found no significant difference in regard to cell shape between them. We include the analysis here for the referees.

C Furthermore, activated Arp2/3 complex in Figure 9D seems to be concentrated within

$$\text{Aspect Ratio} = \frac{r_{major}}{r_{minor}} \quad \text{Roundness Index} = \frac{4}{\pi} \cdot \frac{A}{r_{major}^2}$$

Suppl. Figure for referees: (A,B) The aspect ratio (A) and roundness index (B) do not differ significantly between control and the two CRISPR lines. Data points represent individual cell measurements (n = 30 for each condition), with the mean ± SEM displayed. One-way ANOVA with Dunnett's multiple comparison was used in both cases: ns = not significant, P > 0.05. (C) Mathematical definitions of the two chosen morphology parameters; r_{major} = length of major axis, r_{minor} = length of minor axis, A = area of the fitted ellipse.

the cell body and not at the cell periphery as would be expected. The authors should comment on that.

We have shown that NHSL1 localises to the leading edge of cells by expressing EGFP-tagged NHSL1 as well as endogenous NHSL1 protein using two different antibodies (monoclonal as well as polyclonal). We have added new data indicating that the observed localisation of the EGFP tagged NHSL1 is not due to space filling at potential ruffles at the leading edge (new Fig S1D). In addition, both immunofluorescence analysis as well as EGFP tagged live cell imaging suggests that NHSL1 also localises to vesicles. FRET-FLIM is currently a very slow technique with each image taking 5 minutes to capture thus precluding fast live cell imaging. In this FRET-FLIM technique the activity of a biosensor like our Arp2/3 biosensor is measured as an average over the entire cell. We agree with this referee that looking at the lifetime images, it appears that Arp2/3 is activated on vesicles in addition to the leading edge of cells (new Fig. 8D). We attempted to quantify the activation of the Arp2/3 complex using our biosensor at the leading edge of the NHSL1 CRISPR cell lines, though the accuracy of this data is currently limited by the resolution of our FRET-FLIM microscope. Nevertheless, we observed a trend for an increase of Arp2/3 activity in the NHSL1 CRISPR 2 and 21 cell lines compared to control which did not reach significance (new Fig. S13H). Thus, this non-

significant trend agrees with a potential role for NHSL1 to inhibit Scar/WAVE-Arp2/3 activity at the leading edge. In addition, NHSL1 may inhibit Scar/WAVE-Arp2/3 activity at vesicles. For good reasons (technical limitations) most labs using FRET-FLIM do not attempt to measure activity at sub-cellular resolutions and thus a more definitive answer awaits technological advances in FRET-FLIM.

Further evidence for a role of NHSL1 at the leading edge comes from our new data that Arp2/3 and F-actin intensity at the leading edge is increased in the NHSL1 CRISPR KO cells (new Fig. 7D,E). In addition, we have presented data showing that overexpression of wild-type NHSL1 but not overexpression of NHSL1 cDNA which cannot interact with the Scar/WAVE complex reduces lifeAct-EGFP intensity at the leading edge (new Fig. 7F). This suggests that NHSL1 indeed may function to inhibit Arp2/3 mediated actin nucleation at the leading edge of cells.

8. NHS has been shown to co-localize with Focal Adhesion Complexes and observing NHSL1's role in cell migration, its important to explore whether NHSL1:

a. Localize at focal adhesion complex

b. Does overexpression/ Knock down affect Focal adhesion dynamics, distribution and/or structure.

Indeed, NHS has been shown to localise to focal adhesions in Brooks et al., 2010. However, we do not observe NHSL1 at focal adhesions in B16-F1 cells as can be seen in immunofluorescence as well as in EGFP-NHSL1 live cell imaging (see old and new Fig. 1 and 4). Since NHS and NHSL1 are only 30.3% conserved at the amino acid level (new Fig. S14B), the localisation of NHS may be specific for this family member. Nevertheless, a role of NHS in adhesion and its putative conservation in NHSL1 needs to be more thoroughly investigated but this is beyond the scope of this paper and will be done in a future study.

Other Comments:

1. The authors report a “dynamic localization of NHSL1 to lamellipodia...” Does that mean that there are cells, in which either expressed or endogenously stained NHSL1 is not localized to lamellipodia and if so, what does that mean for its function?

We apologise since we realise that the way we have phrased this can be misunderstood and we have rephrased this in the revised manuscript to “We found that NHSL1 localises to the very edge of protruding lamellipodia...”

2. Fig 1A, B insets do not have scale bars.

We apologise for this omission and have now added scale bars to all insets (see new Fig. 1F,G; 3B-F; 4A-E).

3. Figure 2C: The unit for mean track persistence is missing.

Mean track persistence does not have a unit, since the parameter is a dimensionless value, which is now indicated in the figure legend of new Fig. 2E (old Fig. 2C)

$$\left[\text{Avr} \left(\frac{R}{l} \right) \right] = \left[\frac{\text{length}}{\text{length}} \right]$$

The value is also normalised to the Random walk, but this normalisation factor is also dimensionless. See new Fig. 2C for an explanation of mean track persistence).

3. Figure 2C: Likewise, one of the significance bars is not shown.

We have added the missing significance bar.

4. Fig 3 can be clubbed with Fig 4.

We have now combined the old Fig. 3 with the upper part of old Fig. 4 A-C into new Fig. 4.

It should also have a volume marker control.

We have performed a volume marker control co-expressing mScarlet-I together with EGFP-NHSL1 showing that EGFP-NHSL1 localisation to the leading edge is not due to space filling of the cytoplasm by the fluorescent protein at the very edge of lamellipodia (see new Fig. S1D).

5. Fig 4F, lower blot has different exposure level. This should be accounted or corrected for.

Commercial antibodies display different background levels explaining these differences.

6. Fig 5C, last blot has different exposure levels and could be corrected.

We have now linearly adjusted the contrast of the blots in new Fig. S9A (old Fig. 5C) to display equivalent background levels in all blots of this panel.

7. Fig 6F needs to have (i) Input lanes for myc-Rac, (ii) IgG control lanes.

We apologise for this omission and have now added the input lane for Myc-Rac in new Fig. 3G (old Fig. 6F). We also now show the full blots of this experiment in new Fig. S6A.

Since this experiment is a Myc-trap pulldown with anti-Myc nanobodies from Chromotec, the appropriate control is a Myc-trap pulldown with Myc-Rac-DA and EGFP to control for unspecific binding of EGFP to the Myc-Rac-DA on the Myc-trap beads. We have also now added the reciprocal experiment in which we pulled down EGFP-Rac-DA or EGFP with EGFP-trap beads co-expressed in both cases with Myc-NHSL1 (new Fig. S6B). This experiment also showed specific interaction only between EGFP-Rac-DA and Myc-NHSL1 and no interaction between EGFP and Myc-NHSL1 verifying the specificity of the interaction.

8. Input lane for NHSL1 in Fig S1B shows no protein and has to be corrected.

We have repeated this experiment and show now in new Fig. S1F (old Fig. S1B) input lanes for all tagged proteins.

REVIEWER COMMENTS

Reviewer #1 (Remarks to the Author):

My critiques, including properly controlled interactions and rescue experiments, have been satisfactorily addressed in the revised version. This report is complex. Only part of the phenotype goes through the Scar/WAVE complex (SWC), migration speed, but not persistence, is specifically rescued by the WT, but not the SWC binding mutant, overexpression and KO of NHSL1 produce the same phenotype of increased persistence of cell migration,... So clearly we do not have yet explanations for all the reported effects. However, this manuscript has the merit to report a novel partner of the Scar/WAVE complex and a mutant of NHSL1 that does no longer bind to the SWC, to perform careful quantitative analyses of lamellipodium protrusion and cell migration, and to craft a useful tool for the field with the Arp2/3 biosensor. I recommend publication of the revised version of the manuscript in Nat Comm.

Alexis Gautreau

Reviewer #2 (Remarks to the Author):

In this manuscript, Law et al. demonstrate that NHSL1 is a novel binding partner of the Scar/WAVE complex, which localizes to the edge of lamellipodia. Moreover, NHSL1, which binds to the active Rac, negatively regulates cell migration speed and persistence by restricting Arp2/3 activity. In response to the reviewers' remarks, the authors performed additional experiments and improved their manuscript. However, several additional improvements are necessary before it reaches publication quality. First, the write-up of the manuscript needs to be streamlined. Second, additional experiments, data analysis and/or revision of their statements are required:

1. Some key findings need to be verified with a second cell line (MCF10A). Specifically, Figs. 8F, 10C-D.
2. Quantification of WB data (Fig. 2A). The authors only provided quantification of qPCR data showing reduction of mRNA levels upon NHSL1 knockout. What about the protein levels?
3. Quantification of co-localization patterns shown in Fig. 4.
4. Results central to the authors' conclusions (Figs. 9B-C, 10F, S13H) do not show statistically significant differences.
 - a. Lines 380-383: "Re-expression of wild type NHSL1 (EGFP-NHSL1 WT) or the NHSL1 Scar/WAVE complex binding mutant (EGFP-NHSL1 SW Mut) appeared to increase flow speed compared to the NHSL1 CRISPR knockout lamellipodia but this did not reach significance (Fig. 9A,B)". This statement would have been accurate if the p value was less 0.1 but not less than 0.05. Otherwise, their statement is misleading.
 - b. The same holds for the authors' statement in Lines 390-394.
5. Showing images and providing quantification of Arp2/3 intensity at lamellipodia, as well as co-localization of Arp2/3 with NHSL1 at lamellipodia, would strengthen the authors' argument and

compensate for technical limitations of their FLIM experiments.

6. The experiments performed by the authors still do not directly demonstrate that NHSL1 regulates Arp2/3 through the Scar/WAVE complex. Having said this, the migration experiments they have carried out are interesting. I suggest that they also perform FLIM using their novel Arp2/3 sensor with wild-type NHSL1 or NHSL1 mutated in the Scar/WAVE binding sites.

Minor Comments:

1. Line 295: Add "Mut": ... NHSL1 but not NHSL1 "Mut".

2. Line 1124: "NHSL1 reduces actin retrograde flow speed" should be changed to "NHSL1 knockout reduces actin retrograde flow speed".

Reviewer #3 (Remarks to the Author):

The revised manuscript by Law et al., does not significantly improve the claims put forward in their previous manuscript. There are still apparent technical issues that the authors need to address:

1. Throughout the manuscript the authors provide western blots of different exposures which clouds the overall representation of the results.

2. I am not sure why the authors had to load such high amounts of lysates in Fig2A as evident by the blasting loading control bands. Also, these gels cannot be quantified; coupled with the qPCR results for CRISPR line 21 (Fig 2B), I have serious doubts on the cell lines being used. Moreover from the whole blots it's not evident how specific the antibody is. the authors also did not perform a KO control for the antibody either through WB or through IF to establish the validity of the antibody being used.

3. The overall quality of the western blots need to improve as different figures represent different levels of exposure for the blots. I am not sure why the GFP blot in Fig 5G is having such high exposure?

4. For Fig 7E and several other figures, the authors clearly need to show the images or the phenotype instead of simply putting forward the quantification.

5. The authors compare Arp2/3 levels in Fig S13 through imaging which is a little surprising. This shows a significant decrease in the levels of Arp2/3 in CRISPR line 21 which the authors need to explain. Moreover, the authors need to clearly show images and instead perform comparative western blots to come to any conclusion.

6. The authors fail to explain the regulation of NHSL1 and Scar/Wave through active Rac I, which is an important part of this manuscript.

REVIEWER COMMENTS

Reviewer #1 (Remarks to the Author):

My critiques, including properly controlled interactions and rescue experiments, have been satisfactorily addressed in the revised version. This report is complex. Only part of the phenotype goes through the Scar/WAVE complex (SWC), migration speed, but not persistence, is specifically rescued by the WT, but not the SWC binding mutant, overexpression and KO of NHSL1 produce the same phenotype of increased persistence of cell migration,... So clearly we do not have yet explanations for all the reported effects. However, this manuscript has the merit to report a novel partner of the Scar/WAVE complex and a mutant of NHSL1 that does no longer bind to the SWC, to perform careful quantitative analyses of lamellipodium protrusion and cell migration, and to craft a useful tool for the field with the Arp2/3 biosensor. I recommend publication of the revised version of the manuscript in Nat Comm.

Alexis Gautreau

We thank the referee for this fair assessment and recommendation to publish.

Reviewer #2 (Remarks to the Author):

In this manuscript, Law et al. demonstrate that NHSL1 is a novel binding partner of the Scar/WAVE complex, which localizes to the edge of lamellipodia. Moreover, NHSL1, which binds to the active Rac, negatively regulates cell migration speed and persistence by restricting Arp2/3 activity. In response to the reviewers' remarks, the authors performed additional experiments and improved their manuscript. However, several additional improvements are necessary before it reaches publication quality. First, the write-up of the manuscript needs to be streamlined. Second, additional experiments, data analysis and/or revision of their statements are required:

Following the advice of this referee, we have now performed many additional experiments, additional data analyses and/or revised our statements and streamlined the text of the manuscript.

1. Some key findings need to be verified with a second cell line (MCF10A). Specifically, Figs. 8F, 10C-D.

We would like to bring to the attention of the reviewer that we have already verified the key finding of this paper (i.e. NHSL1 negatively regulates cell migration) in an additional cell line (i.e. MCF10A, see Fig. S5A-C). In addition, the newly added data showing that CRISPR knockout of NHSL1 reduced total cellular F-actin content in B16-F1 cells (Fig. 8C) were confirmed using shRNA-mediated knockdown in the HEK293 cell line (Fig. S16C). In view of the large amount of data already in this manuscript and the current circumstances of limited access to laboratories, we would prefer not to repeat all our other experiments with the MCF10A cell line, especially because this would not provide additional mechanistic insights. We would also like to highlight that many eminent studies in the cell motility field use only one cell line throughout and do not repeat any of their findings in a second cell line. Taking this approach often provides consistency and clarity when elucidating complex cellular mechanisms. Some examples are below:

Schaks et al., 2018 Curr Biol. DOI: 10.1016/J.CUB.2018.10.002

Mueller et al., 2017 Cell, DOI: 10.1016/j.cell.2017.07.051,

Nordenfelt et al., 2017, Nature Comm, DOI: 10.1038/s41467-017-01848-y

2. Quantification of WB data (Fig. 2A). The authors only provided quantification of qPCR data showing reduction of mRNA levels upon NHSL1 knockout. What about the protein levels?

We repeated these blots and used a higher dilution of the HSC70 antibodies since we needed to load sufficient amounts of lysate to be able to visualise NHSL1, which is expressed at relatively low levels in cells and runs at 260 kDa and is thus a challenging protein to efficiently immunoblot. This new blot now replaces the old blot in Fig. 2A and S3A. We have also now quantified these blots and added this quantification as Fig. 2B and moved the qPCR results to Fig. S3E.

In agreement with the qPCR result, this quantification revealed that NHSL1 expression was greatly reduced in clone CRISPR 2 but less reduced in clone CRISPR 21 (Fig. 2A,B S3A,E).

3. Quantification of co-localization patterns shown in Fig. 4.

We generated line scans and calculated the Pearson's correlation coefficients to verify the co-localisation of NHSL1 with Abi1, or Scar/WAVE2 at the very edge of lamellipodia and included them in Fig. S8A-H.

4. Results central to the authors' conclusions (Figs. 9B-C, 10F, S13H) do not show statistically significant differences.

- a. Lines 380-383: "Re-expression of wild type NHSL1 (EGFP-NHSL1 WT) or the NHSL1 Scar/WAVE complex binding mutant (EGFP-NHSL1 SW Mut) appeared to increase flow speed compared to the NHSL1 CRISPR knockout lamellipodia but this did not reach significance (Fig. 9A,B)". This statement would have been accurate if the p value was less 0.1 but not less than 0.05. Otherwise, their statement is misleading.
- b. The same holds for the authors' statement in Lines 390-394.

The key results central to our conclusions show statistically significant differences: Revision 1 Figures 2D,E, 5F, 6A-C, 7B,D-F, 8C,F, 9B,C, 10C,D.

Fig. 9B-C: Our detailed analysis of retrograde flow and F-actin assembly rates show a significant decrease in the NHSL1 CRISPR KO cells compared to wild type control cells. We observed an increase in the mean value of the retrograde flow and F-actin assembly rates in the NHSL1 CRISPR KO cells rescued with either wild type or the Scar/WAVE binding mutant NHSL1 which was not significant. In our manuscript we have not claimed that the rescue was statistically significant, but in such a biological system it is not uncommon that not all phenotypes can be fully rescued. This can potentially be attributed to a lack of precise control of the re-expression levels of NHSL1 in the rescue experiments. We have followed the advice of this referee and rephrased the text in the results section to be more accurate:

"However, particle image velocimetry analysis revealed a significant reduction of F-actin retrograde flow in the NHSL1 CRISPR 2 lamellipodia compared to wild type control lamellipodia (Fig. 9A,B). Re-expression of ~~wild type NHSL1 (EGFP-NHSL1 WT)~~ in the NHSL1 CRISPR 2 cells changed the mean values of the retrograde flow speed in lamellipodia from 1.34 to 1.43 $\mu\text{m}/\text{min}$ ~~or~~ and re-expression of ~~the NHSL1 Scar/WAVE complex binding mutant (EGFP-NHSL1 SW Mut)~~ changed the mean values of the flow speed from 1.34 to 1.51 $\mu\text{m}/\text{min}$ ~~appeared to increase compared to the NHSL1 CRISPR knockout in lamellipodia~~ but ~~this~~ these changes were ~~did~~ not ~~reach~~ significant~~ee~~ (Fig. 9A,B)."

“...we surprisingly observed a significant reduction of F-actin assembly rate in NHSL1 CRISPR 2 lamellipodia compared to wild-type control lamellipodia (Fig. 9A,C). ~~Again, re-expression of wild type NHSL1 (EGFP-NHSL1 WT) or the NHSL1 Scar/WAVE complex binding mutant (EGFP-NHSL1 SW Mut) changed the mean values of the F-actin assembly rate for both from 4.04 to 4.30 $\mu\text{m}/\text{min}$ appeared to increase assembly rate compared to the NHSL1 CRISPR knockout lamellipodia but this change was did not reach significantee~~ (Fig. 9A,C).

Fig. 10F: We again followed the advice of this referee and rephrased our results section to be more precise:

~~“However, w~~When we compared the standard deviation (S.D.) of the lamellipodia protrusion speed from all frames of a movie as a readout for the temporal stability of protrusions over time, we observed a change from S.D. = 0.376 $\mu\text{m}/\text{min}$ for WT Myc Control to S.D. = 0.265 $\mu\text{m}/\text{min}$ for NHSL1 CRISPR 2 knockout cells. ~~apparent reduction of this parameter in the NHSL1 CRISPR 2 knockout cells compared to wild type control cells. This reduced standard deviation~~This standard deviation of the lamellipodia protrusion speed of the NHSL1 CRISPR 2 knockout cells was significantly ~~rescued~~ increased in the NHSL1 CRISPR 2 cells re-expressing wild type but not the NHSL1 Scar/WAVE complex binding mutant (Fig. 10F).”

Fig. S13H (in Revision 1); this is now Figure S14H in the current revision 2: In Fig. 8D we show that Arp2/3 intensity is significantly increased in the lamellipodium of NHSL1 CRISPR 2 knockout cells compared to wild type control cells suggesting that NHSL1 functions to reduce Arp2/3 activity in the lamellipodium. This is consistent with our findings that Arp2/3 activity in the entire cell is also significantly increased in CRISPR 2 KO cells compared to wild type control cells (Fig. 7F). To further support this data, we attempted to quantify FRET efficiency at the leading edge by manually outlining lamellipodia. This approach is hampered by the technical limitations of the FRET-FLIM microscope but nevertheless we observed a change from 0.01 to 3.97 FRET efficiency for Control vs. CRISPR 2: ns, $p=0.6914$ and 0.01 to 8.87 for Control vs. CRISPR 21: ns, $p=0.2502$ for Arp2/3 activity in the lamellipodium.

We followed the advice of this referee and amended the results text to more accurately describe this additional supporting data shown now in Fig. S14H.

~~“Since NHSL1 and the Scar/WAVE complex co-localise at the very edge of lamellipodia (Fig. 4) To quantify Arp2/3 activity in lamellipodia we sought to estimate Arp2/3 biosensor FRET efficiency at the leading edge by manually outlining lamellipodia. This approach is limited by the resolution of the FRET-FLIM microscope, but nevertheless we observed a change in FRET efficiency from 0.01 to 3.97% for Control vs. CRISPR 2 ($p=0.6914$) and 0.01 to 8.87% for Control vs. CRISPR 21 ($p=0.2502$) for Arp2/3 activity in the lamellipodium. Neither changes are significant also revealed a trend for increased Arp2/3 activity in both NHSL1 CRISPR 2 and 21 cell lines compared to control B16-F1 cells (Fig. S1413H). which is consistent with the increased Arp2/3 intensity in the lamellipodium in the NHSL1 CRISPR cells (Fig. 78D).”~~

5. Showing images and providing quantification of Arp2/3 intensity at lamellipodia, as well as co-localization of Arp2/3 with NHSL1 at lamellipodia, would strengthen the authors' argument and compensate for technical limitations of their FLIM experiments.

Following the advice of this referee we are showing images of Arp2/3 staining of lamellipodia in wild type and NHSL1 CRISPR 2 cells (Fig. S15A) and we are providing

quantification of this Arp2/3 intensity at the leading edge which shows a significant increase in Arp2/3 intensity in lamellipodia upon NHSL1 CRISPR knockout (Fig. 8D).

We would like to point out that Arp2/3 becomes incorporated into the entire width of the lamellipodium whereas NHSL1 is restricted to the very edge of lamellipodia (see detailed co-localisation study comparing the localisation of NHSL1 with components of the Scar/WAVE complex in lamellipodia (see Fig. 4) and the additional line scan quantifications Fig. S8A-H) and thus we don't believe adding images of co-localisation of NHSL1 with Arp2/3 in lamellipodia would provide helpful additional information.

6. The experiments performed by the authors still do not directly demonstrate that NHSL1 regulates Arp2/3 through the Scar/WAVE complex. Having said this, the migration experiments they have carried out are interesting. I suggest that they also perform FLIM using their novel Arp2/3 sensor with wild-type NHSL1 or NHSL1 mutated in the Scar/WAVE binding sites.

As the referee noted, we have provided good evidence that NHSL1 functions via the Scar/WAVE complex to negatively regulate cell migration efficiency (Fig. 6). In addition, overexpression of NHSL1 caused a reduction in lamellipodial F-actin intensity but this was not the case when we overexpressed the NHSL1 Scar/WAVE binding mutant (Fig. 8F). We also show that NHSL1 KO caused increased lamellipodial Arp2/3 levels, which are a readout for Arp2/3 increased activity in the lamellipodium (see above and Fig. 8D; S15A). Since the Scar/WAVE complex is the only Arp2/3 activator in lamellipodia, it is reasonable to conclude that NHSL1 functions to inhibit Arp2/3 through the Scar/WAVE complex.

In addition, we have followed the advice of this referee and also performed the suggested FLIM rescue experiment using our novel Arp2/3 biosensor with wild type NHSL1 or NHSL1 mutated in the Scar/WAVE binding sites expressed in NHSL1 CRISPR KO B16-F1 cells.

We added the following new text to our manuscript: "To test whether the observed phenotypes in the NHSL1 CRISPR knockout clones were not due to off-target effects, we re-expressed Myc-tagged wild type NHSL1 (Myc-NHSL1 WT) or the NHSL1 Scar/WAVE complex binding mutant (Myc-NHSL1 SW Mut) in the NHSL1 CRISPR 2 cell line and also expressed our Arp2/3 biosensor and quantified FRET by FLIM. Again, we observed a significant increase in Arp2/3 activity in the NHSL1 CRISPR 2 cell line compared to control B16-F1 cells, which was rescued with the wild type NHSL1. This confirmed that indeed the observed increase in Arp2/3 activity in the NHSL1 CRISPR 2 cells was due to loss of NHSL1 and not due to off target effects (Fig. S14G). We also found a significant rescue with the NHSL1 Scar/WAVE complex binding mutant (Fig. S14G) suggesting that in whole cells NHSL1 affects Arp2/3 activity through other mechanisms in addition to its interaction with the Scar/WAVE complex."

As expected, these additional experiments clearly validate our previous experiments because we can rescue the NHSL1 KO with the wild type NHSL1 (Fig. S14G). The rescue with the NHSL1 Scar/WAVE binding mutant may be explained either by the inherent inability to precisely finetune expression levels during re-expression or by an as yet undiscovered additional function of NHSL1 in its vesicular location in the cytoplasm on Arp2/3 activity. Nevertheless, all our other data clearly suggests that NHSL1 functions via the Scar/WAVE complex to control efficiency of migration (Fig. 6) and that NHSL1 functions to inhibit Arp2/3 activity and reduce F-actin levels in lamellipodia (Fig. 8D-F).

Minor Comments:

1. Line 295: Add "Mut": ... NHSL1 but not NHSL1 "Mut".

We would like to thank the referee for pointing out this mistake which we have rectified:

“This indicates that wild type NHSL1 but not NHSL1 **SW Mut** (that cannot bind to the Scar/WAVE complex) can rescue the NHSL1 knockout phenotype and suggests that NHSL1 negatively regulates cell migration speed via an interaction with the Scar/WAVE complex.”

2. Line 1124: “NHSL1 reduces actin retrograde flow speed” should be changed to “NHSL1 knockout reduces actin retrograde flow speed”.

We would like to thank the referee for pointing out this mistake which we have rectified:

“Figure 9. NHSL1 increases reduces actin retrograde flow speed and F-actin assembly rate”

Reviewer #3 (Remarks to the Author):

The revised manuscript by Law et al., does not significantly improve the claims put forward in their previous manuscript. There are still apparent technical issues that the authors need to address:

1. Throughout the manuscript the authors provide western blots of different exposures which clouds the overall representation of the results.

Following the advice of this referee we have repeated some blots in order to improve them (Fig. 2A, S3A). We would like to bring to the attention of the referee that it is standard in the field to optimise exposure times for western blots with different antibodies. Of course, the same exposure time was used for the same antibody for different lysates on the same blot. Only linear contrast adjustment has been performed as is standard in the field.

2. I am not sure why the authors had to load such high amounts of lysates in Fig2A as evident by the blasting loading control bands. Also, these gels cannot be quantified; coupled with the qPCR results for CRISPR line 21 (Fig 2B), I have serious doubts on the cell lines being used.

Following the advice of this referee we have repeated these blots (see also comment of referee 2 point 2) and used a higher dilution of the HSC70 antibodies since we needed to load sufficient amounts of lysate to be able to visualise NHSL1, which is expressed at relatively low levels in cells and runs at 260 kDa and thus is a challenging protein to efficiently immunoblot. This new blot now replaces the old blot in Fig. 2A and S3A. We have also quantified these blots and added this quantification as Fig. 2B and moved the qPCR results to Fig. S3E.

In agreement with the qPCR result, this quantification revealed that NHSL1 expression was greatly reduced in clone CRISPR 2 but less reduced in clone CRISPR 21 (Fig. 2A,B S3A,E). The partially reduced mRNA and protein levels (Fig. 2A,B; S3A,E) in the CRISPR 21 cell line are consistent with Cas9 induced indels only in some alleles (Fig S2A), whereas the absence of NHSL1 wild type alleles (Fig S2B) and mRNA in the CRISPR 2 cell line (Fig. S3E) indicates that the CRISPR 2 line represents a full NHSL1 knockout.

Moreover from the whole blots it's not evident how specific the antibody is. the authors also did not perform a KO control for the antibody either through WB or through IF to establish the validity of the antibody being used.

We apologise as this was not sufficiently clear, but we performed a KO control for our NHSL1 antibody through western blot in Fig. 2A; S3A. In order to improve these, we have now repeated these blots Fig. 2A,B and S3A and quantified them (Fig. 2B) (see above).

Furthermore: 1) two different antibodies (i.e. rabbit polyclonal and mouse monoclonal) were utilised to confirm the cellular localisation of NHSL1 (Fig. 4, S8); and 2) NHSL1 localisation data were also obtained with antibody-independent techniques, i.e. by the expression of N-terminally and C-terminally EGFP tagged NHSL1 (Fig. 1, 4, S1).

3. The overall quality of the western blots need to improve as different figures represent different levels of exposure for the blots.

Please refer to our response to comment 1.

I am not sure why the GFP blot in Fig 5G is having such high exposure?

Blotting at 260 kDa is not very efficient and thus we had to increase the contrast (applied linearly across the entire blot as is customary in this field).

4. For Fig 7E and several other figures, the authors clearly need to show the images or the phenotype instead of simply putting forward the quantification.

As also suggested by referee 2 point 5, we have provided new images of Arp2/3 and F-actin (phalloidin) staining for all experiments to complement our quantification (see Fig. S15A,B; Fig. S16A; Fig. S17E).

5. The authors compare Arp2/3 levels in Fig S13 through imaging which is a little surprising. This shows a significant decrease in the levels of Arp2/3 in CRISPR line 21 which the authors need to explain. Moreover, the authors need to clearly show images and instead perform comparative western blots to come to any conclusion.

We have followed the advice of this referee and provided images (Fig. S15A) to complement our quantification of Arp2/3 levels in the NHSL1 CRISPR KO cells (Fig. S15C). As suggested, to complement our immunofluorescence analysis, we performed additional western blots which also revealed a decrease in Scar/WAVE2 and ARPC2 (Arp2/3 complex) levels upon NHSL1 knockout (Fig. S15D-F). As a consequence of the reduction of the total cellular Arp2/3 complex levels it is not surprising that we also found that total cellular F-actin content was reduced (see new Fig. 8C; and corresponding images see Fig. S16A). This phenotype was rescued in the NHSL1 CRISPR 2 cells re-expressing wild type Myc-NHSL1 but not Myc-NHSL1 SW Mut (Fig. 8C; S16A) suggesting that NHSL1 controls cellular Arp2/3 activity, and consequential Arp2/3 complex and F-actin levels, by binding to the Scar/WAVE complex. We verified this phenotype in another cell line (HEK293) and again observed a significant reduction in total cellular F-actin content upon knockdown of NHSL1 and conversely increased total cellular F-actin content upon overexpression of EGFP-NHSL1 (see new Fig. S16 B,C).

A potential reason for these observations may be that in the NHSL1 KO cells, as a compensation for constant overactivation of the Scar/WAVE and Arp2/3 complexes, the Scar/WAVE and Arp2/3 complex levels may be downregulated through ubiquitination and degradation through the proteasome. Such a proteosomal degradation of the Scar/WAVE complex and the Arp2/3 complex through ubiquitination have been previously described in

31. Ichikawa, D., Mizuno, M., Yamamura, T. & Miyake, S. GRAIL (gene related to anergy in lymphocytes) regulates cytoskeletal reorganization through ubiquitination and degradation of Arp2/3 subunit 5 and coronin 1A. *J. Biol. Chem.* **286**, 43465–43474 (2011).
32. Joseph, N. *et al.* A conformational change within the WAVE2 complex regulates its degradation following cellular activation. *Sci. Rep.* **7**, 44863 (2017).

We have added the following paragraph to our manuscript (including the two new citations above):

“Similarly, western blot analysis revealed that both total cellular Arp2/3 complex (subunit ARPC2) and Scar/WAVE2 levels were reduced in the NHSL1 CRISPR 2 line (Fig. S15D-F), suggesting that persistent overactivation of the Scar/WAVE and Arp2/3 complexes in the absence of NHSL1 may lead to their proteasomal degradation^{31,32}. Consistent with a reduction in total Arp2/3 levels, we observed that the total cellular F-actin content was significantly reduced in NHSL1 CRISPR 2 cells compared to wild type cells. This phenotype was rescued by re-expressing wild type Myc-NHSL1 but not Myc-NHSL1 SW Mut in the NHSL1 CRISPR 2 cells (Fig. 8C; S16A) suggesting that NHSL1 controls cellular F-actin levels by binding to the Scar/WAVE complex. Similarly, we observed a significant reduction in total cellular F-actin content upon knockdown of NHSL1 and, conversely, an increased total cellular F-actin content upon overexpression of EGFP-NHSL1 in HEK293 cells (Fig. S16 B,C).”

6. The authors fail to explain the regulation of NHSL1 and Scar/Wave through active Rac I, which is an important part of this manuscript.

We agree with this referee that the regulation of the interaction between NHSL1 and the Scar/WAVE complex by active Rac needs to be more thoroughly investigated. However, we believed that it was more important to strengthen the core arguments in this manuscript (which already contains a considerable amount of data) as requested by all referees. Therefore, we will revisit the detailed analysis of the regulation of the interaction between NHSL1 and the Scar/WAVE complex by active Rac later and this will form a large part of a future study.

REVIEWERS' COMMENTS

Reviewer #2 (Remarks to the Author):

The authors performed additional experiments and improved their re-revised manuscript. Although I sincerely appreciate the authors' efforts, I am concerned with the robustness of their findings. Importantly, the two clones give different results in Fig. 8D (lamellipodia Arp2/3 intensity), and Fig. 11C (longest uninterrupted lamellipodia length). Especially, the latter figure is critical to the authors' main conclusion, and they seem to dismiss the results obtained from one of the two clones (clone 21). In light of these divergent findings between clones 2 and 21, this reviewer's question is pertinent to whether their conclusion is clone-specific.

Reviewer #3 (Remarks to the Author):

In the revised manuscript the authors have satisfactorily addressed most of my concerns and have improved the quality of their western blot analysis in the various figures. Though the authors have not addressed the mechanism of active Rac1 mediated control of NHSL1 and Scar/Wave, the data presented is interesting and should interest general audience in the cell migration field. I advocate publication of the manuscript in the current form in Nature Communication.

Rajarshi Chakrabarti

Point-by-point response to the reviewers' comments to revision 2

REVIEWERS' COMMENTS

Reviewer #2 (Remarks to the Author):

The authors performed additional experiments and improved their re-revised manuscript. Although I sincerely appreciate the authors' efforts, I am concerned with the robustness of their findings. Importantly, the two clones give different results in Fig. 8D (lamellipodia Arp2/3 intensity), and Fig. 11C (longest uninterrupted lamellipodia length). Especially, the latter figure is critical to the authors' main conclusion, and they seem to dismiss the results obtained from one of the two clones (clone 21). In light of these divergent findings between clones 2 and 21, this reviewer's question is pertinent to whether their conclusion is clone-specific.

We would like to point out that we clearly stated in our manuscript that the CRISPR 21 cell line is only knockout in some alleles and thus is a partial knockout whereas the CRISPR2 cell line represents a full NHSL1 knock out.

“Clonal cell lines were tested by western blot and qPCR analysis, which revealed that NHSL1 expression was greatly reduced in clone CRISPR 2 but less reduced in clone CRISPR 21 (Fig. 2A,B S3A,E). The partially reduced mRNA and protein levels (Fig. 2A,B; S3A,E) in the CRISPR 21 cell line are consistent with Cas9 induced indels only in some alleles (Fig S2A), whereas the absence of NHSL1 wild type alleles (Fig S2B) and mRNA in the CRISPR 2 cell line (Fig. S3E) indicates that the CRISPR 2 line represents a full NHSL1 knockout.”

Thus, we would expect that the CRISPR 21 cell line show weaker phenotypes compared to the CRISPR 2 cell line. Indeed, in Fig. 8D (lamellipodia Arp2/3 intensity) the mean for the CRISPR 21 cell line (1.104) is higher than the mean of the control wild-type cells (0.9789) but this change is not significant whereas for the CRISPR 2 it is (1.149) and the change is significant.

To more precisely report the results from the two clones we modified the text and included the new text shown in red:

“We found that relative Arp2/3 complex (ARPC2) intensity was significantly increased in the lamellipodium in the NHSL1 CRISPR 2 cells (the mean changed from 0.9789 to 1.149). Also, the mean of the NHSL1 CRISPR 21 cells changed from (0.9789) for the control wild type cells to (1.104) but the latter change was not significant (Fig. 8D; S15A). The different results from these two CRISPR KO clones are consistent with our finding that the NHSL1 CRISPR 21 cell line represents a partial knockout whereas the NHSL1 CRISPR 2 cell line represents a full knockout (Suppl. Fig. 2).”

In the Fig. 10C (longest uninterrupted lamellipodia length), the mean of the longest uninterrupted lamellipodia length for the wild type control cell is (11.72 μ m). This is not changed in the CRISPR 21 cells (11.91 μ m) whereas for the CRISPR 2 cells it is significantly increased (18.22 μ m). To more accurately describe these results, we have modified the text in our revised manuscript:

“We thus analysed lamellipodia stability and observed that the NHSL1 CRISPR 2 cells displayed a significantly increased uninterrupted lamellipodium while this parameter was not changed in the NHSL1 CRISPR 21 cell line (Fig. 10a,c, Suppl. Movie 9). The different results from these two CRISPR KO clones are consistent with our finding that the NHSL1 CRISPR 21 cell line represents a partial knockout whereas the NHSL1 CRISPR 2 cell line represents a full knockout (Suppl. Fig. 2). Conversely, cells

overexpressing EGFP-NHSL1 (Fig. 10b,d, Suppl. Movie 10) showed a significantly decreased uninterrupted lamellipodium compared to controls.”

We would like to point out that we verified our data from the NHSL1 CRISPR 2 cell line with both overexpression experiments as well as rescue experiments:

1. The overexpression phenotype is as expected the opposite as the NHSL1 CRISPR 2 cell line phenotype.
2. We have verified these results by re-expressing wild type or Scar/WAVE binding mutant NHSL1 cDNA in the NHSL1 CRISPR 2 cell line:

“When we compared the standard deviation (S.D.) of the lamellipodia protrusion speed from all frames of a movie as a readout for the temporal stability of protrusions over time, we observed a change from S.D. = 0.376 $\mu\text{m}/\text{min}$ for WT Myc Control to S.D. = 0.265 $\mu\text{m}/\text{min}$ for NHSL1 CRISPR 2 knockout cells. This standard deviation of the lamellipodia protrusion speed of the NHSL1 CRISPR 2 knockout cells was significantly increased in the NHSL1 CRISPR 2 cells re-expressing wild type but not the NHSL1 Scar/WAVE complex binding mutant (Fig. 10f).”

Taken together, the NHSL1 CRISPR 21 cell line shows weaker phenotypes compared to the NHSL1 CRISPR 2 cell line, but this is in agreement with the CRISPR 21 cell line being only a partial knockout. In addition, all major phenotypes observed with the CRISPR 2 cell line could be verified by re-expressing the wild type NHSL1 cDNA.

Reviewer #3 (Remarks to the Author):

In the revised manuscript the authors have satisfactorily addressed most of my concerns and have improved the quality of their western blot analysis in the various figures. Though the authors have not addressed the mechanism of active Rac1 mediated control of NHSL1 and Scar/Wave, the data presented is interesting and should interest general audience in the cell migration field. I advocate publication of the manuscript in the current form in Nature Communication.

Rajarshi Chakrabarti

We thank the referee for this fair assessment and recommendation to publish.